# A Framework of SO(3)-equivariant Non-linear Representation Learning and its Application to Electronic-Structure Hamiltonian Prediction

## Abstract

We propose both a theoretical and a methodological framework to address a critical challenge in applying deep learning to physical systems: the reconciliation of non-linear expressiveness with SO(3)-equivariance in predictions of SO(3)-equivariant quantities, such as the electronic-structure Hamiltonians. Inspired by covariant theory in physics, we present a solution by exploring the mathematical relationships between SO(3)-invariant and SO(3)-equivariant quantities and their representations. We first construct theoretical SO(3)-invariant quantities derived from the SO(3)-equivariant regression targets, and use these invariant quantities as supervisory labels to guide the learning of high-quality SO(3)-invariant features. Given that SO(3)-invariance is preserved under non-linear operations, the encoding process for invariant features can extensively utilize non-linear mappings, thereby fully capturing the non-linear patterns inherent in physical systems. Building on this, we propose a gradient-based mechanism to induce SO(3)-equivariant encodings of various degrees from the learned SO(3)-invariant features. This mechanism can incorporate non-linear expressive capabilities into SO(3)-equivariant representations, while theoretically preserving their equivariant properties as we prove, establishing a strong foundation for regressing complex SO(3)-equivariant targets. We apply our theory and method to the electronic-structure Hamiltonian prediction tasks, experimental results on eight benchmark databases covering multiple types of systems and challenging scenarios show substantial improvements on the state-of-the-art prediction accuracy of deep learning paradigm. Our method boosts Hamiltonian prediction accuracy by up to 40% and enhances downstream physical quantities, such as occupied orbital energy, by a maximum of 76%. Our method also significantly promotes the acceleration performance for the convergence of traditional Density Functional Theory methods.

## 1 Introduction

Electronic structure calculations provide crucial insights into the behavior of electrons in condensed matter, forming the foundation for predicting material properties such as conductivity, magnetism, optical response, and chemical reactivity, playing a key role in applications ranging from semiconductors to renewable energy and catalysis, ultimately helping to reshape our lives. The core of electronic structure calculations lies in solving the Hamiltonian matrices, from which critical physical quantities such as orbital energy, band structure and electronic wavefunction are induced. However, traditional Density Functional Theory (DFT) methods (Hohenberg & Kohn, 1964; Kohn & Sham, 1965) suffer from the extremely time-consuming self-consistent field (SCF) algorithms used to obtain the Hamiltonians. These algorithms involve the exhaustive iterative diagonalization of large matrices, with each diagonalization scaling with a high computational complexity of $\mathcal{O}(N^3)$, where $N$ is the number of atoms in the system. With advantages in computational complexity and generalization capabilities, the deep learning paradigm has significantly advanced physics research (Zhang et al., 2023). In the task of predicting the Hamiltonians, deep learning approaches (Unke et al., 2021; Gu et al., 2022; Li et al., 2022; Yu et al., 2023b; Gong et al., 2023) bypass the SCF process,

dramatically reducing the computational complexity of solving Hamiltonians, opening up new possibilities for analyzing extremely large atomic systems, enabling efficient materials simulation and design that was previously unimaginable. See Appendix B for more details.

However, deep learning methods still face substantial challenges when processing physical systems. To align with fundamental physical laws, these methods must strictly adhere to symmetry principles. For example, physical quantities such as force fields and electronic-structure Hamiltonians must be equivariant under 3D rotational transformations, i.e., elements from the SO(3) group. Besides, the calculation of physical quantities calls for high numerical accuracy, necessitating that the neural networks possess strong capabilities to express the complex non-linear mappings from atomic structures to the regression targets. However, it is challenging for deep learning methods to simultaneously ensure strict SO(3)-equivariance and high numerical accuracy on modeling physical systems. The root cause of this problem lies in the conflict between SO(3)-equivariance and non-linear expressiveness: specifically, directly applying non-linear activation functions on SO(3)-equivariant features (with degree $l \geq 1$) may lead to the loss of equivariance, while bypassing non-linear mappings severely restricts the network's expressive capabilities and thereby lowering down the achievable accuracy. This issue is commonly found in physics-oriented machine learning tasks that demand both strict equivariance and fine-grained generalization performance, as analyzed by Zitnick et al. (2022). Some recent efforts have been made to alleviate this issue (Zitnick et al., 2022; Passaro & Zitnick, 2023; Wang et al., 2024c; Yin et al., 2024) while compromising strict SO(3)-equivariance.

To address the equivariance-expressiveness dilemma, we make theoretical and methodological explorations on unifying strict SO(3)-equivariance with strong non-linear expressiveness within the realm of deep representation learning. We are inspired by the insight that invariant quantities in transformation (Resnick, 1991) often reflect the mathematical nature of physical laws and can induce other quantities with equivariant properties, and aim to extend the relationship between invariance and equivariance from specific physical quantities to representation learning of neural networks. From the perspective of deep representation learning, the attribute that invariance is preserved under non-linear operations is a significant advantage, given its compatibility with non-linear expressive capabilities. Built upon these insights, we propose a solution to the equivariance-expressiveness dilemma by intensively exploring and making use of the intrinsic relationships between SO(3)-invariant and SO(3)-equivariant quantities and representations: we first dedicate efforts to learning high-quality SO(3)-invariant features with ample non-linear expressiveness, and subsequently, we derive SO(3)-equivariant non-linear representations and the target quantities from these SO(3)-invariant ones. Specifically:

First, we propose a theoretical construct of SO(3)-invariant quantities, namely $tr(\mathbf{Q} \cdot \mathbf{Q}^\dagger)$, where $tr(\cdot)$ signifies the trace operation, $\dagger$ denotes the conjugate transpose operation, and $\mathbf{Q}$ denotes the SO(3)-equivariant regression targets we aim to predict (e.g. the electronic Hamiltonian in this work). A significant advantage of these SO(3)-invariant quantities lies in the fact that they are directly derived from the SO(3)-equivariant target labels and can serve as unique supervision labels for the effective learning of informative SO(3)-invariant features that capture the intrinsic symmetry properties of the mathematical structure of $\mathbf{Q}$ without requiring additional labeling resources.

Second, we propose a gradient-based mechanism to induce SO(3)-equivariant representations for predicting the regression target $\mathbf{Q}$ in the inference phase from high-quality non-linear SO(3)-invariant features learned under the supervision of $tr(\mathbf{Q} \cdot \mathbf{Q}^\dagger)$ in the training phase. Taking SO(3)-invariant features as a bridge, this mechanism can incorporate non-linear expressive capabilities into SO(3)-equivariant representations while preserving their equivariant properties, as we prove, laying a solid foundation for accurately inferring complex SO(3)-equivariant targets.

We develop our theory into an SO(3)-equivariant non-linear representation learning method and apply it to the computation of the electronic-structure Hamiltonian. This task poses significant challenges for machine learning techniques due to the intrinsic SO(3) symmetry and high-dimensional complexity of the Hamiltonian, as highlighted by Yin et al. (2024). Our method significantly improves the state-of-the-art performance in Hamiltonian prediction on eight databases from the well-known DeepH and QH9 benchmark series (Li et al., 2022; Gong et al., 2023; Yu et al., 2023a). It demonstrates excellent generalization performance to both crystalline and molecular systems, covering challenging scenarios such as thermal motions, bilayer twists, scale variations, and new trajectories. Furthermore, as observed from the experiments on the QH9 benchmark series, our approach also substantially enhances the prediction accuracy of downstream physical quantities of

Hamiltonian including occupied orbital energy and electronic wavefunction. Moreover, our method also demonstrates superior performance in accelerating the convergence of classical DFT by providing predicted Hamiltonians as initialization matrices. Our leading performance comprehensively demonstrates that our method, while satisfying SO(3)-equivariance, possesses excellent expressive power and generalization performance, providing an effective deep learning tool for efficient and accurate electronic-structure calculations of atomic systems.

## 2 RELATED WORK

The SO(3)-equivariant representation learning paradigm (Thomas et al., 2018; Geiger & Smidt, 2022) typically developed group theory-based symmetry operators, such as linear scaling, element-wise sum, direct products, direct sums, Clebsch-Gordan decomposition, and equivariant normalization, to encode equivariant features. These operators have been used to construct graph neural network architectures for tasks in 3D point cloud analysis (Fuchs et al., 2020), molecular property prediction and dynamic simulation (Musaelian et al., 2023; Liao & Smidt, 2023), as well as Hamiltonian prediction (Schütt et al., 2019; Unke et al., 2021; Gong et al., 2023; Zhong et al., 2023). However, as non-linear activation functions may result in the loss of equivariance, they are restricted when applied to SO(3)-equivariant features with degree $l$ greater than one. This restriction severely limits the network's capability to model complex non-linear mappings. To alleviate this issue, methods like DeepH-E3 (Gong et al., 2023), QHNet (Yu et al., 2023b) and Equiformer (Liao & Smidt, 2023) introduced a gated activation mechanism that feeds SO(3)-invariant features ($l = 0$) into non-linear activation functions and uses these features as linear gating coefficients for SO(3)-equivariant features ($l \geq 1$), aiming to enhance their expressive power while maintaining strict equivariance. However, as demonstrated in our numerical study in Appendix H, multiplying SO(3)-equivariant features with linear coefficients may not be fully effective in enhancing non-linear expressiveness.

In order to improve non-linear expressiveness, Zitnick et al. (2022) decomposed SO(3)-equivariant features into SO(3)-invariant coefficients of spherical harmonic basis functions. These SO(3)-invariant coefficients were processed by non-linear neural networks to enhance expressiveness, with equivariance regained by recombining the updated coefficients with the basis functions. In subsequent developments (Passaro & Zitnick, 2023; Liao et al., 2024; Wang et al., 2024b;c), this approach has demonstrated a remarkable capacity to fit complex functions. However, as pointed out by existing literature (Zhang et al., 2023), this approach degenerates from continuous to discrete rotational equivariance, losing strict equivariance to continuous rotational transformations due to the decomposition based on inner-product operations with discrete basis functions; Li et al. (2022) proposed a local coordinate strategy, projecting rotating global coordinates onto SO(3)-invariant local ones for the non-linear neural network to encode. However, this strategy is effective only for global rigid rotations and fails to maintain symmetry under non-rigid perturbations like thermal fluctuations or bilayer twists, as it lacks a neural mechanism to enforce equivariance; Yin et al. (2024) proposed a hybrid framework consisting of both group theory-guaranteed SO(3)-equivariant mechanisms and non-linear mechanisms to regress Hamiltonians. In this framework, the non-linear mechanisms showed remarkable capability at learning SO(3)-equivariance from the data with the help of the theoretically SO(3)-equivariant mechanisms, and released powerful non-linear expressive capabilities to achieve more numerical accuracy. However, the equivariance achieved through data-driven methods does not have strict theoretical guarantee, even with rotational data augmentation. In some applications, the demands for symmetry are extremely high. Even minor deviations from perfect SO(3)-equivariance can result in incorrect physical results. In this paper, we propose a framework that theoretically combines strict SO(3)-equivariance with the non-linear expressiveness of neural networks to resolve the equivariance-expressiveness dilemma.

## 3 PROBLEM FORMALIZATION

For an introduction to the foundational concepts relevant to the problem addressed in this paper, please see Appendix A. Here, we directly focus on the equivariance of physical quantities under 3D rotational operations that form the SO(3) group. Let $\mathbf{Q}^{l_p \otimes l_q}$ denote an SO(3)-equivariant quantity in direct-product state formed by $l_p \otimes l_q$, i.e., the direct product between degrees $l_p$ and $l_q$. It obeys the following SO(3)-equivariant law:

$$\mathbf{Q}(\mathbf{R})^{l_p \otimes l_q} = \mathbf{D}^{l_p}(\mathbf{R}) \cdot \mathbf{Q}^{l_p \otimes l_q} \cdot (\mathbf{D}^{l_q}(\mathbf{R}))^{\dagger} \tag{1}$$

where † denotes the conjugate transpose operation. $\mathbf{R} \in \mathbb{R}^{3 \times 3}$ is the rotational matrix, $\mathbf{D}^{l_p}(\mathbf{R}) \in \mathbb{R}^{(2l_p+1) \times (2l_p+1)}$ and $\mathbf{D}^{l_q}(\mathbf{R}) \in \mathbb{R}^{(2l_q+1) \times (2l_q+1)}$ are the Wigner-D matrices of degrees $l_p$ and $l_q$, respectively; $\mathbf{Q}(\mathbf{R})^{l_p \otimes l_q} \in \mathbb{R}^{(2l_p+1) \times (2l_q+1)}$ denotes the transformed results of $\mathbf{Q}^{l_p \otimes l_q} \in \mathbb{R}^{(2l_p+1) \times (2l_q+1)}$ through the rotational operation by $\mathbf{R}$.

$\mathbf{Q}^{l_p \otimes l_q}$ in the direct-product state can be further decomposed into a series of direct-sum state components, i.e., $\mathbf{q}^l(|l_p - l_q| \leq l \leq l_p + l_q)$, which follows SO(3)-equivariant law mathematically equivalent to Eq. 1 but with a simpler form:

$$\mathbf{q}(\mathbf{R})^l = \mathbf{D}^l(\mathbf{R}) \cdot \mathbf{q}^l, |l_p - l_q| \leq l \leq l_p + l_q \tag{2}$$

where $\mathbf{q}^l \in \mathbb{R}^{2l+1}$ and $\mathbf{q}(\mathbf{R})^l \in \mathbb{R}^{2l+1}$ respectively denote the components with degree $l$ before and after the rotational operation by $\mathbf{R}$.

For ease of processing, the internal representations of SO(3)-equivariant neural networks (Gong et al., 2023; Liao & Smidt, 2023) are typically in the direct-sum form. To obey the equivariant law of Eq. 2 for the regression target, these hidden representations must also satisfy the same form of equivariance:

$$\mathbf{f}(\mathbf{R})^{(k)l} = \mathbf{D}^l(\mathbf{R}) \cdot \mathbf{f}^{(k)l} \tag{3}$$

where $\mathbf{f}^{(k)l} \in \mathbb{R}^{2l+1}$ and $\mathbf{f}(\mathbf{R})^{(k)l} \in \mathbb{R}^{2l+1}$ respectively denote one channel of hidden features with degree $l$ before and after the rotational operation by $\mathbf{R}$, at the $k$ th hidden layer.

Due to the intrinsic complexity and non-linearity of physic quantities, neural networks on regressing these quantities are supposed to equip with non-linear mappings to fully capture the intrinsic patterns of the physical quantities, which is crucial for precise and generalizable prediction performance. Meanwhile, the non-linear mappings, denoted as $g_{nonlin}(\cdot)$, must also preserve SO(3)-equivariance, which is expressed as:

$$\mathbf{f}(\mathbf{R})^{(k+1)l} = \mathbf{D}^l(\mathbf{R}) \cdot \mathbf{f}^{(k+1)l}, \quad \text{subject to } \mathbf{f}^{(k+1)l} = g_{nonlin}(\mathbf{f}^{(k)l}) \tag{4}$$

However, directly implementing $g_{nonlin}(\cdot)$ as neural network module with non-linear activation functions, such as $Sigmoid$, $Softmax$ and $SiLU$, may result in the destruction of strict equivariance. How to make $g_{nonlin}(\cdot)$ both theoretically SO(3)-equivariant and capable of non-linear expressiveness, and effectively apply it to the prediction of SO(3)-equivariant complex physical quantities, i.e., electronic-structure Hamiltonian in this context, is the core problem this paper aims at solving.

## 4 THEORY

**Theorem 1.** *The quantity* $\mathbf{T} = tr(\mathbf{Q} \cdot \mathbf{Q}^{\dagger})$ *is SO(3)-invariant,* where $\mathbf{Q}$ is the simplified representation (without superscripts) of $\mathbf{Q}^{l_p \otimes l_q}$ defined in Section 3, and † denotes the conjugate transpose operation, $tr(\cdot)$ is the trace operation.

**Theorem 2.** *The non-linear neural mapping* $g_{nonlin}(\cdot)$ *defined as the following is SO(3)-equivariant:*

$$\mathbf{v} = g_{nonlin}(\mathbf{f}) = \frac{\partial z}{\partial \mathbf{f}}, \quad \text{subject to} \quad z = s_{nonlin}(u), u = CGDecomp(\mathbf{f} \otimes \mathbf{f}, 0) \tag{5}$$

where $\mathbf{f}$ is an input SO(3)-equivariant feature with degree $l$ in the direct-sum state, $\otimes$ denotes the direct-product operation of tensors, CGDecomp$(\cdot, 0)$ refers to performing a Clebsch-Gordan decomposition of the tensor and returning the scalar component of degree 0, $s_{nonlin}(\cdot)$ represents arbitrary differentiable non-linear neural modules, and $\mathbf{v}$ is the outputted feature encoded by $g_{nonlin}(\cdot)$.

The proofs of the two theorems above are presented in Appendix C.

**Remark 1.** *It is straightforward to demonstrate that* $\mathbf{Q}$ *and* $\mathbf{T}$ *in Theorem 1 satisfy the following relationship:* $\mathbf{Q} = \frac{\partial \mathbf{T}}{\partial Conj(\mathbf{Q})}$, *where* $Conj(\cdot)$ *denotes the complex conjugate. This shows that* $\mathbf{Q}$ *can be derived via a differentiable mapping from the invariant space to the equivariant space, which inherently imposes a strong constraint on the relationships between the components of* $\mathbf{Q}$. *This mechanism is universal and extends beyond labels to representation learning in neural networks.*

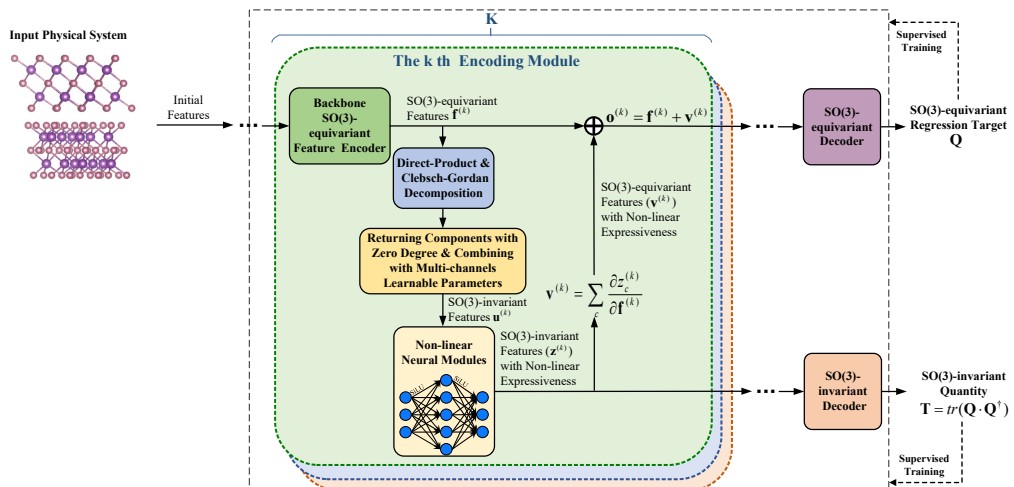

Figure 1: Methodological framework for learning SO(3)-equivariant representations with non-linear expressiveness to regress complex SO(3)-equivariant targets.

*In Theorem 2, we propose a construction of SO(3)-equivariant features: $\mathbf{v} = \frac{\partial z}{\partial \mathbf{f}}$, where $\mathbf{v}$ (used to regressing $\mathbf{Q}$, as detailed in Section 5) and $z$ (regressing $\mathbf{T}$) reflect the partial derivative relationship between $\mathbf{Q}$ and $\mathbf{T}$. Compared to the existing gated activation mechanism which can be expressed as $\mathbf{v} = z \cdot \mathbf{f}$, this approach imposes stronger non-linearity and physical constraints on the relationships between the components of the equivariant features and enables effective joint learning of $z$ and $\mathbf{v}$, supervised by $\mathbf{T}$ and $\mathbf{Q}$.*

*Notably, while $\mathbf{Q} = \frac{\partial \mathbf{T}}{\partial Conj(\mathbf{Q})}$ involves a derivative with respect to the complex conjugate of $\mathbf{Q}$, and $\mathbf{v} = \frac{\partial z}{\partial \mathbf{f}}$ involves a derivative with respect to $\mathbf{f}$ (since $\mathbf{v}$ is unknown), both are consistent at a higher abstraction level. They show that an SO(3)-equivariant quantity (or feature) can be derived from an associated SO(3)-invariant quantity (or feature) through a derivative relationship, whether for $\mathbf{T} = tr(\mathbf{Q} \cdot \mathbf{Q}^\dagger)$ or $z = s_{nonlin}(CGDecomp(\mathbf{f} \otimes \mathbf{f}, 0))$.*

*In summary, our gradient-based mechanism leverages stronger non-linearity via invariant functions and enforces strict equivariance with gradient-based physical constraints that regulate the relationships between components, making it more effective than the gated mechanism.*

## 5 METHOD

As shown in Fig. 1, taking $z$ in Eq. 5 serving as the bridge between Theorem 1 and Theorem 2, we propose a method for learning non-linear representations that satisfy the SO(3)-equivariance property outlined in Eq. 4. The core of our method can be abstractly referred to as **TraceGrad**. At the label level, it incorporates the SO(3)-invariant **trace** quantity $tr(\mathbf{Q} \cdot \mathbf{Q}^\dagger)$ introduced in Theorem 1 as a supervisory signal for learning $z$, i.e., the SO(3)-invariant internal representations of $g_{\text{nonlin}}(\cdot)$ in Theorem 2. Given the attribute that invariance is preserved under non-linear operations, $z$ is encoded by $s_{nonlin}(\cdot)$ to obtain non-linear expressive capabilities. At the representation level, by taking the **grad**ient of $z$ with respect to $\mathbf{f}$, the non-linear expressiveness of $z$ is transferred to the equivariant feature $\mathbf{v}$, while maintaining strict SO(3)-equivariance as we prove. Subsequently, merging $\mathbf{v}$ and $\mathbf{f}$, and applying $g_{\text{nonlin}}(\cdot)$ in a stacked manner can yield rich SO(3)-equivariant non-linear representations for inferring the SO(3)-equivariant regression target.

### 5.1 ENCODING AND DECODING FRAMEWORK

Our encoding framework corresponds to a total of $K$ encoding modules, which are sequentially connected to form a deep encoding framework. For the $k$-th module ($1 \leq k \leq K$), it first introduces an SO(3)-equivariant backbone encoder, e.g., the encoders from DeepH-E3 (Gong et al., 2023) or QHNet (Yu et al., 2023b) which is composed of recently developed equivariant operators (Thomas

et al., 2018; Geiger & Smidt, 2022) like linear scaling, direct products, direct sums, Clebsch-Gordan decomposition, gated activation, equivariant normalization, and etc., to encode the physical system's initial representations, such as spherical harmonics (Schrodinger, 1926), or representations passed from the previous neural layers, as equivariant feature $\mathbf{f}^{(k)}$ in the current hidden layer. Next, we construct the feature $\mathbf{v}^{(k)} = g_{\text{nonlin}}(\mathbf{f}^{(k)})$, to achieve sufficient non-linear expressiveness while maintaining SO(3)-equivariance, where $g_{\text{nonlin}}(\cdot)$, $s_{nonlin}(\cdot)$ are the non-linear functions defined in Eq. 5. The function $s_{nonlin}(\cdot)$ can be implemented as any differentiable non-linear neural network module, such as feed-forward layers with non-linear activation functions like $SiLU$ and normalization operations like $Layernorm$.

For an expressive representation of a complex physical system, $\mathbf{f}^{(k)}$ is usually not a feature of single degree but a direct-sum concatenation of series of components $\{\mathbf{f}^{(k)l_1}, \mathbf{f}^{(k)l_2}, \mathbf{f}^{(k)l_3}, ...\}$ at multiple degrees, i.e., $L^{(k)} = \{l_1, l_2, l_3, ...\}$, where some components share the same degree while others differ. In this case, it becomes necessary to extend the decomposition operator, i.e., CGDecomp$(\cdot)$ in Eq. 5, to accommodate various components of degrees. Moreover, in the context of neural networks, introducing learnable parameters $W$ and more feature channels $C$ may improve the model capacity. Based on these considerations, when constructing the encoding module, the CGDecomp$(\cdot)$ operation in Eq. 5 can be expanded as CGDecomp$_{ext}(\cdot)$, and its outputs can be expanded as $\mathbf{u}^{(k)}$, as shown in Eq. 6:

$$\mathbf{u}^{(k)} = [u_1^{(k)}, ..., u_c^{(k)}, ..., u_C^{(k)}],$$
$$u_c^{(k)} = \text{CGDecomp}_{\text{ext}}(\mathbf{f}^{(k)} \otimes \mathbf{f}^{(k)}, 0, W) = \sum_{l_i, l_j \in L^{(k)}, l_i = l_j} W_{ij}^c \cdot \text{CGDecomp}(\mathbf{f}^{(k)l_i} \otimes \mathbf{f}^{(k)l_j}, 0) \quad (6)$$

where $W_{ij}^c$ represents learnable parameters, $u_c^{(k)}$ is the $c$ th channel of $\mathbf{u}^{(k)}$. CGDecomp$_{\text{ext}}(\cdot)$ adds learnable parameters to CGDecomp$(\cdot)$ and expands its output from a single channel to multiple channels. To further enhance the model capacity, we also expand the output of $s_{\text{nonlin}}(\cdot)$ to multiple channels as follows: $\mathbf{z}^{(k)} = [z_1^{(k)}, \ldots, z_c^{(k)}, \ldots, z_C^{(k)}]$. We define $\mathbf{v}_c^{(k)} = \frac{\partial z_c^{(k)}}{\partial \mathbf{f}^{(k)}}$, and construct the new features $\mathbf{v}^{(k)}$ by $\mathbf{v}^{(k)} = \sum_c \mathbf{v}_c^{(k)}$. It is evident that these extensions maintain the SO(3)-invariance of $\mathbf{u}^{(k)}$ and $\mathbf{z}^{(k)}$, as well as the SO(3)-equivariance of $\mathbf{v}^{(k)}$. $\mathbf{v}^{(k)}$ is combined with $\mathbf{f}^{(k)}$ in a residual manner like $\mathbf{o}^{(k)} = \mathbf{f}^{(k)} + \mathbf{v}^{(k)}$ as the output of the $k$ th encoding module.

We follow previous literature (Gong et al., 2023; Yu et al., 2023b) to send the features from the last layer of the encoder, i.e., $\mathbf{o}^{(K)}$ in our framework, into the SO(3)-equivariant decoder to regress the predictions of $\mathbf{Q}$. For this, we can directly utilize the mature design of the SO(3)-equivariant decoders in DeepH-E3 or QHNet. The new part in the decoding phase introduced by our method is the SO(3)-invariant decoder, consisting of feed-forward layers taking $\mathbf{z} = [\mathbf{z}^{(1)}, \ldots, \mathbf{z}^{(k)}, \ldots, \mathbf{z}^{(K)}]$ as the input to predict $\mathbf{T}$. For the Hamiltonian prediction task, $\mathbf{Q}$ corresponds to each basic Hamiltonian block, namely $\mathbf{H}^{l_p \otimes l_q}$ ($1 \le p \le P$, $1 \le q \le Q$), while $\mathbf{T} = \text{tr}(\mathbf{H}^{l_p \otimes l_q} \cdot (\mathbf{H}^{l_p \otimes l_q})^\dagger)$.

## 5.2 TRAINING

The training loss function is shown as the following:

$$\min_\theta loss = loss_Q + \mu(loss_Q, loss_T) \cdot loss_T,$$
$$loss_Q = Error(\widehat{\mathbf{Q}}, \mathbf{Q}^*), \quad loss_T = Error(\widehat{\mathbf{T}}, \mathbf{T}^*) \quad (7)$$

where $\theta$ denotes all of the learnable parameters of our framework, $\widehat{\mathbf{Q}}$, $\widehat{\mathbf{T}}$ and $\mathbf{Q}^*$, $\mathbf{T}^*$ respectively denote the predictions and labels of $\mathbf{Q}$ and $\mathbf{T}$. In order to prevent the numerical disparity between the two loss terms and stabilize the training for both of the SO(3)-equivariant and SO(3)-invariant branches, we apply $\mu(loss_Q, loss_T)$, i.e., a coefficient to regularize the relative scale of the two loss terms:

$$\mu(loss_Q, loss_T) = \lambda \cdot No\_Grad(\frac{loss_Q}{loss_T}) \quad (8)$$

where $No\_Grad(\cdot)$ denotes gradient discarding when calculating such coefficient, as this coefficient is only used for adjusting the relative scale between the two loss terms and should not itself be a

Table 1: Experimental results measured by the $MAE_{all}^H$, $MAE_{cha\_s}^H$ and $MAE_{cha\_b}^H$ metrics (meV) on the Monolayer Graphene ($MG$), Monolayer MoS2 ($MM$), Bilayer Graphene ($BG$), Bilayer Bismuthene ($BB$), Bilayer Bi2Te3 ($BT$) and Bilayer Bi2Se3 ($BS$) databases, where the superscripts $nt$ and $t$ respectively denote the non-twisted and twisted subsets.

| Methods | $MG$ | | | $MM$ | | |
|---|---|---|---|---|---|---|
| | $MAE$ ($\downarrow$) | | | | | |
| | $MAE_{all}^H$ | $MAE_{cha\_s}^H$ | $MAE_{cha\_b}^H$ | $MAE_{all}^H$ | $MAE_{cha\_s}^H$ | $MAE_{cha\_b}^H$ |
| DeepH-E3 (Baseline) | 0.251 | 0.357 | 0.362 | 0.406 | 0.574 | 1.103 |
| DeepH-E3+TraceGrad | **0.175** | **0.257** | **0.228** | **0.285** | **0.412** | **0.808** |
| **Methods** | $BG^{nt}$ | | | $BG^t$ | | |
| DeepH-E3 (Baseline) | 0.389 | 0.453 | 0.644 | 0.264 | 0.429 | 0.609 |
| DeepH-E3+TraceGrad | **0.291** | **0.323** | **0.430** | **0.198** | **0.372** | **0.406** |
| **Methods** | $BB^{nt}$ | | | $BB^t$ | | |
| DeepH-E3 (Baseline) | 0.274 | 0.304 | 1.042 | 0.468 | 0.602 | 2.399 |
| DeepH-E3+TraceGrad | **0.226** | **0.256** | **0.740** | **0.384** | **0.503** | **1.284** |
| **Methods** | $BT^{nt}$ | | | $BT^t$ | | |
| DeepH-E3 (Baseline) | 0.447 | 0.480 | 1.387 | 0.831 | 0.850 | 4.572 |
| DeepH-E3+TraceGrad | **0.295** | **0.312** | **0.718** | **0.735** | **0.755** | **4.418** |
| **Methods** | $BS^{nt}$ | | | $BS^t$ | | |
| DeepH-E3 (Baseline) | 0.397 | 0.424 | 0.867 | 0.370 | 0.390 | 0.875 |
| DeepH-E3+TraceGrad | **0.300** | **0.332** | **0.644** | **0.291** | **0.302** | **0.674** |

source of training gradients, otherwise it would counteract the gradients from $loss_T$ in Eq. 7. All of the encoding and decoding modules are trained jointly by Eq. 7.

For the implementation details of our method, including the specific design of the network modules, parameter settings, and training specifics, please refer to Appendix E.

## 6 EXPERIMENTS

### 6.1 EXPERIMENTAL CONDITIONS

We apply our theory and method to the electronic-structure Hamiltonian prediction task, and collect results on eight benchmark databases, i.e., Monolayer Graphene ($MG$), Monolayer MoS2 ($MM$), Bilayer Graphene ($BG$), Bilayer Bismuthene ($BB$), Bilayer Bi2Te3 ($BT$), Bilayer Bi2Se3 ($BS$), QH9-stable ($QS$), and QH9-dynamic ($QD$). The first six databases, consisting of periodic crystalline systems with elements like C, Mo, S, Bi, Te and Se, are from the DeepH benchmark series (Li et al., 2022; Gong et al., 2023). The last two databases are from the QH9 benchmark series (Yu et al., 2023a), composed of molecular systems with elements like C, H, O, N and F. These databases present diverse and complex challenges to a regression model. Regarding $MG$, $MM$, and $QD$, as their samples are prepared from an temperature environment at three-hundred Kelvin, the thermal motions lead to complex non-rigid deformations, increasing the difficulty of Hamiltonian prediction. For $BG$, $BB$, $BT$, and $BS$, the twisted structures, with an interplay of SO(3)-equivariant effects and van der Waals (vdW) force variations bring significant generalization challenges, which are further exacerbated by the absence of any twisted samples in the training sets. Besides, $BB$, $BT$, and $BS$ exhibit strong spin-orbit coupling (SOC) effects, which further increase the complexity of Hamiltonian modeling. For the $QS$ database, the 'ood' strategy from the official settings is used to split the training, validation, and testing sets, ensuring that the atom number of samples do not overlap across the three subsets. For the $QD$ database, the 'mol' strategy provided by Yu et al. (2023a) is applied to split the training, validation, and testing sets, ensuring that there are no thermal motion samples from the same temporal trajectory across the three subsets. The 'mol' and 'ood'

Table 2: Experimental results measured by the $MAE_{all}^H$, $MAE_{diag}^H$, $MAE_{non\_diag}^H$, $MAE^\epsilon$, and $Sim(\psi)$ metrics on the QH9-stable (QS) and QH9-dynamic (QD) databases respectively using 'ood' and 'mol' split strategies (Yu et al., 2023a). ↓ means lower values correspond to better accuracy, while ↑ means higher values correspond to better performance. The units of MAE metrics are meV, while $Sim(\psi)$ is the cosine similarity which is dimensionless.

| Methods | QS | | | | $Sim(\psi)$ (↑) |
|---|---|---|---|---|---|
| | MAE (↓) | | | | |
| | $MAE_{all}^H$ | $MAE_{diag}^H$ | $MAE_{non\_diag}^H$ | $MAE^\epsilon$ | |
| QHNet (Baseline) | 1.962 | 3.040 | 1.902 | 17.528 | 0.937 |
| QHNet+TraceGrad | **1.191** | **2.125** | **1.139** | **8.579** | **0.948** |
| **Methods** | QD | | | | |
| QHNet (Baseline) | 4.733 | 11.347 | 4.182 | 264.483 | 0.792 |
| QHNet+TraceGrad | **2.819** | **6.844** | **2.497** | **63.375** | **0.927** |

strategies aim to assess the regression model's extrapolation capability with respect to the number of atoms as well as the temporal trajectories, respectively. Detailed statistic information of these databases can be found in the Appendix D.

Implementation details of our method for experiments on these databases are presented in Appendix E.

We use a comprehensive set of metrics to deeply evaluate the accuracy performance of deep learning electronic-structure Hamiltonian prediction models. On the databases from the DeepH benchmark series (Li et al., 2022; Gong et al., 2023), we follow Yin et al. (2024) to adopt a set of Mean Absolute Error (MAE) metrics between predicted and ground truth Hamiltonians, including $MAE_{all}^H$ for measuring average MAE of all samples and matrix elements, $MAE_{cha\_s}^H$ for measuring the MAE of challenging samples where the baseline model performs the worst, $MAE_{block}^H$ for measuring the MAE of different basic blocks in the Hamiltonian matrix, and $MAE_{cha\_b}^H$ for measuring the MAE on the most challenging Hamiltonian block where the baseline model shows the poorest performance (with the largest $MAE_{block}^H$). These metrics comprehensively reflect the accuracy performance, covering not only the average accuracy but also the accuracy on difficult samples and challenging blocks of the Hamiltonian matrices. On the two databases from the QH9 benchmark series, we adopt the metrics introduced by their original paper (Yu et al., 2023a), including MAE of Hamiltonian matrices, which are further subdivided into $MAE_{all}^H$ for measuring average MAE, $MAE_{diag}^H$ for measuring MAE of Hamiltonian matrix formed by an atom with itself, and $MAE_{non\_diag}^H$ for measuring MAE of Hamiltonian matrices formed by different atoms; as well as the MAE ($MAE^\epsilon$) of occupied orbital energies $\epsilon$ induced by the predicted Hamiltonians and compared to the ground truth ones, and the cosine similarity ($Sim(\psi)$) between the electronic wavefunctions $\psi$ induced by the predicted and ground truth Hamiltonians. $\epsilon$ and $\psi$ are crucial downstream physical quantities for determining multiple properties of the atomic systems as well as their dynamics, highly reflecting the application values of the Hamiltonian regression model.

## 6.2 Results and Analysis

We compare experimental results from two setups: the first one is the experimental results of the baseline SO(3)-equivariant regression model (Gong et al., 2023; Yu et al., 2023b) for Hamiltonian prediction, and the second one is the experimental results of extending the architecture and pipeline of the baseline model through the proposed TraceGrad method, which incorporates non-linear expressiveness into the SO(3)-equivariant features of the baseline model with the gradient operations of SO(3)-invariant non-linear features learned under the supervision of the trace targets. We choose DeepH-E3 (Gong et al., 2023) as the baseline model for databases from the DeepH benchmark series (Li et al., 2022; Gong et al., 2023); and we choose QHNet (Yu et al., 2023b) as the baseline model for databases from the QH9 benchmark series (Yu et al., 2023a). They are the respective state-of-the-art (SOTA) methods with strict SO(3)-equivariance on the corresponding databases.

We list the results of DeepH-E3 and DeepH-E3+TraceGrad in Tables 1 for databases from the DeepH benchmark series, reporting the values of $MAE_{all}^H$, $MAE_{cha\_s}^H$, and $MAE_{cha\_b}^H$. The results of DeepH-E3 to be compared are copied from Yin et al. (2024). The results of DeepH-E3+TraceGrad are the average from 10 independent repeated experiments. Regarding the metric of $MAE_{block}^H$ for every Hamiltonian block, due to its large data volume, we just present its values for all databases from the DeepH series in Appendix F. We use the same fixed random seed as adopted by DeepH-E3 for all random processes in experiments on these six databases. As a result, the standard deviation of the Hamiltonian prediction MAE across repeated experiments does not exceed 0.007 meV for each of the six databases and is negligible.

From results presented in Tables 1, we could find that the proposed TraceGrad method dramatically enhances the accuracy performance of the baseline method DeepH-E3, both on average and for challenging samples and blocks, both on the non-twisted samples and the twisted samples. Specifically, on the corresponding datasets, TraceGrad lowers down the $MAE_{all}^H$ and $MAE_{cha}^H$ of DeepH-E3 with relative ratios of up to 34% and 35%, respectively. Furthermore, from the results included in Appendix F, TraceGrad significantly improves the performance for the vast majority of basic blocks. Particularly, for the blocks where DeepH-E3 perform the worst, TraceGrad reduces the $MAE$ ($MAE_{cha\_b}^H$) by a maximum of 48%. The leading performance on the $MG$ and $MM$ databases prepared at three-hundred Kelvin temperature demonstrates the robustness of our method against thermal motion. The high accuracy on the $BB$, $BT$, and $BS$ databases, which have strong SOC effects, indicates our method's strong capability to model such effects. The excellent performance on the $BG^t$, $BB^t$, $BT^t$, and $BS^t$ subsets showcases the method's superior generalization to twisted structures, which are not present in the training data. The outstanding performance on such samples highlights the good potential for studying twist-related phenomena, a hot research topic that may bring new electrical and transport properties (Cao et al., 2018; Wang et al., 2024a; He et al., 2024). Additionally, the $BG^t$, $BB^t$, $BT^t$, and $BS^t$ subsets contain significantly larger unit cells compared to the training set (see Appendix D for statistics of their sizes), yet our method still excels on these subsets as measured by the multiple MAE metrics, demonstrating its good scalability on the sizes of atomic systems it handles.

In Table 2, we present the results of QHNet and QHNet+TraceGrad under the metrics of $MAE_{all}^H$, $MAE_{diag}^H$, $MAE_{non\_diag}^H$, $MAE^\epsilon$, and $Sim(\psi)$ for the $QS$ and $QD$ databases. The results of QHNet to be compared are taken from their original paper (Yu et al., 2023a), and for the unification of MAE units, we convert the units of MAE from $10^{-6}$ Hartree ($E_h$) in the original paper to meV[1]. The results of QHNet+TraceGrad are the average from 10 independent repeated experiments. To ensure reproducibility, we use the same fixed random seeds as employed in QHNet for all random processes in the experiments on the $QS$ and $QD$ databases. As a result, the standard deviation of the Hamiltonian prediction MAE across repeated experiments is no greater than 0.009 meV for both $QS$ and $QD$ and is also negligible.

The results presented in Table 2 demonstrate that the proposed TraceGrad method significantly enhances the accuracy of the baseline QHNet model across all metrics on the $QS$ and $QD$ databases. Specifically, TraceGrad reduces $MAE_{all}^H$, $MAE_{non\_diag}^H$, $MAE_{diag}^H$, $MAE^\epsilon$, and $Sim(\psi)$ of QH-Net with relative reductions of up to 40%, 39%, 40%, 76%, and 17%, respectively, on the corresponding databases. The significant accuracy improvements on the $QS$ database, partitioned using the 'ood' split strategy (Yu et al., 2023a) without scale overlapping among the training, validation, and testing sets, once again demonstrate the method's strong generalization capabilities across different scales of atomic systems. Meanwhile, performance on the $QD$ database under the 'mol' strategy (Yu et al., 2023a), which partitions the training, validation, and testing sets with samples from completely different thermal motion trajectories, highlight our method's robustness in generalizing to new thermal motion sequences. Furthermore, the substantial improvement in the prediction accuracy of $\epsilon$, i.e., occupied orbital energies crucial for determining electronic properties such as optical characteristics and conductivity in atomic systems, and $\psi$, i.e., the electronic wavefunctions essential for understanding electron distribution and interactions, underscores the potential values of our method for applications like material design, molecular pharmacology, and quantum computing.

---

[1] $1E_h = 27211.4$ meV

## 7 SUMMARY OF APPENDICES

Due to the page limit of the main manuscript, we have to provide some important contents in the Appendices, summarized as follows:

- Appendix A provides the definitions of foundational concepts relevant to the problem addressed in this paper.
- Appendix B describes the application tasks in this paper, i.e., the electronic-structure Hamiltonian prediction task.
- Appendix C provides the proofs for all of the proposed theorems.
- Appendix D presents the detailed information of the experimental databases.
- Appendix E presents the implementation details of the experiments.
- Appendix F compares the block-level MAE statistics ($MAE_{block}^H$ and $MAE_{cha\_b}^H$) for DeepH-E3 and DeepH-E3+TraceGrad.
- Appendix G reports the results of ablation study. Experimental results indicate that each individual mechanism of our method can contribute individually to the performance. Moreover, their combination provides even better performance.
- Appendix H reports the quantitative comparison between the proposed gradient-based mechanism (Grad) with the existing gated activation mechanism (Gate), demonstrating better accuracy performance of the proposed gradient-based mechanism compared to the gated mechanism, indicating that our method may be a better choice in terms of expressing complex non-linear mappings.
- Appendix I provides a theoretical analysis of the computational complexity advantage of our method over traditional DFT calculations.
- Appendix J makes a joint discussion on GPU time costs and performance gains brought by our TraceGrad method. It underscores the superiority of the TraceGrad method in improving accuracy performance while maintaining time efficiency.
- Appendix K quantifies the acceleration performance of the deep learning methods for DFT computations. Experimental results show that while combining TraceGrad introduces only a slight increase from the inference time of QHNet, it delivers significant improvements in accelerating the convergence of DFT methods.
- Appendix L corresponds to the synergy of our method with an approximately SO(3)-equivariant methodology (Yin et al., 2024).
- Appendix M discusses future work.

## 8 CONCLUSION

We propose a theoretical and methodological framework to tackle the issue of reconciling non-linear expressiveness with SO(3)-equivariance in deep learning frameworks for physical system modeling, through deeply investigating the mathematical connections between SO(3)-invariant and SO(3)-equivariant quantities, as well as their representations. We first constructs SO(3)-invariant quantities from SO(3)-equivariant regression targets, using them to train informative SO(3)-invariant non-linear representations. From these, SO(3)-equivariant features are derived with gradient operations, achieving non-linear expressiveness while maintaining strict SO(3)-equivariance. We apply our theory and method to the challenging electronic-structure Hamiltonian prediction tasks, achieving dramatic promotions in prediction accuracy across eight benchmark databases. Experimental results demonstrate that this approach not only improves the accuracy of Hamiltonian prediction but also significantly enhances the prediction for downstream physical quantities, and also markedly improves the acceleration ratios for traditional DFT algorithms.

### ETHICS STATEMENT

This work develops a representation learning method that exhibits strong non-linear expressive capabilities while strictly adhering to SO(3) equivariance. This method has demonstrated superior

accuracy in predicting electronic-structure Hamiltonians and related physical quantities, showcasing its potential to accelerate research in materials science and molecular pharmacology. While we recognize that our research area has not yet revealed direct negative social or ethical implications, several issues warrant our vigilance. Currently, although our method yields accurate predictions, the decision-making processes of deep learning systems often lack transparency, hindering a comprehensive understanding of the learning outcomes and limiting our ability to gain deeper insights. We believe it is important to investigate the interpretability of such models, particularly in terms of how they apply physical knowledge in a comprehensible way. Additionally, it is crucial to continually improve the correctness and fairness of deep learning models on this area. Ensuring high-quality and diverse training data, implementing sound model designs, and performing ongoing validation and refinement are necessary to guarantee model accuracy and the broad applicability of their results.

## REPRODUCIBILITY STATEMENT

We provide the proofs for all of the proposed Theorems in Appendix C. We provide the implementation details in Appendix E. The codes and download links of experimental data for this work, is available in the supplementary materials.

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

# APPENDICES

## A  DEFINITION OF CONCEPTS

This section provides definitions of foundational concepts relevant to the problem addressed in this paper. For additional background, please refer to the book by Dresselhaus et al. (2007).

**Definition 1.** *Group. A set $G$, denoted as $G = \{\ldots, g, \ldots\}$, equipped with a binary operation $\cdot$, is called a* group *if it satisfies the following four axioms:*

1. *Closure: For any $f, g \in G$, the result of their operation $f \cdot g$ is also an element of $G$: $f \cdot g \in G$.*

2. *Associativity: For all $f, g, h \in G$, the operation satisfies $(f \cdot g) \cdot h = f \cdot (g \cdot h)$.*

3. *Identity Element: There exists a unique element $e \in G$ (called the* identity*) such that for all $f \in G$, $e \cdot f = f \cdot e = f$.*

4. *Inverse Element: For each $f \in G$, there exists a unique element $f^{-1} \in G$ (called the* inverse*) such that $f \cdot f^{-1} = f^{-1} \cdot f = e$.*

**Definition 2.** *SO(3) Group. The special orthogonal group $SO(3)$ is the group of all $3 \times 3$ orthogonal matrices with determinant 1. Formally, it is defined as:*

$$SO(3) = \{\mathbf{R} \in \mathbb{R}^{3 \times 3} \mid \mathbf{R}^T \mathbf{R} = I, \det(\mathbf{R}) = 1\},$$

*where $\mathbf{R}^T$ denotes the transpose of $\mathbf{R}$, and $I$ is the $3 \times 3$ identity matrix. The elements of $SO(3)$ represent rotations in three-dimensional Euclidean space.*

**Definition 3.** *Group Representation. A representation of a group $G$ on a tensor space $T(V)$ is a homomorphism $\rho$ from $G$ to the general linear group $GL(T(V))$, the group of all invertible linear transformations on $T(V)$. Here, $T(V)$ represents the tensor space associated with the vector space $V$, encompassing all tensors that can be formed from elements of $V$. Formally, the homomorphism $\rho$ is defined as:*

$$\rho : G \to GL(T(V))$$

*such that for all $g_1, g_2 \in G$,*

$$\rho(g_1 g_2) = \rho(g_1)\rho(g_2),$$

*and $\rho(e) = I$, where $e$ is the identity element of $G$, and $I$ is the identity transformation on $T(V)$.*

**Definition 4.** *Irreducible Representation.* *A representation $\rho : G \to GL(V)$ of a group $G$ on a vector space $V$ is said to be* irreducible *if there is no proper subspace $W \subset V$ such that $\rho(g)W \subset W$ for all $g \in G$. In other words, the only invariant subspaces under the group action are the trivial subspaces $\{0\}$ and $V$ itself. If such a nontrivial invariant subspace exists, the representation is said to be* reducible.

**Definition 5.** *SO(3) Group Representation.* *A representation of the special orthogonal group $SO(3)$ on a vector space $V$ is a homomorphism*

$$\rho : SO(3) \to GL(V),$$

*where $GL(V)$ denotes the group of all invertible linear transformations on $V$. The representations of $SO(3)$ are typically classified into irreducible representations, each labeled by a degree $l$, which often corresponds to the quantum number for total angular momentum in quantum physics.*

**Definition 6.** *Wigner-D Matrices.* *One of the irreducible representations of $SO(3)$ is given by the Wigner-D matrices $\mathbf{D}^l(\mathbf{R})$, where $l$ is the degree of the representation, typically corresponding to the quantum number for total angular momentum in quantum physics. These matrices represent the transformation properties of angular momentum states under the action of the rotation group.*

*The Wigner-D matrices are the matrix elements of the rotation operator $\mathbf{R} \in SO(3)$ in the space of angular momentum eigenstates $|l, m\rangle$. Specifically, they are defined as:*

$$\rho(\mathbf{R})_{m',m} = \langle l, m'|\mathbf{R}|l, m\rangle = D^l_{m'm}(\mathbf{R}),$$

*where $|l, m\rangle$ are the eigenstates of the total angular momentum operator $\hat{L}^2$ and the $z$-component $\hat{L}_z$. The matrices $\mathbf{D}^l(\mathbf{R})$ describe how angular momentum states transform under rotations in $SO(3)$, and the index $l$ characterizes the irreducible representation corresponding to a specific angular momentum state.*

**Definition 7.** *Equivariance with Respect to a Group.* *Let $G$ be a group, and let $\rho_{T(V)} : G \to GL(T(V))$ and $\rho_{T(W)} : G \to GL(T(W))$ be representations of $G$ on tensor spaces $T(V)$ and $T(W)$, respectively. A map $f : T(V) \to T(W)$ is said to be equivariant with respect to the group $G$ if the following condition holds:*

$$f(\rho_{T(V)}(g) \circ v) = \rho_{T(W)}(g) \circ f(v) \quad \text{for all } v \in T(V) \text{ and } g \in G.$$

*where $\circ$ generally denotes the operation defined on the tensor space.*

**Definition 8.** *Invariance with Respect to a Group.* *Let $G$ be a group, and let $\rho_{T(V)} : G \to GL(T(V))$ be a representation of $G$ on a tensor space $T(V)$. A function $f : T(V) \to T(W)$ is said to be invariant under the group $G$ if the following condition holds:*

$$f(\rho_{T(V)}(g) \circ v) = f(v) \quad \text{for all } v \in T(V) \text{ and } g \in G.$$

*This definition indicates that the function $f$ remains unchanged under the action of the group $G$.*

**Definition 9.** *Direct-Product State.* *Let $V_1$ and $V_2$ be two vector spaces, and let $|v_1\rangle \in V_1$ and $|v_2\rangle \in V_2$ be arbitrary elements of these spaces. The direct-product state of $|v_1\rangle$ and $|v_2\rangle$ is defined as the element $|v_1\rangle \otimes |v_2\rangle$ in the direct-product space $V_1 \otimes V_2$. The direct product state represents all possible combinations of the elements of $V_1$ and $V_2$, and forms a new vector space that captures the joint state of two systems. The action of the group on this state is defined by the direct product of the individual actions on $|v_1\rangle$ and $|v_2\rangle$.*

**Definition 10.** *Direct-Product Physical Quantity Formed by Two Degrees.* *In quantum mechanics, direct-product spaces are used to describe the combined states of systems with distinct degrees of freedom, such as angular momentum. The combined state captures both the independent action of each degree and their joint transformation under rotations governed by the $SO(3)$ group.*

*Let $l_p$ and $l_q$ represent the angular momentum quantum numbers of two physical quantities. A direct-product state $\mathbf{Q}^{l_p \otimes l_q} \in \mathbb{R}^{(2l_p+1) \times (2l_q+1)}$ combines these degrees and transforms under the $SO(3)$ group according to:*

$$\mathbf{Q}(\mathbf{R})^{l_p \otimes l_q} = \mathbf{D}^{l_p}(\mathbf{R}) \cdot \mathbf{Q}^{l_p \otimes l_q} \cdot (\mathbf{D}^{l_q}(\mathbf{R}))^{\dagger},$$

*where $\mathbf{D}^{l_p}(\mathbf{R}) \in \mathbb{R}^{(2l_p+1) \times (2l_p+1)}$ and $\mathbf{D}^{l_q}(\mathbf{R}) \in \mathbb{R}^{(2l_q+1) \times (2l_q+1)}$ are the Wigner-D matrices for the degrees $l_p$ and $l_q$, respectively, and $\dagger$ denotes the conjugate transpose, ensuring unitary transformations.*

**Definition 11. *Direct-Sum State.*** *Let $V_1$ and $V_2$ be two vector spaces, and let $|v_1\rangle \in V_1$ and $|v_2\rangle \in V_2$ be arbitrary elements of these spaces. The* direct-sum state *of $|v_1\rangle$ and $|v_2\rangle$ is defined as the element $|v_1\rangle \oplus |v_2\rangle$ in the direct-sum space $V_1 \oplus V_2$.*

*The direct-sum space $V_1 \oplus V_2$ consists of ordered pairs of elements, where each element is drawn from one of the original spaces. The operations of vector addition and scalar multiplication in $V_1 \oplus V_2$ are defined component-wise:*

$$(a_1, a_2) + (b_1, b_2) = (a_1 + b_1, a_2 + b_2), \quad c \cdot (a_1, a_2) = (c \cdot a_1, c \cdot a_2),$$

*for $(a_1, a_2), (b_1, b_2) \in V_1 \oplus V_2$ and $c \in \mathbb{F}$, where $\mathbb{F}$ is the field over which $V_1$ and $V_2$ are defined.*

*The direct-sum state $|v_1\rangle \oplus |v_2\rangle$ represents a combination where the components remain in their respective vector spaces and do not interact with each other. The action of a group $G$ on the direct-sum state is defined by its independent action on each component:*

$$g \cdot (|v_1\rangle \oplus |v_2\rangle) = (g \cdot |v_1\rangle) \oplus (g \cdot |v_2\rangle), \quad \forall g \in G.$$

**Definition 12. *Clebsch-Gordan Decomposition for $SO(3)$ Group.*** *For the $SO(3)$ group, the Clebsch-Gordan decomposition explains how the direct product of two irreducible representations $V_{l_1}$ and $V_{l_2}$ can be expressed as a sum of irreducible representations.*

*If $|v_1\rangle \in V_{l_1}$ and $|v_2\rangle \in V_{l_2}$, their direct-product state $|v_1\rangle \otimes |v_2\rangle$ can be written as a linear combination of states $|v_l\rangle$ belonging to irreducible representations $V_l$, where $l$ ranges from $|l_1 - l_2|$ to $l_1 + l_2$:*

$$|v_l\rangle = \sum_{m_1, m_2} C^m_{l, m_1, m_2} |v_1\rangle \otimes |v_2\rangle,$$

*where $C^m_{l, m_1, m_2}$ are the Clebsch-Gordan coefficients.*

*For example, the direct-product state $\mathbf{Q}^{l_p \otimes l_q}$ can be decomposed into a direct sum of irreducible representations $\bigoplus \mathbf{q}^l$, where:*

$$\mathbf{q} = \bigoplus_{l=|l_p - l_q|}^{l_p + l_q} \mathbf{q}^l, \quad \text{and} \quad q^l_m = \sum_{m_p, m_q} C^m_{l, m_p, m_q} Q^{l_p \otimes l_q}_{m_p, m_q}.$$

*Here, $Q^{l_p \otimes l_q}_{m_p, m_q}$ represents the product states, $\mathbf{q}$ represents the direct sum of irreducible states $\mathbf{q}^l$, and $q^l_m$ represents the components of the irreducible state $\mathbf{q}^l$.*

## B    APPLICATION TASK DESCRIPTION: ELECTRONIC-STRUCTURE HAMILTONIAN CALCULATION

Density Functional Theory (DFT) (Hohenberg & Kohn, 1964; Kohn & Sham, 1965) has become a cornerstone of modern electronic structure theory, playing a pivotal role in condensed matter physics, quantum chemistry, and materials science. Introduced in the 1960s through the foundational work of Hohenberg, Kohn, and Sham, DFT provides a framework for studying many-electron systems by replacing the computationally expensive many-body wavefunction with the electron density $\rho(\mathbf{r})$ as the fundamental variable. This reformulation significantly reduces computational complexity while preserving essential quantum mechanical effects, enabling researchers to investigate systems of practical interest with manageable computational resources. Over the decades, DFT has proven

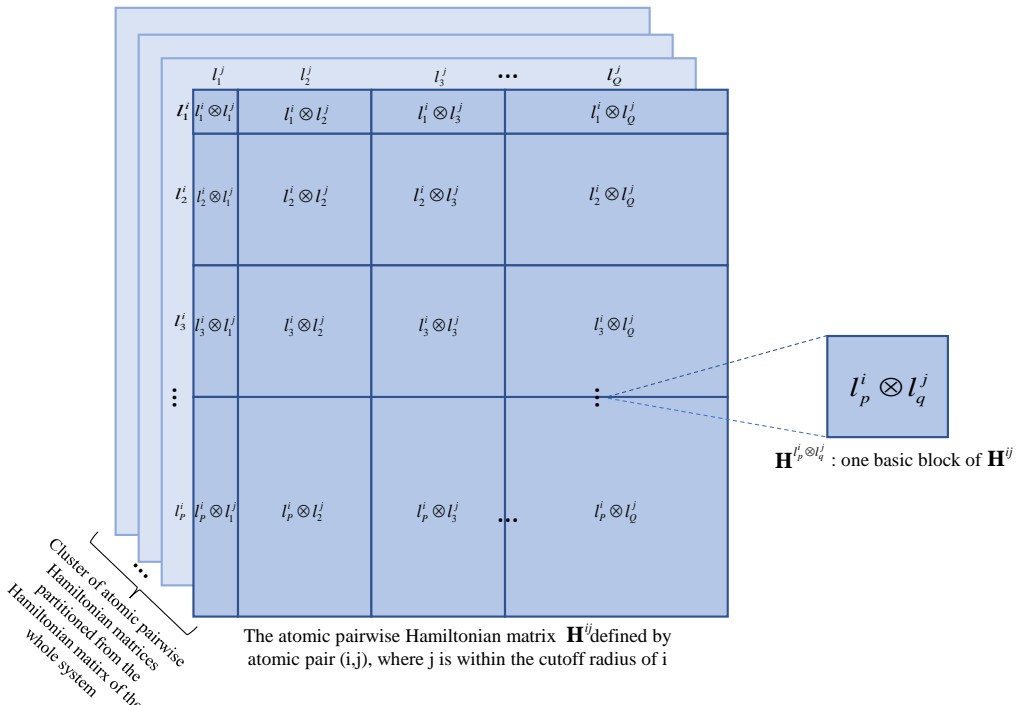

Figure 2: Illustration of atomic pairwise Hamiltonian matrices partitioned from the complete Hamiltonian matrix of the whole system. Each Hamiltonian matrix contains multiple basic blocks, with the $\mathbf{Q} = \mathbf{Q}^{l_p \otimes l_q}$ defined in Section 3 corresponding to a basic block $\mathbf{H}^{l_p^i \otimes l_q^j}$ here.

to be instrumental in calculating electronic band structures, optimizing structural geometries, and exploring a wide range of material properties, underscoring its versatility and importance across diverse scientific disciplines.

At the heart of DFT lies the Kohn-Sham equation (Kohn & Sham, 1965), which simplifies the many-body problem into a set of single-particle equations. These equations are expressed as:

$$\hat{H}\psi_i(\mathbf{r}) = \epsilon_i \psi_i(\mathbf{r}), \quad \text{subject to} \quad \hat{H} = -\frac{\hbar^2}{2m}\nabla^2 + V_{\text{ext}}(\mathbf{r}) + V_{\text{HXC}}[\rho](\mathbf{r}) \tag{9}$$

where $\hat{H}$ is the Hamiltonian operator, the Hartree-exchange-correlation potential $V_{\text{HXC}}[\rho](\mathbf{r}) = V_{\text{H}}[\rho](\mathbf{r}) + V_{\text{XC}}[\rho](\mathbf{r})$ is a functional of the electron density $\rho(\mathbf{r})$. The electron density is obtained from the Kohn-Sham orbitals $\psi_i(\mathbf{r})$ as:

$$\rho(\mathbf{r}) = \sum_{i=1}^{M} |\psi_i(\mathbf{r})|^2,$$

where $M$ represents the total number of electrons.

Atomic orbitals offer a computationally efficient basis for electronic structure calculations because they require fewer basis functions compared to other types of basis sets (Lin et al., 2023). Additionally, their inherently localized nature makes them particularly advantageous for large systems. When expressed in an atomic orbital basis set, the Kohn-Sham equations can be formulated into as a generalized eigenvalue problem:

$$\mathbf{HC} = \epsilon \mathbf{SC}, \tag{10}$$

where $\mathbf{H}$ is the Hamiltonian matrix incorporating the contributions from $V_{\text{ext}}$ and $V_{\text{HXC}}$, $\mathbf{S}$ is the overlap matrix, $\mathbf{C}$ contains the orbital coefficients, and $\epsilon$ represents the eigenvalues of the system.

The atomic orbitals used as the basis functions typically take the form:

$$\phi_\mu(\mathbf{r}) = R_\mu(r) Y_{l_\mu m_\mu}(\theta, \phi),$$

where $R_\mu(r)$ is the radial part of the wavefunction, and $Y_{l_\mu m_\mu}(\theta, \phi)$ are the spherical harmonics describing the angular dependence. These orbitals are localized around their respective atomic centers, decay rapidly as $r$ increases, and are truncated to zero beyond a cutoff radius $r_c$.

The matrix elements of the Hamiltonian and overlap matrices, $H_{\mu\nu}$ and $S_{\mu\nu}$, are defined as:

$$H_{\mu\nu} = \int \phi_\mu^*(\mathbf{r}) \hat{H} \phi_\nu(\mathbf{r}) \, d\mathbf{r}, \quad S_{\mu\nu} = \int \phi_\mu^*(\mathbf{r}) \phi_\nu(\mathbf{r}) \, d\mathbf{r}.$$

The localized nature of atomic orbitals ensures that matrix elements are non-zero only when the orbitals $\phi_\mu$ and $\phi_\nu$ overlap according to their cutoff radius. Beyond this range, the corresponding matrix elements are treated as zero, leading to sparsity in the Hamiltonian and overlap matrices. This sparsity significantly reduces computational effort and memory usage, especially for large systems.

Solving $H_{\mu\nu}$ involves an iterative self-consistent process: starting with an initial guess for $\rho^{(1)}(\mathbf{r})$, the potentials $V_{\text{HXC}}[\rho](\mathbf{r})$ are updated, and the equations are solved repeatedly until $\rho(\mathbf{r})$ converges. This process can be illustrated as $\rho^{(1)}(\mathbf{r}) \to V_{\text{HXC}}^{(1)}[\rho](\mathbf{r}) \to H_{\mu\nu}^{(1)} \to \psi_\mu^{(1)}(\mathbf{r}) \to \rho^{(2)}(\mathbf{r}) \to \ldots \to \rho^{(T)}(\mathbf{r})$, until the charge density $\rho(\mathbf{r})$ converges. At that point, the Hamiltonian matrix is outputted by $\rho^{(T)}(\mathbf{r}) \to H_{\mu\nu}^{(T)}$, from which downstream physical properties such as orbital energies and band structures are induced, determining the electronic, magnetic, and transport characteristics of the electron system.

Despite the remarkable success of Kohn-Sham DFT in advancing fields such as materials science, energy, and biomedicine over recent decades (Nagy, 1998; Jones, 2015), the challenge of high computational complexity remains unresolved. The main computational bottleneck lies in iterative solving the Kohn-Sham equation, especially in the matrix diagonalization of Eq. 10, which has a complexity of $\mathcal{O}(N^3)$, where $N$ is the number of atoms in the system. To address this challenge, recent approaches (Li et al., 2022; Yu et al., 2023b; Gong et al., 2023) have applied the deep graph learning paradigm to predict the *self-consistent* Hamiltonian. These methods use the Hamiltonian matrix $\mathbf{H}^{(T)}$, calculated from traditional DFT methods, as labels. This matrix can be partitioned into a series of atomic pairwise Hamiltonian matrices $\{\mathbf{H}^{ij} \mid j \in \Omega(i)\}$, as shown in Fig. 2, where $i$ and $j$ represent two atoms in the system. These methods train an efficient graph neural network to predict each $\mathbf{H}^{ij}$ from the 3D structure of the atomic system, thereby circumventing the extremely time-consuming self-consistent iterations. During the inference phase, these methods successfully reduced the computational complexity of calculating the Hamiltonian matrices to $\mathcal{O}(N)$, while showing good potential in generalizing to larger atomic systems, even though the training set, constrained by the expensive DFT labels, only includes smaller systems. Once the Hamiltonian matrices are obtained, many downstream physical properties can be efficiently calculated with $\mathcal{O}(N)$ complexity. A detailed analysis of the computational complexity is provided in Appendix I. This paper aims to tackle the challenge of predicting electronic Hamiltonians with high accuracy and reliability, while rigorously maintaining SO(3)-equivariance.

## C  PROOFS OF THEOREMS

*Proof of Theorem 1.* Under an SO(3) rotation represented by the rotational matrix $\mathbf{R}$, $\mathbf{Q} = \mathbf{Q}^{l_p \otimes l_q}$ is transformed as $\mathbf{Q}(\mathbf{R})$:

$$\mathbf{Q}(\mathbf{R}) = \mathbf{D}^{l_p}(\mathbf{R}) \cdot \mathbf{Q} \cdot \mathbf{D}^{l_q}(\mathbf{R})^\dagger,$$

where $\mathbf{D}^{l_p}(\mathbf{R})$ and $\mathbf{D}^{l_q}(\mathbf{R})$ are the Wigner-D matrices for the degrees of $l_p$ and $l_q$, respectively, corresponding to the rotation $\mathbf{R}$.

The conjugate transpose of the transformed quantity is:

$$\mathbf{Q}(\mathbf{R})^\dagger = \mathbf{D}^{l_q}(\mathbf{R}) \cdot \mathbf{Q}^\dagger \cdot \mathbf{D}^{l_p}(\mathbf{R})^\dagger.$$

Using the cyclic property of the trace, which states that the trace of a product of matrices remains unchanged under cyclic permutations (i.e., $tr(ABC) = tr(BCA) = tr(CAB)$), and combining the properties that $\mathbf{D}^{l_p}(\mathbf{R}) \cdot \mathbf{D}^{l_p}(\mathbf{R})^\dagger = \mathbf{I}$ and $\mathbf{D}^{l_q}(\mathbf{R}) \cdot \mathbf{D}^{l_q}(\mathbf{R})^\dagger = \mathbf{I}$, we can rearrange the terms inside the trace as follows:

$$\mathbf{T}(\mathbf{R}) = tr(\mathbf{Q}(\mathbf{R}) \cdot \mathbf{Q}(\mathbf{R})^\dagger) = tr((\mathbf{D}^{l_p}(\mathbf{R}) \cdot \mathbf{Q} \cdot \mathbf{D}^{l_q}(\mathbf{R})^\dagger) \cdot (\mathbf{D}^{l_q}(\mathbf{R}) \cdot \mathbf{Q}^\dagger \cdot \mathbf{D}^{l_p}(\mathbf{R})^\dagger))$$

$$= tr(\mathbf{D}^{l_p}(\mathbf{R}) \cdot \mathbf{Q} \cdot \mathbf{Q}^\dagger \cdot \mathbf{D}^{l_p}(\mathbf{R})^\dagger) = tr(\mathbf{Q} \cdot \mathbf{Q}^\dagger \cdot \mathbf{D}^{l_p}(\mathbf{R})^\dagger \cdot \mathbf{D}^{l_p}(\mathbf{R})) = tr(\mathbf{Q} \cdot \mathbf{Q}^\dagger) = \mathbf{T}.$$

Therefore, $\mathbf{T} = tr(\mathbf{Q} \cdot \mathbf{Q}^\dagger)$ is invariant under SO(3) transformations, its SO(3)-invariance is proved. □

*Proof of Theorem 2.* Under the given condition, the input feature $\mathbf{f}$ in direct-sum state is SO(3)-equivariant, meaning that under an SO(3) rotation represented by $\mathbf{R}$, it transforms as follows:

$$\mathbf{f}(\mathbf{R}) = \mathbf{D}^l(\mathbf{R}) \cdot \mathbf{f}$$

where $\mathbf{D}^l(\mathbf{R})$ is the Wigner-D matrix corresponding to degree $l$.

First, according to group theory, $u = \text{CGDecomp}(\mathbf{f} \otimes \mathbf{f}, 0)$ is an SO(3)-invariant scalar as the degree-zero component from the Clebsch-Gordan decomposition is invariant under rotations. Since applying a non-linear operation to an SO(3)-invariant quantity does not change its invariance, $z = s_{\text{nonlin}}(u)$ is also SO(3)-invariant, independent to the specific form of $s_{\text{nonlin}}(\cdot)$. It formally holds that:

$$z(\mathbf{R}) = z \tag{11}$$

Next, we apply the chain rule in Jacobian form. Considering $\mathbf{f}(\mathbf{R})$ is in the form of a column vector, to facilitate the application of the chain rule in vector form, we first transpose it into a row vector $\mathbf{f}(\mathbf{R})^T$, then differentiate:

$$\frac{\partial z(\mathbf{R})}{\partial \mathbf{f}^T(\mathbf{R})} = \frac{\partial z}{\partial \mathbf{f}^T(\mathbf{R})} = \frac{\partial z}{\partial \mathbf{f}^T} \frac{\partial \mathbf{f}^T}{\partial \mathbf{f}(\mathbf{R})^T} = \frac{\partial z}{\partial \mathbf{f}^T} \cdot \mathbf{D}^l(\mathbf{R})^{-1} = \frac{\partial z}{\partial \mathbf{f}^T} \cdot \mathbf{D}^l(\mathbf{R})^T \tag{12}$$

Here we utilize the property that $\mathbf{D}^l(\mathbf{R})^{-1} = \mathbf{D}^l(\mathbf{R})^T$ [2]. Since the representations of neural networks are generally real numbers, the corresponding Wigner-D matrix is also real unitary.

Finally, we transpose the result back to a column vector:

$$\mathbf{v}(\mathbf{R}) = g_{nonlin}(\mathbf{f}(\mathbf{R})) = (\frac{\partial z(\mathbf{R})}{\partial \mathbf{f}^T(\mathbf{R})})^T = (\frac{\partial z}{\partial \mathbf{f}^T} \cdot \mathbf{D}^l(\mathbf{R})^T)^T = \mathbf{D}^l(\mathbf{R}) \cdot \frac{\partial z}{\partial \mathbf{f}} = \mathbf{D}^l(\mathbf{R}) \cdot \mathbf{v} \tag{13}$$

This proves that $g_{\text{nonlin}}(\cdot)$ is an SO(3)-equivariant non-linear operator: when applying its non-linearity to a SO(3)-equivariant feature $\mathbf{f}$, the output feature $\mathbf{v}$ remains SO(3)-equivariant. □

## D    Information of Experimental Databases

In this part, we provide detailed information about the experimental databases, including the statistical information of the six databases from the DeepH benchmark series (Li et al., 2022; Gong et al., 2023) and the two databases from the QH9 benchmark series (Yu et al., 2023a), listed in Table 3 and Table 4, respectively. Additionally, we visualize two types of challenging testing samples: samples with non-rigid deformation from thermal motions, as well as the bilayer samples with interlayer twists, which are shown in Fig. 3 and Fig. 4, respectively.

## E    Implementation Details

The hardware environment for our experiments is a server cluster equipped with Nvidia RTX A6000 GPUs, each with 48 GiB of memory. Other experimental details may differ across the DeepH and QH9 benchmark series, which we will describe separately.

### E.1    Implementation Details on the DeepH Benchmark Series

The software environment used is Pytorch 2.0.1 for experiments on the six crystalline databases from the DeepH benchmark series. When combining the proposed TraceGrad method with the DeepH-E3 architecture, the implementation of DeepH-E3 is based on the project [3] provided by

---

[2] In Theorem 1 and Theorem 2, the Wigner-D matrices are in the complex and real fields, respectively, since the target quantity may be complex, whereas the internal representations of neural networks are typically in the real field. Nonetheless, neural network representations in the real field can still predict complex-valued targets with SO(3)-equivariance. Previous literature (Gong et al., 2023) has provided mechanisms for converting the network outputs in the real field into regression targets with real and imaginary parts.

[3] https://github.com/Xiaoxun-Gong/DeepH-E3

Gong et al. (2023), keeping the architecture and model hyperparameters consistent with their setup. In our framework, we use the same number of encoding modules $K$ as DeepH-E3, which is set to 3. In each encoding module, we apply the $g_{nonlin}(\cdot)$ module proposed in Section 4 and 5 to each SO(3)-equivariant edge feature, enabling non-linear expressiveness. We set the number of channels for the SO(3)-invariant feature $\mathbf{u}^{(k)}$ ($1 \leq k \leq 3$) to 1024. The neural network module $s_{nonlin}(\cdot)$ within $g_{nonlin}(\cdot)$ is implemented as a three-layers fully-connected module: the input size is set to 1024, consistent with $\mathbf{u}^{(k)}$; the hidden layer size is also 1024, with SiLU as the non-linear activation function and LayerNorm as the normalization mechanism; and the output layer size (i.e., the dimensionality of $\mathbf{z}^{(k)}$) is set to be equal to the number of basic blocks for a Hamiltonian matrix, which is 25 for $MG$ and $BG$, 49 for $MM$, and 196 for $BB$, $BT$, and $BS$. It is worth noting that, while $s_{nonlin}(\cdot)$ can be implemented as any differentiable neural network module, we here implement it as a simple fully-connected module. This decision is made to avoid adding significant computational burden to the whole network. Meanwhile, as DeepH-E3 already incorporates complex graph network mechanisms for information aggregation and message-passing, there is no need for $s_{nonlin}(\cdot)$ to be overly complex. Its role is focusing on to filling in the gaps left by the existing equivariant mechanisms in DeepH-E3: to introduce a non-linear mapping mechanism that maintains equivariance, thereby activating and unleashing the expressive power of the overall network architecture through non-linearity. The SO(3)-equivariant decoder we adopt is the same as that of DeepH-E3; The SO(3)-invariant decoder we adopt is a four-layers fully-connected module: the input size is 3 ($K$) times of the dimensionality of $\mathbf{z}^k$, e.g., 75 for $MG$; the hidden layers have 1024 neurons with SiLU as the non-linear activation function and LayerNorm as the normalization mechanism; the size of the output layer is the number of basic blocks for an atomic pairwise Hamiltonian matrix. Since each basic block of the Hamiltonian matrix can compute a trace, the total number of trace variables corresponds to the number of basic matrix blocks. Regarding the error metric in the loss function Eq. 7, for the first term, we follow DeepH-E3 to use MSE (Mean Squared Error); for the second term, we choose between MSE and MAE based on performance on the validation sets, ultimately selecting MAE. $\lambda$ in the training loss function is set according to parameter selection on the validation sets, searching from $\{0.1, 0.2, ..., 1.0\}$. Here, we aim to obtain a more general parameter setting for $\lambda$ on crystalline structures, and thus we determined $\lambda$ based on the overall performance on the validation sets of the six crystalline databases and the searched value is 0.3. To ensure the convergence of the TraceGrad method, we set the maximum training epochs to $5,000$. Other hyper-parameters and configurations are the same as DeepH-E3 (Gong et al., 2023): the initial learning rates for experiments on the $MG$, $MM$, $BG$, $BB$, $BT$, and $BS$ databases are set to 0.003, 0.005, 0.003, 0.005, 0.004, and 0.005, respectively; the training batch size is set as 1; the optimizer is chosen as Adam; the scheduler is configured as a slippery slope scheduler.

### E.2 IMPLEMENTATION DETAILS ON THE QH9 BENCHMARK SERIES

The software environment used is Pytorch 1.11.0 for experiments on the two molecular databases from the QH9 benchmark series. When combining the proposed TraceGrad method with the QHNet architecture, the implementation of QHNet is based on the project [4] provided by Yu et al. (2023b), keeping the architecture and network configurations consistent with their setup. In our framework, we use the same number of encoding modules as QHNet: 5 node feature encoding modules and 2 edge feature encoding modules. We opt to apply the $g_{nonlin}(\cdot)$ module proposed in Section 4 and 5 to each SO(3)-equivariant edge feature. We set the number of channels for $\mathbf{u}^{(k)}$ ($1 \leq k \leq 2$) as 1024. The neural network module $s_{nonlin}(\cdot)$ within $g_{nonlin}(\cdot)$ is implemented as a three-layers fully-connected module: the input size is set to 1024, consistent with $\mathbf{u}^{(k)}$, the hidden layer size is also 1024, with SiLU as the non-linear activation function and LayerNorm as the normalization mechanism, and the output layer size (i.e., the dimensionality of $\mathbf{z}^{(k)}$) is set to be equal to the number of basic blocks for a Hamiltonian matrix, which is 36 for $QS$ and $QD$ databases. The SO(3)-equivariant decoder we adopt is the same as that of QHNet; the SO(3)-invariant decoder we adopt is a four-layers fully-connected module: the input size is 2 ($K$) times of the dimensionality of $\mathbf{z}^{(k)}$, e.g., 72 for $QS$ and $QD$; the hidden layers have 1024 neurons with SiLU as the non-linear activation function and LayerNorm as the normalization mechanism; the size of the output layer is the number of basic blocks for an atomic pairwise Hamiltonian matrix. Regarding the error metric in the loss function Eq. 7, for the first term, we follow QHNet to use a combination of MSE and MAE; for the second term, we choose between MSE and MAE based on performance on the validation sets,

---

[4] https://github.com/divelab/AIRS

Table 3: Statistical information of the six benchmark databases, i.e., Monolayer Graphene ($MG$), Monolayer MoS2 ($MM$), Bilayer Graphene ($BG$), Bilayer Bismuthene ($BB$), Bilayer Bi2Te3 ($BT$), Bilayer Bi2Se3 ($BS$), from the DeepH benchmark series (Li et al., 2022; Gong et al., 2023). SOC: effects of Spin-Orbit Coupling. $m$: number of samples in the current dataset; $a_{max}$: maximum number of atoms from a unit cell in the current dataset. $a_{min}$: minimum number of atoms from a unit cell in the current dataset. $nt$: non-twisted samples. $t$: twisted samples.

| **Statistic Types** | | **MG** | **MM** | **BG** | **BB** | **BT** | **BS** |
|---|---|---|---|---|---|---|---|
| Elements | | C | Mo, S | C | Bi | Bi, Te | Bi, Se |
| SOC | | weak | weak | weak | strong | strong | strong |
| Training ($nt$) | $m$ | 270 | 300 | 180 | 231 | 204 | 231 |
| | $a_{max}$ | 72 | 75 | 64 | 36 | 90 | 90 |
| | $a_{min}$ | 72 | 75 | 64 | 36 | 90 | 90 |
| Validation ($nt$) | $m$ | 90 | 100 | 60 | 113 | 38 | 113 |
| | $a_{max}$ | 72 | 75 | 64 | 36 | 90 | 90 |
| | $a_{min}$ | 72 | 75 | 64 | 36 | 90 | 90 |
| Testing ($nt$) | $m$ | 90 | 100 | 60 | 113 | 12 | 113 |
| | $a_{max}$ | 72 | 75 | 64 | 36 | 90 | 90 |
| | $a_{min}$ | 72 | 75 | 64 | 36 | 90 | 90 |
| Testing ($t$) | $m$ | - | - | 9 | 4 | 2 | 2 |
| | $a_{max}$ | - | - | 1084 | 244 | 130 | 190 |
| | $a_{min}$ | - | - | 28 | 28 | 70 | 70 |

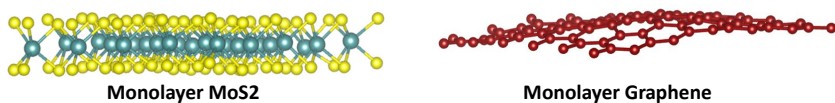

**Monolayer MoS2**   **Monolayer Graphene**

Figure 3: Visualization of testing samples exhibiting non-rigid deformations due to thermal motions

ultimately selecting MAE. $\lambda$ in the training loss function is set according to parameter selection on the validation sets, searching from $\{0.1, 0.2, ..., 1.0\}$. Here, we aim to obtain a more general parameter setting for $\lambda$ on molecular structures, and thus we determined $\lambda$ based on the overall performance on the validation sets of the two molecular databases and the searched value is $0.2$. Other hyper-parameters and configurations are the same as QHNet: the maximum training steps are set as $300,000$ for $QS$ and $260,000$ for $QD$, the initial learning rates for all experiments are set as $5 \times 10^{-4}$, the training batch size is set as $32$, the optimizer is set as AdamW, and a learning rate scheduler is implemented: the scheduler gradually increases the learning rate from $0$ to a maximum value of $5 \times 10^{-4}$ over the first $1,000$ warm-up steps. Subsequently, the scheduler linearly reduces the learning rate, ensuring it reaches $1 \times 10^{-7}$ by the final step.

## F   VISUALIZATION OF BLOCK-LEVEL MAE STATISTICS

As shown in Fig. 2, each Hamiltonian matrix consists of numerous basic block, with each basic block representing the direct product of two degrees. Here, we follow Yin et al. (2024) to measure the MAE performance of deep models on each basic block, denoted as $MAE_{block}^{H}$. The values of $MAE_{block}^{H}$ for the two setups, i.e., DeepH-E3 and DeepH-E3+TraceGrad, on different blocks of the Hamiltonian matrix for six databases from the DeepH benchmark series are illustrated in Fig. 5 and 6. Fig. 5 presents the results for monolayer structures, while Fig. 6 focuses on bilayer structures. From these figures, it can be observed that our method, TraceGrad, brings significant accuracy improvements over the baseline method, DeepH-E3, across the vast majority of blocks of the Hamiltonian matrices, particularly on blocks where DeepH-E3 struggles with lower accuracy.

Table 4: Statistical information of the two benchmark databases, QH9-stable ($QS$) and QH9-dynamic ($QD$), from the QH9 benchmark series (Yu et al., 2023a). The $QS$ database is split using the 'ood' strategy, while the $QD$ database is split using the 'mol' strategy. $m$: number of samples in the current dataset. $a_{max}$: maximum number of atoms for a sample in the current dataset. $a_{min}$: minimum number of atoms for a sample in the current dataset.

| **Statistic Types** | | **QS** | **QD** |
|---|---|---|---|
| Elements | | C, H, O, N, F | C, H, O, N, F |
| Training | $m$ | 104,001 | 79,900 |
| | $a_{max}$ | 20 | 19 |
| | $a_{min}$ | 3 | 10 |
| Validation | $m$ | 17,495 | 9,900 |
| | $a_{max}$ | 22 | 19 |
| | $a_{min}$ | 21 | 10 |
| Testing | $m$ | 9,335 | 10,100 |
| | $a_{max}$ | 29 | 19 |
| | $a_{min}$ | 23 | 10 |

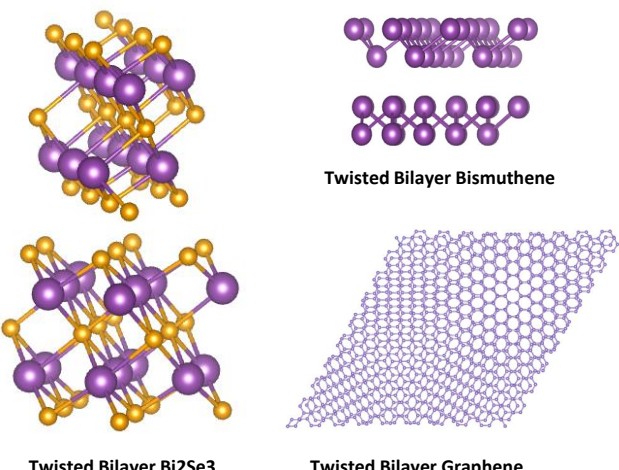

**Twisted Bilayer Bismuthene**

**Twisted Bilayer Bi2Se3**

**Twisted Bilayer Graphene**

Figure 4: Visualization of testing samples with interlayer twists.

## G ABLATION STUDY

We conduct fine-grained ablation study on the six databases from DeepH benchmark series, comparing results from the four setups:

• DeepH-E3 (Gong et al., 2023): the baseline model.

• DeepH-E3+Trace: this experimental setup, an ablation term, only implements half part of our method. Specifically, it extends the architecture of DeepH-E3 by adding our SO(3)-invariant encoding and decoding branches and using the **trace** quantity $\mathbf{T} = \mathrm{tr}(\mathbf{Q} \cdot \mathbf{Q}^{\dagger}) = \mathrm{tr}(\mathbf{H}^{l_p \otimes l_q} \cdot (\mathbf{H}^{l_p \otimes l_q})^{\dagger})$ to train them. As for ablation study, this setup does not include the gradient-based mechanism delivering non-linear expressiveness from SO(3)-invariant features to encode SO(3)-equivariant features; instead, it directly uses the SO(3)-equivariant features outputted by DeepH-E3 for Hamiltonian regression. In this configuration, the SO(3)-invariant branches only contribute indirectly during the training phase by backpropagating the supervision signals from the trace quantity to the earlier layers.

• DeepH-E3+Grad: this setup is also an ablation term and implements the other half part of our method in contrast to the previous ablation term. Specifically, it incorporates our SO(3)-invariant

encoder branch as well as the **grad**ient-induced operator to deliver non-linear expressiveness from SO(3)-invariant features to encode SO(3)-equivariant features. As for ablation study, this setup continues to use the single-task training pipeline of DeepH-E3, supervised only with the Hamiltonian label without joint supervised training through the trace of Hamiltonian.

• DeepH-E3+TraceGrad: this is a complete implementation of our framework extending beyond of the architecture and training pipeline of DeepH-E3, at the label level, we introduce the **trace** quantity to guide the learning of SO(3)-invariant features; Meanwhile, at the representation level, we leverage the **grad**ient operator to yield SO(3)-equivariant non-linear features for Hamiltonian prediction.

Table 5: Ablation study MAE results (meV) on the Monolayer Graphene ($MG$) and Monolayer MoS2 ($MM$) databases. ↓ means lower values of the metrics correspond to better accuracy.

| Methods | $MG$ | | | $MM$ | | |
|---|---|---|---|---|---|---|
| | $MAE (\downarrow)$ | | | | | |
| | $MAE_{all}^H$ | $MAE_{cha\_s}^H$ | $MAE_{cha\_b}^H$ | $MAE_{all}^H$ | $MAE_{cha\_s}^H$ | $MAE_{cha\_b}^H$ |
| DeepH-E3 (Baseline) | 0.251 | 0.357 | 0.362 | 0.406 | 0.574 | 1.103 |
| DeepH-E3+Trace | 0.230 | 0.344 | 0.348 | 0.378 | 0.537 | 1.091 |
| DeepH-E3+Grad | 0.185 | 0.269 | 0.258 | 0.308 | 0.453 | 0.924 |
| DeepH-E3+TraceGrad | **0.175** | **0.257** | **0.228** | **0.285** | **0.412** | **0.808** |

Experimental results of the four setups are listed in Table 5 and 6. Table 5 presents the results for monolayer structures, while Table 6 focuses on bilayer structures. From the results of ablation terms, we can obtain a more fine-grained experimental analysis. By comparing among the results of DeepH-E3, DeepH-E3+Trace, DeepH-E3+Grad, and DeepH-E3+TraceGrad, we can conclude that the two core mechanisms of our method, i.e., the SO(3)-invariant trace supervision mechanism (Trace) at the label level as well as the gradient-based induction mechanism (Grad) at the representation layer, can contribute to the performance individually. Moreover, their combination provides even better performance. This is because, on one hand, with the gradient-based induction mechanism as a bridge, the non-linear expressiveness of SO(3)-invariant features learned from the trace label can be transformed into the SO(3)-equivariant representations during inference; on the other hand, with trace label, the SO(3)-invariant network branch has a strong supervisory signal, enabling it to learn the intrinsic symmetry and complexity of the regression targets, enhancing the quality of SO(3)-invariant features and ultimately benefits the encoding of SO(3)-equivariant features. The value of such complementarity has been fully demonstrated in the experimental results.

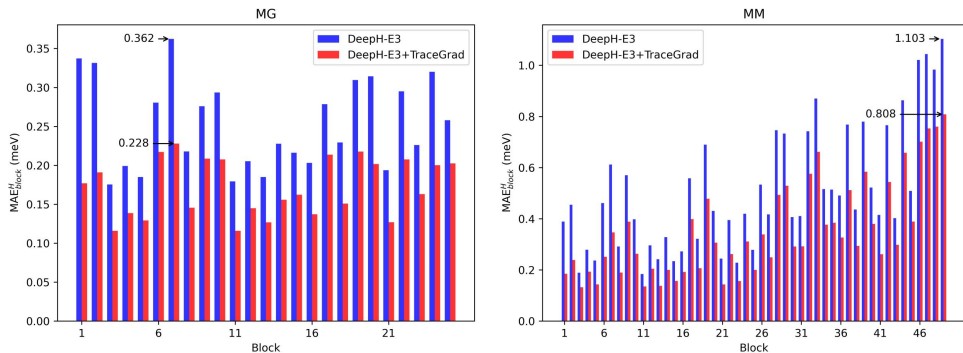

Figure 5: Visualization of $MAE_{block}^H$ on each basic block of the Hamiltonian matrices for the Monolayer Graphene ($MG$) and Monolayer MoS2 ($MM$) databases.

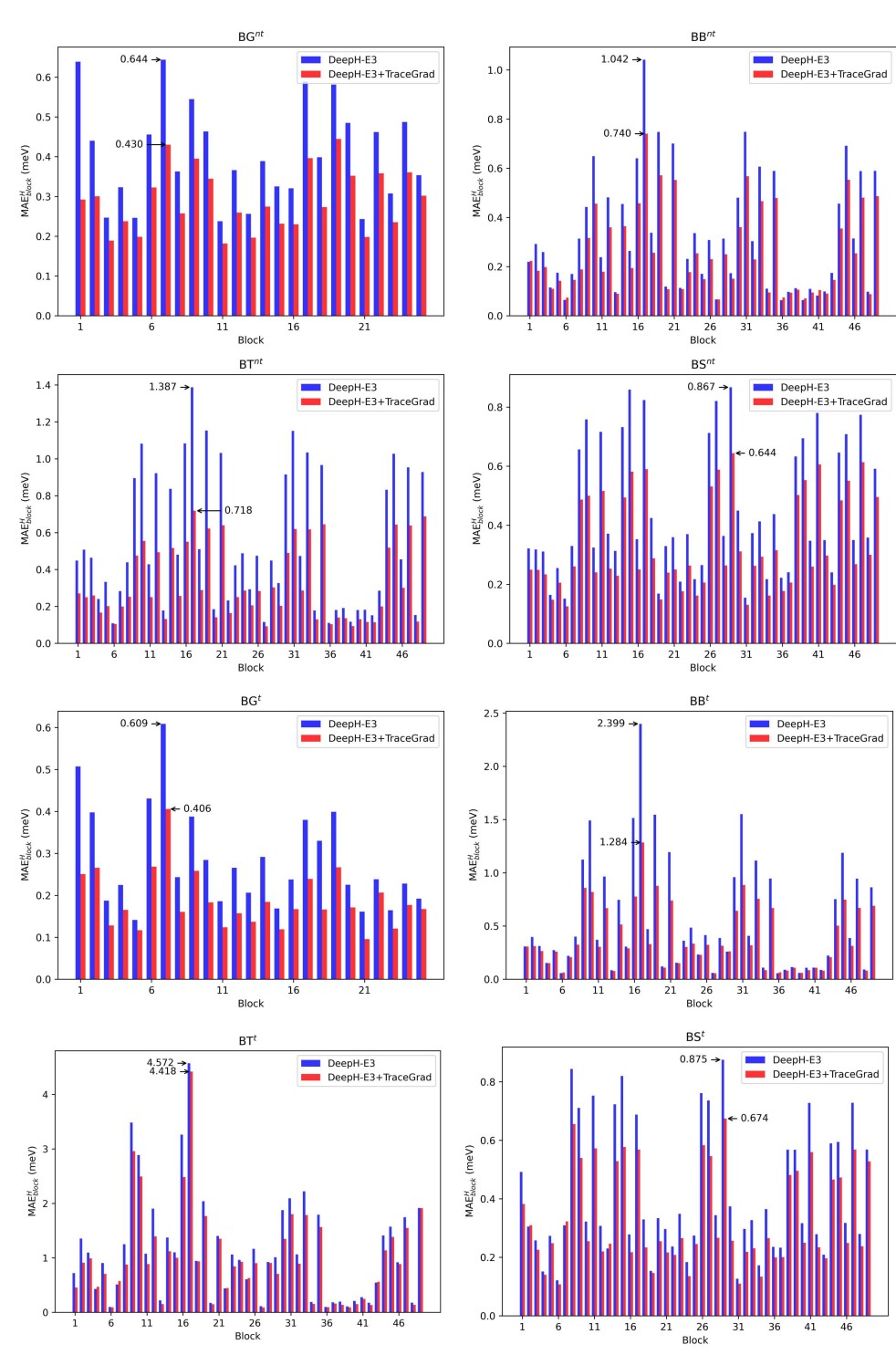

Figure 6: Visualization of $MAE^H_{block}$ on each basic block of the Hamiltonian matrices for the non-twisted (marked with superscripts $nt$) and twisted (marked with superscripts $t$) testing subsets of Bilayer Graphene ($BG$), Bilayer Bismuthene ($BB$), Bilayer Bi2Te3 ($BT$), and Bilayer Bi2Se3 ($BS$).

Table 6: Ablation study MAE results (meV) on the Bilayer Graphene ($BG$), Bilayer Bismuthene ($BB$), Bilayer Bi2Te3 ($BT$), and Bilayer Bi2Se3 ($BS$) databases. The superscripts $nt$ and $t$ respectively denote the non-twisted and twisted subsets. ↓ means lower values of the metrics correspond to better accuracy.

| Methods | $BG^{nt}$ | | | $BG^{t}$ | | |
| --- | --- | --- | --- | --- | --- | --- |
| | $MAE(\downarrow)$ | | | | | |
| | $MAE_{all}^{H}$ | $MAE_{cha\_s}^{H}$ | $MAE_{cha\_b}^{H}$ | $MAE_{all}^{H}$ | $MAE_{cha\_s}^{H}$ | $MAE_{cha\_b}^{H}$ |
| DeepH-E3 (Baseline) | 0.389 | 0.453 | 0.644 | 0.264 | 0.429 | 0.609 |
| DeepH-E3+Trace | 0.362 | 0.417 | 0.593 | 0.251 | 0.401 | 0.480 |
| DeepH-E3+Grad | 0.320 | 0.356 | 0.511 | 0.222 | 0.389 | 0.446 |
| DeepH-E3+TraceGrad | **0.291** | **0.323** | **0.430** | **0.198** | **0.372** | **0.406** |
| Methods | $BB^{nt}$ | | | $BB^{t}$ | | |
| | $MAE_{all}^{H}$ | $MAE_{cha\_s}^{H}$ | $MAE_{cha\_b}^{H}$ | $MAE_{all}^{H}$ | $MAE_{cha\_s}^{H}$ | $MAE_{cha\_b}^{H}$ |
| DeepH-E3 (Baseline) | 0.274 | 0.304 | 1.042 | 0.468 | 0.602 | 2.399 |
| DeepH-E3+Trace | 0.259 | 0.285 | 0.928 | 0.429 | 0.570 | 1.782 |
| DeepH-E3+Grad | 0.243 | 0.272 | 0.824 | 0.406 | 0.542 | 1.431 |
| DeepH-E3+TraceGrad | **0.226** | **0.256** | **0.740** | **0.384** | **0.503** | **1.284** |
| Methods | $BT^{nt}$ | | | $BT^{t}$ | | |
| | $MAE_{all}^{H}$ | $MAE_{cha\_s}^{H}$ | $MAE_{cha\_b}^{H}$ | $MAE_{all}^{H}$ | $MAE_{cha\_s}^{H}$ | $MAE_{cha\_b}^{H}$ |
| DeepH-E3 (Baseline) | 0.447 | 0.480 | 1.387 | 0.831 | 0.850 | 4.572 |
| DeepH-E3+Trace | 0.406 | 0.462 | 1.239 | 0.784 | 0.812 | 4.520 |
| DeepH-E3+Grad | 0.342 | 0.365 | 0.750 | 0.742 | 0.786 | 4.463 |
| DeepH-E3+TraceGrad | **0.295** | **0.312** | **0.718** | **0.735** | **0.755** | **4.418** |
| Methods | $BS^{nt}$ | | | $BS^{t}$ | | |
| | $MAE_{all}$ | $MAE_{cha\_s}^{H}$ | $MAE_{cha\_b}^{H}$ | $MAE_{all}$ | $MAE_{cha\_s}^{H}$ | $MAE_{cha\_b}^{H}$ |
| DeepH-E3 (Baseline) | 0.397 | 0.424 | 0.867 | 0.370 | 0.390 | 0.875 |
| DeepH-E3+Trace | 0.382 | 0.397 | 0.843 | 0.351 | 0.367 | 0.838 |
| DeepH-E3+Grad | 0.343 | 0.365 | 0.696 | 0.324 | 0.339 | 0.746 |
| DeepH-E3+TraceGrad | **0.300** | **0.332** | **0.644** | **0.291** | **0.302** | **0.674** |

## H EMPIRICAL STUDY COMPARING THE PROPOSED GRADIENT-BASED MECHANISM WITH THE GATED ACTIVATION MECHANISM

Taking the Bilayer Graphene ($BG$) and Bilayer Bismuthene ($BB$) databases as representative, we conduct an empirical study comparing our proposed gradient-based mechanism with the gated activation mechanism. We compare the accuracy performance of the following four experimental setups:

• DeepH-E3: the baseline model.

• DeepH-E3+Grad: Same as that introduced in Appendix G.

• DeepH-E3+Gate: This variant modifies the experimental setup from DeepH-E3+Grad by replacing the gradient mechanism, which constructs equivariant features as $\mathbf{v} = \frac{\partial z}{\partial \mathbf{f}}$, with a gated activation mechanism, constructing equivariant features as $\mathbf{v} = z \cdot \mathbf{f}$. All other aspects remain the same.

• DeepH-E3+TraceGrad: Same as that introduced in Appendix G, this is a complete implementation of our framework combined with DeepH-E3.

• DeepH-E3+TraceGate: This variant modifies the experimental setup from DeepH-E3+TraceGrad by replacing the gradient mechanism, which constructs equivariant features as $\mathbf{v} = \frac{\partial z}{\partial \mathbf{f}}$, with a gated activation mechanism, constructing equivariant features as $\mathbf{v} = z \cdot \mathbf{f}$. All other aspects remain the same.

Experimental results are recorded in the following Table:

Table 7: Comparison between the gradient-based mechanism (Grad) and the gated activation mechanism (Gate): MAE results (meV) on the Bilayer Graphene ($BG$) and Bilayer Bismuthene ($BB$) databases. The superscripts $nt$ and $t$ respectively denote the non-twisted and twisted subsets. $\downarrow$ indicates that lower values of the metrics correspond to better accuracy.

| Methods | $BG^{nt}$ | | | $BG^{t}$ | | |
|---|---|---|---|---|---|---|
| | $MAE$ ($\downarrow$) | | | | | |
| | $MAE_{all}^{H}$ | $MAE_{cha\_s}^{H}$ | $MAE_{cha\_b}^{H}$ | $MAE_{all}^{H}$ | $MAE_{cha\_s}^{H}$ | $MAE_{cha\_b}^{H}$ |
| DeepH-E3 (Baseline) | 0.389 | 0.453 | 0.644 | 0.264 | 0.429 | 0.609 |
| DeepH-E3+Grad | 0.320 | 0.356 | 0.511 | 0.222 | 0.389 | 0.446 |
| DeepH-E3+Gate | 0.368 | 0.441 | 0.626 | 0.241 | 0.405 | 0.602 |
| DeepH-E3+TraceGrad | **0.291** | **0.323** | **0.430** | **0.198** | **0.372** | **0.406** |
| DeepH-E3+TraceGate | 0.354 | 0.403 | 0.580 | 0.239 | 0.388 | 0.464 |
| **Methods** | $BB^{nt}$ | | | $BB^{t}$ | | |
| | $MAE_{all}^{H}$ | $MAE_{cha\_s}^{H}$ | $MAE_{cha\_b}^{H}$ | $MAE_{all}^{H}$ | $MAE_{cha\_s}^{H}$ | $MAE_{cha\_b}^{H}$ |
| DeepH-E3 (Baseline) | 0.274 | 0.304 | 1.042 | 0.468 | 0.602 | 2.399 |
| DeepH-E3+Grad | 0.243 | 0.272 | 0.824 | 0.406 | 0.542 | 1.431 |
| DeepH-E3+Gate | 0.268 | 0.301 | 0.991 | 0.450 | 0.593 | 2.276 |
| DeepH-E3+TraceGrad | **0.226** | **0.256** | **0.740** | **0.384** | **0.503** | **1.284** |
| DeepH-E3+TraceGate | 0.252 | 0.279 | 0.908 | 0.417 | 0.561 | 1.740 |

From these results, it is evident that both DeepH-E3+Gate and DeepH-E3+TraceGate show improvement over the baseline DeepH-E3, as these configurations introduce an additional neural network module $s_{nonlin}(\cdot)$ in learning the feature $z$, which increases model capacity. Moreover, DeepH-E3+TraceGate also incorporates our proposed trace supervision signal, which guides the learning of SO(3)-invariant features. However, their performance falls short of DeepH-E3+Grad and DeepH-E3+TraceGrad, respectively, indicating that the gated activation mechanism may not fully capture the system's non-linearity patterns, whereas the proposed gradient-based mechanism demonstrates stronger generalization performance on Hamiltonian prediction and may be a better choice in terms of expressive capability.

## I  THEORETICAL ANALYSIS ON COMPUTATIONAL COMPLEXITY

The total number of non-zero Hamiltonian matrix elements to be calculated is proportional to the number of local atomic pairs in the system, with a complexity of $\mathcal{O}(N\overline{E})$, where $N$ is the total number of atoms and $\overline{E}$ is the average number of neighboring atoms within the cutoff radius per atom. Since the atomic orbital basis set has a finite range, the Hamiltonian matrix elements vanish beyond a certain distance. In small systems, where all atoms lie within each other's cutoff radius, $\overline{E}$ scales with $N$, resulting in a total number of non-zero elements proportional to $N^2$. However, in sufficiently large systems, the finite range of the atomic orbitals ensures that $\overline{E}$ remains constant, independent of $N$. As a result, for large atomic systems, the total number of non-zero elements simplifies to $\mathcal{O}(N)$.

The baseline models we select, whether DeepH-E3 or QHNet, are SO(3)-equivariant graph neural network models with efficient information aggregation and message-passing mechanisms. These models cleverly balance the locality of Hamiltonian definitions with the long-range interactions present in the system. As a result, the computational complexity asymptotically scales as $\mathcal{O}(N)$ as $N$ increases, which is consistent with the growth of the scales of non-zero Hamiltonian matrix elements. The proposed TraceGrad method directly updates each SO(3)-equivariant feature of the baseline models, and the computational amount is proportinal to the number of features of the baseline models. Therefore, combining TraceGrad, the computational complexity also scales as $\mathcal{O}(N)$.

Traditional DFT methods require $T$ iterations of diagonalizing $N \times N$ matrices, each with a time complexity of $\mathcal{O}(N^3)$, because all occupied states are needed to compute the charge density. As $N$

Table 8: Average inference time per test sample (abbreviated as Time) in seconds and average MAE performance in meV ($MAE_{all}^H$) on the Monolayer Graphene ($MG$), Monolayer MoS2 ($MM$), QH9-stable ($QS$), and QH9-dynamic ($QD$) databases. ↓ indicates that lower values of the metrics correspond to better accuracy. All models are tested individually on a single Nvidia RTX A6000 in single-task mode.

| Methods | $MG$ | | $MM$ | |
|---|---|---|---|---|
| | $Time$ (↓) | $MAE_{all}^H$ (↓) | $Time$ (↓) | $MAE_{all}^H$ (↓) |
| DeepH-E3 (Baseline) | 0.247 | 0.251 | 0.256 | 0.406 |
| DeepH-E3$^{\times 2}$ | 0.483 | 0.244 | 0.510 | 0.387 |
| DeepH-E3+TraceGrad | 0.264 | 0.175 | 0.274 | 0.285 |

| Methods | $QS$ | | $QD$ | |
|---|---|---|---|---|
| | $Time$ (↓) | $MAE_{all}^H$ (↓) | $Time$ (↓) | $MAE_{all}^H$ (↓) |
| QHNet (Baseline) | 0.233 | 1.962 | 0.174 | 4.733 |
| QHNet$^{\times 2}$ | 0.497 | 1.845 | 0.385 | 4.532 |
| QHNet+TraceGrad | 0.248 | 1.191 | 0.187 | 2.819 |

increases, this cubic complexity leads to significant computational overhead, making it challenging to simulate large atomic systems within a reasonable time frame. In contrast, our deep learning framework enables the efficient and accurate construction of the Hamiltonian for large atomic systems with a linear time complexity of $\mathcal{O}(N)$, eliminating the need for self-consistent iterations. Moreover, since most physical properties, such as transport, optical, and topological properties, depend only on the energy bands near the Fermi level, it is unnecessary to solve for the eigenfunctions of all occupied states once the Hamiltonian is known. Since the Hamiltonian matrix is sparse and only a limited number of bands near the Fermi level are needed, these eigenstates can be efficiently computed using methods like the shift-invert approach available in the ARPACK package (Lehoucq et al., 1998), with a computational complexity of $\mathcal{O}(N)$.

## J A JOINT DISCUSSION ON GPU TIME COSTS AND PERFORMANCE GAINS

We here provide a joint comparison of the GPU time cost and corresponding accuracy of different models across four databases as representative: Monolayer Graphene ($MG$), Monolayer MoS2 ($MM$), QH9-stable ($QS$), and QH9-dynamic ($QD$). This comparison includes the average inference time per sample for each model, with the test batch size set as 1 and hardware environment set as Nvidia RTX A6000 GPU in single-task mode without computational sharing with other processes.

For $MG$ and $MM$, we compare among DeepH-E3, DeepH-E3+TraceGrad, and DeepH-E3$^{\times 2}$, where DeepH-E3$^{\times 2}$ refers to a model obtained by doubling the number of encoding blocks in DeepH-E3 and training it from scratch until convergence. For $QS$ and $QD$, we compare among QHNet, QHNet+TraceGrad, and QHNet$^{\times 2}$, where QHNet$^{\times 2}$ refers to a model obtained by doubling the number of encoding blocks in QHNet and training it from scratch until convergence. The experimental results are documented in the Table 8.

From this Table, we find that adding the TraceGrad module results in only a slight increase in inference time compared to the baseline models. Given the substantial accuracy improvements introduced by the TraceGrad method, we consider this minor increase in computational time acceptable for practical applications. In contrast, simply increasing the depth of DeepH-E3 or QHNet results in a significant rise in inference time but yields only limited accuracy improvements. In contrast, DeepH-E3+TraceGrad demonstrates significantly better accuracy performance than DeepH-E3$^{\times 2}$, and similarly, QHNet+TraceGrad achieves notably higher accuracy than QHNet$^{\times 2}$. Furthermore, the inference time of DeepH-E3+TraceGrad and QHNet+TraceGrad are both lower than their respective DeepH-E3$^{\times 2}$ and QHNet$^{\times 2}$ counterparts. These results underscore the superiority of the TraceGrad method in enhancing expressive capability and improving accuracy performance, while maintaining time efficiency.

## K    ACCELERATION PERFORMANCE FOR THE CONVERGENCE OF TRADITIONAL DFT ALGORITHMS

Despite the increasing ability of deep learning models to independently handle more electronic-structure computation tasks, there are still applications with extremely high numerical precision requirements and very low tolerance for error, where traditional DFT algorithms must perform the final calculations. In such cases, the predictions from deep models can be used as initial matrices to accelerate the convergence of traditional DFT algorithms. We evaluate the acceleration performance brought by the proposed method for the convergence of classical DFT algorithms implemented by PySCF (Sun et al., 2018). Specifically, we adopt the two groups of metrics on acceleration performance:

The first group of metrics are defined in Yu et al. (2023a), as follows:

- Achieved ratio. This metric calculates the number of DFT optimization steps taken when initializing with the Hamiltonian matrices predicted by the deep model compared to using initial guess methods like `minao` and `1e`.
- Error-level ratio. This metric measures the number of DFT optimization steps required, starting from random initialization, to reach the same error level as the deep model's predictions, relative to the total number of steps in the DFT process.

Experimental results on these metrics are recorded in Table 9, where the results for the compared method QHNet are taken from Yu et al. (2023a), while the results of QHNet+TraceGrad, come from our experiments. In our experiments, the DFT calculation settings follow those in Section 4 of Yu et al. (2023a) (the DFT parameters and the 50 testing samples) , except for the CPU environment, where we use a single thread of an Intel(R) Xeon(R) Gold 6330 @ 2.00GHz CPU in single-task mode. It is worth noting that this does not affect the fairness of the comparison, as the achieved ratio and error-level ratio measure the ratio of iteration counts rather than runtime, and differences in CPU computation times are negligible in these metrics. From Table 9, we could observe that the proposed TraceGrad method brings significant improvements to the baseline model QHNet on the acceleration ratio of DFT calculation, notably reducing the achieved ratio and enhancing the error-level ratio.

The second group of metrics measures the wall time savings that deep learning methods contribute to DFT calculations, specifically quantifying the incremental time savings brought by our proposed TraceGrad method in accelerating DFT calculations. For a fair comparison, we report the average wall time costs per sample (/s) across three metrics:

- $t1$: The wall time required for a DFT calculation initialized with a random guess initialization, such as `1e` or `minao`.
- $t2$: The wall time for inference using deep learning methods (i.e., QHNet or QHNet+TraceGrad).
- $t3$: The total wall time for the combined process, including both deep learning inference and the subsequent DFT calculation initialized with the deep model's outputs. $t3$ here provides a more comprehensive evaluation of the actual time savings achieved when incorporating deep learning methods.

Experimental results on these metrics are recorded in Table 10. Here all time-related measurements are conducted on a single thread of an Intel(R) Xeon(R) Gold 6330 @ 2.00GHz CPU, including the experiments for QHNet, which are reproduced under the same conditions to ensure fairness. Unlike the time measurements in Appendix J, which are performed on GPUs, all deep learning models here are evaluated on the CPU thread to maintain consistency in the comparison with DFT software. From the experimental results, we observe three key findings:

- Comparing $t2$ and $t1$, deep models are significantly faster than DFT calculations, achieving speeds tens of times greater for the testing samples. It is worth noting that, given that the testing samples here are all small molecular systems, deep models have already demonstrated a significant time efficiency advantage compared to DFT. Based on the computational complexity analysis of deep learning methods compared to DFT methods in

Table 9: The acceleration ratios of QHNet and QHNet+TraceGrad for DFT calculation. Both models are evaluated on a set of $50$ molecules chosen by Yu et al. (2023a), with the mean and standard deviation of the metrics across these samples reported. $\downarrow$ means lower values correspond to better accuracy, while $\uparrow$ means higher values correspond to better performance.

| Methods | Training databases | DFT initialization | Metric | Ratio |
|---------|--------------------|--------------------|--------|-------|
| QHNet (Baseline) | QS | 1e | Achieved ratio $\downarrow$ | $0.400 \pm 0.030$ |
| | | | Error-level ratio $\uparrow$ | $0.620 \pm 0.037$ |
| | | minao | Achieved ratio $\downarrow$ | $0.715 \pm 0.033$ |
| | | | Error-level ratio $\uparrow$ | $0.406 \pm 0.021$ |
| | QD | 1e | Achieved ratio $\downarrow$ | $0.512 \pm 0.138$ |
| | | | Error-level ratio $\uparrow$ | $0.622 \pm 0.048$ |
| | | minao | Achieved ratio $\downarrow$ | $0.882 \pm 0.217$ |
| | | | Error-level ratio $\uparrow$ | $0.406 \pm 0.066$ |
| QHNet+TraceGrad | QS | 1e | Achieved ratio $\downarrow$ | $\mathbf{0.345} \pm 0.038$ |
| | | | Error-level ratio $\uparrow$ | $\mathbf{0.685} \pm 0.037$ |
| | | minao | Achieved ratio $\downarrow$ | $\mathbf{0.647} \pm 0.061$ |
| | | | Error-level ratio $\uparrow$ | $\mathbf{0.466} \pm 0.035$ |
| | QD | 1e | Achieved ratio $\downarrow$ | $\mathbf{0.440} \pm 0.101$ |
| | | | Error-level ratio $\uparrow$ | $\mathbf{0.645} \pm 0.046$ |
| | | minao | Achieved ratio $\downarrow$ | $\mathbf{0.761} \pm 0.167$ |
| | | | Error-level ratio $\uparrow$ | $\mathbf{0.435} \pm 0.052$ |

    Appendix I, it is reasonable to infer that for larger atomic systems, the disparity between $t2$ and $t1$ will expand rapidly.

- Comparing $t2$ values between QHNet and QHNet+TraceGrad, the additional runtime introduced by TraceGrad is relatively minor on the CPU, consistent with the GPU-based results reported in Appendix J.

- Comparing $t3$ and $t1$, the total runtime of using a deep model to predict initial values and then running DFT calculations is significantly lower than performing DFT calculations from a random guess initialization. Particularly, comparing row 4 and row 16, row 7 and row 19, row 10 and row 22, as well as row 13 and row 25 in Table 10, it can be observed that combing TraceGrad further reduces $t_3$, demonstrating that the time saved by TraceGrad in DFT calculations far exceeds the minimal additional time required for its inference.

## L  EMPIRICAL STUDY ON COMBINING OUR METHOD WITH APPROXIMATELY SO(3)-EQUIVARIANCE FRAMEWORK

While non-strict SO(3)-equivariance, which may limit the depth of theoretical exploration, is not the main focus of this study aiming at bridging rigorous SO(3)-equivariance with the non-linear expressive capabilities of neural networks, considering that they remain of interest in a few numerical computation applications where precision is highlighted over strict equivariance, we also conduct empirical study combining our method with approximately SO(3)-equivariant techniques. Taking the Monolayer MoS2 ($MM$) database as a case study, we evaluate the performance of combining our trace supervision and gradient induction method (TraceGrad) with the an approximately equivariant approach HarmoSE (Yin et al., 2024). We here take HarmoSE as the backbone encoder, and yields features by TraceGrad to enrich its representations. The experimental results in Table 11 and Fig. 7 demonstrate that TraceGrad significantly enhances the accuracy of HarmoSE, surpassing DeepH-2 and achieving SOTA results. Both DeepH-2 and HarmoSE sacrificed strict SO(3)-equivariance to fully release the expressive capabilities of graph Transformers, aiming for the ultimate in prediction accuracy. Despite this, our method still manages to significantly exceed their accuracy, further confirming the superiority of our method in learning expressive representations of physical systems.

Table 10: Average wall time costs per sample (/s) for three experimental settings: $t1$ represents the wall time required for DFT calculation with a random initial guess like 1e of minao; $t2$ denotes the wall time for inference of deep learning methods (i.e., QHNet or QHNet+TraceGrad); $t3$ is the total time of the process including deep learning inference and the DFT calculation that uses the outputs of deep learning methods as initialization. Both models are evaluated on a set of 50 molecules chosen by Yu et al. (2023a), with the mean and standard deviation of the metrics across these samples reported. ↓ indicates that lower values correspond to better performance. In order to ensure fairness in comparison, all experimental settings including DFT calculation and deep learning inference are measured on a single thread of an Intel(R) Xeon(R) Gold 6330 @ 2.00GHz CPU in single-task mode.

| Methods | Training databases | DFT initialization | Metric | Time |
|---|---|---|---|---|
| QHNet (Baseline) | QS | 1e | t1 | $120.896 \pm 9.134$ |
| | | | t2 ↓ | $1.724 \pm 0.025$ |
| | | | t3 ↓ | $48.604 \pm 7.741$ |
| | | minao | t1 | $63.193 \pm 5.335$ |
| | | | t2 ↓ | $1.724 \pm 0.025$ |
| | | | t3 ↓ | $48.604 \pm 7.741$ |
| | QD | 1e | t1 | $87.161 \pm 12.075$ |
| | | | t2 ↓ | $1.280 \pm 0.019$ |
| | | | t3 ↓ | $44.146 \pm 9.342$ |
| | | minao | t1 | $51.396 \pm 5.870$ |
| | | | t2 ↓ | $1.280 \pm 0.019$ |
| | | | t3 ↓ | $44.146 \pm 9.342$ |
| QHNet+TraceGrad | QS | 1e | t1 ↓ | $120.896 \pm 9.134$ |
| | | | t2 ↓ | $1.852 \pm 0.020$ |
| | | | t3 ↓ | $\mathbf{41.941 \pm 6.783}$ |
| | | minao | t1 | $63.193 \pm 5.335$ |
| | | | t2 ↓ | $1.852 \pm 0.020$ |
| | | | t3 ↓ | $\mathbf{41.941 \pm 6.783}$ |
| | QD | 1e | t1 | $87.161 \pm 12.075$ |
| | | | t2 ↓ | $1.361 \pm 0.010$ |
| | | | t3 ↓ | $\mathbf{39.712 \pm 9.076}$ |
| | | minao | t1 | $51.396 \pm 5.870$ |
| | | | t2 ↓ | $1.361 \pm 0.010$ |
| | | | t3 ↓ | $\mathbf{39.712 \pm 9.076}$ |

Table 11: MAE results (meV) for DeepH-2, HarmoSE, and HarmoSE+TraceGrad on the $MM$ database. ↓ means lower values of the metrics correspond to better accuracy. The results of the compared methods are taken from the corresponding literature (Wang et al., 2024b; Yin et al., 2024), where the empty items are due to the data not being provided in the original paper.

| Methods | $MM$ | | |
|---|---|---|---|
| | $MAE$ (↓) | | |
| | $MAE_{all}^H$ | $MAE_{cha\_s}^H$ | $MAE_{cha\_b}^H$ |
| **DeepH-2** (Wang et al., 2024b) | 0.21 | - | - |
| **HarmoSE** (Yin et al., 2024) | 0.233 | 0.293 | 0.406 |
| **HarmoSE+TraceGrad** | **0.178** | **0.228** | **0.296** |

## M  FUTURE WORK

In future research, various extensions are conceivable across theoretical, methodological, and application fields:

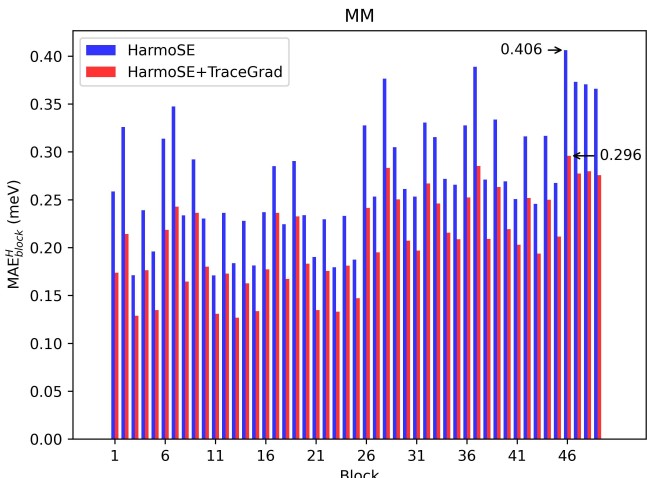

Figure 7: Comparison on the $MAE_{block}^H$ metric for HarmoSE and HarmoSE+TraceGrad on the $MM$ database.

First, from a theoretical and methodological perspective, we may extend our framework on achieving SO(3) symmetry with expressiveness to a broader range and more complex groups, e.g., SU(2), SO(n), and Spin(n), in deep learning research. This would enable deep learning frameworks to incorporate richer mathematical structures and physical quantities.

Second, while our theoretical framework, detailed in Theorem 1 and Theorem 2, incorporating concepts such as SO(3)-equivariant variables $\mathbf{Q}$, SO(3)-invariant features $\mathbf{T}$, SO(3)-equivariant features $\mathbf{f}$ and $\mathbf{v}$, and SO(3)-invariant features $z$, along with their mathematical relationships (proven in Appendix C), is general in nature and not limited to specific physical quantities, from an application perspective, the empirical effectiveness of our method beyond predicting Hamiltonian and its downstream quantities remains to be validated. In future work, we plan to extend the current methodology beyond predicting electronic-structure Hamiltonians for predicting a wide range of physical quantities and properties that exhibit equivariance, such as force constant matrices, Born effective charges, and more.

Furthermore, in principle, our approach also has potential to find applications in fields such as robotics, autonomous vehicles, and motion tracking systems to harmonize SO(3)-equivariance with non-linear expressiveness. In these areas, data are typically represented as 3D point clouds, which is conceptually similar to describing the 3D structure of atomic systems using atomic point clouds. This similarity facilitates the transferability of our method to these domains. For example, in vision tasks such as autonomous vehicles (Wang et al., 2023), the relative positions of cameras and objects are not fixed, causing the sampled 3D point clouds to undergo coordinate transformations, with rotation being a common example. Given the safety-critical nature of these tasks, ensuring the reliability and robustness of pattern recognition systems are of utmost importance. Consequently, there is a significant demand for systems that are robust to coordinate transformations of 3D point clouds. The mainstream approach has been to approximate SO(3)-equivariance through data augmentation. However, this method does not ensure absolute reliability or safety. Our work suggests considerable potential for constructing deep models with strong generalization performance, grounded in strict SO(3)-equivariance, and could contribute to advancements in these fields. Therefore, our next step may involve applying our method to the autonomous vehicles domain for 3D point cloud object segmentation and recognition, with the goal of achieving more robust recognition results. Specifically, in the task of 3D point cloud object recognition, the position vectors of the corner points of 3D bounding boxes relative to their center point are SO(3)-equivariant quantities, corresponding to $\mathbf{Q}$ in this work, while their magnitudes are SO(3)-invariant quantities, corresponding to $\mathbf{T}$ in this work. These can serve as regression targets and supervision signals for the SO(3)-equivariant and

SO(3)-invariant branches of our method, respectively, and may enable the learning of informative SO(3)-equivariant non-linear features for regressing 3D bounding boxes.

To summarize, while our method is theoretically applicable to other tasks, its effectiveness in those areas has yet to be demonstrated. Much work remains to be done.

