# OpenReview forum: "A Framework of SO(3)-equivariant Non-linear Representation Learning and its Application to Electronic-Structure Hamiltonian Prediction"
_ICLR.cc/2025/Conference — ICLR 2025 Conference Withdrawn Submission_

### Official Review · Reviewer_7Q7d · 2024-10-24

**Soundness:** 3
**Presentation:** 3
**Contribution:** 3
**Rating:** 6
**Confidence:** 4

**Summary:**

The paper introduces a method combining SO(3)-equivariance with non-linear expressiveness, and applied it for Hamiltonian prediction. This method has a theoretical foundation, and experiments demonstrate the effectiveness of the approach.

**Strengths:**

- Theoretical analysis is provided to ensure equivariance is maintained alongside non-linear expressiveness.
- Experimental results indicate significant improvement in Hamiltonian prediction accuracy, outperforming QHNet and DeepH-E3.
- Detailed explanations are provided for the experimental setup, showcasing a comprehensive evaluation.
- The proposed method is simple and can be easily combined with existing approaches.

**Weaknesses:**

- To show whether the acceleration in DFT outweighs the additional computation, it would be helpful if the paper provided a comparison of the wall time cost for accelerating traditional DFT algorithms.
- Since SO(3)-equivariance is important in many contexts, could the authors provide an example of how this model might be applied to another prediction task?

**Questions:**

see above

---

> ### Author Response · Authors · 2024-11-25
> **Response to Reviewer 4**
>
> Thank you sincerely for your thoughtful and insightful feedback. We deeply appreciate the time and effort you dedicated to reviewing our manuscript. Your suggestions have been instrumental in improving both the clarity and depth of our work. We have carefully considered and addressed each of your comments during the revision process. We hope that our responses and the changes made to the manuscript meet your expectations. Below, we provide a detailed, point-by-point response to your feedback, along with a summary of the corresponding revisions.

---

> ### Author Response · Authors · 2024-11-25
> **Response to Weakness 1**
>
> **Comment from the reviewer**: To show whether the acceleration in DFT outweighs the additional computation, it would be helpful if the paper provided a comparison of the wall time cost for accelerating traditional DFT algorithms.
>
> **Response**:
> Thank you for your valuable feedback! In Appendix K titled ``Acceleration Performance for the Convergence of Traditional DFT Algorithms" of the revised paper, we add a group of metrics measures the wall time savings that deep learning methods contribute to DFT calculations, specifically quantifying the incremental time savings brought by our proposed TraceGrad method in accelerating DFT calculations. For a fair comparison, we report the average wall time costs per sample (/s) across three metrics:
>
> - $t1$: The wall time required for a DFT calculation initialized with a random guess initialization, such as $1e$ or $minao$.
> - $t2$: The wall time for inference using deep learning methods (i.e., QHNet or QHNet+TraceGrad).
> - $t3$: The total wall time for the combined process, including both deep learning inference and the subsequent DFT calculation initialized with the deep model's outputs. $t3$ here provides a more comprehensive evaluation of the actual time savings achieved when incorporating deep learning methods.
>
>
> Experimental results on these metrics are recorded in the Table 10 of the revised paper (**Due to the complexity of this table, it is quite challenging to convert it into Markdown. We  kindly ask the reviewer to refer to the revised paper for the table and apologize for the inconvenience**). Here all time-related experiments are conducted on a single thread of an Intel(R) Xeon(R) Gold 6330 @ 2.00GHz CPU, including the experiments for QHNet, which are reproduced under the same conditions to ensure fairness. Unlike the time measurements in Appendix J, which are performed on GPUs, all deep learning models here are evaluated on the CPU to maintain consistency in the comparison with DFT software. From the experimental results, we observe three key findings:
>
> - Comparing $t2$ and $t1$, deep models are significantly faster than DFT calculations, achieving speeds tens of times greater for the testing samples. It is worth noting that, given that the testing samples here are all small molecular systems, deep models have already demonstrated a significant time efficiency advantage compared to DFT. Based on the computational complexity analysis of deep learning methods compared to DFT methods in Appendix I, it is reasonable to infer that for larger atomic systems, the disparity between $t2$ and $t1$ will expand rapidly.
>
> - Comparing $t2$ values between QHNet and QHNet+TraceGrad, the additional runtime introduced by TraceGrad is minor on the CPU, consistent with the GPU-based results reported in Appendix J.
>
> - Comparing $t3$ and $t1$, the total runtime of using a deep model to predict initial values and then running DFT calculations is significantly lower than performing DFT calculations from a random guess initialization. Particularly, comparing row 4 and row 16, row 7 and row 19, row 10 and row 22, as well as row 13 and row 25 in  Table 10 of the revised paper, **it can be observed that combing TraceGrad further reduces $t_3$, demonstrating that the time saved by TraceGrad in DFT calculations far exceeds the minimal additional time required for its inference**.

---

> ### Author Response · Authors · 2024-11-26
> **Response to Weakness 2**
>
> **Comment from the reviewer**: Since SO(3)-equivariance is important in many contexts, could the authors provide an example of how this model might be applied to another prediction task?
>
> **Response**: Thanks for the good suggestion! We discuss how this model might be applied to another prediction task in Appendix M, titled ``Future Work", of the revised paper.
>
> From the application perspective, the current methodology extends beyond predicting electronic-structure Hamiltonians and is capable of generally predicting multiple physical quantities and properties that are equivariant, such as force constant matrices and Born effective charges, and more.
> Furthermore, our approach provides an effective and strictly equivariant deep learning method, extending its applicability beyond physics to fields such as robotics, autonomous vehicles, and motion tracking.
>  In these fields, data is typically represented as 3D point clouds, conceptually similar to our description of 3D atomic system structures using atomic point clouds. In vision tasks like autonomous vehicles [1], the relative positions of cameras and objects are not fixed, resulting in the sampled 3D point clouds being subject to coordinate transformations, with rotation being a common transformation. Given the safety-critical nature of these tasks, the reliability and robustness of pattern recognition systems are of utmost importance. The mainstream approach achieves approximate $\mathrm{SO}(3)$-equivariance through data augmentation, but this lacks absolute reliability and safety.
>  In contrast, our work provide a strict  SO(3)-equivariance deep-learning method to solve this critical problem. For example, in the task of 3D point cloud object recognition, the position vectors of the corner points of 3D bounding boxes relative to their center point are SO(3)-equivariant quantities ($\mathbf{Q}$), while the magnitudes of these vectors are SO(3)-invariant ($\mathbf{T}$).
>  These can serve as regression targets and supervision signals for the SO(3)-equivariant and SO(3)-invariant branches of our method, respectively, to learn informative SO(3)-equivariant non-linear features for regressing 3D bounding boxes. In summary, looking forward to future endeavors, the foundational and pivotal role of this work is clear.
>
> [1] Yingjie Wang, Qiuyu Mao, Hanqi Zhu, Jiajun Deng, Yu Zhang, Jianmin Ji, Houqiang Li, and Yanyong Zhang. Multi-modal 3d object detection in autonomous driving: a survey. International Journal of Computer Vision, 131(8):2122–2152, 2023.

---

> > ### Comment · Reviewer_7Q7d · 2024-11-26
> >
> > Thank you for your response. I keep my original rating of 6.

---

> > > ### Author Response · Authors · 2024-11-29
> > > **Experimental evidence on the applicability of our method to energy/force field prediction tasks (part 1)**
> > >
> > > We sincerely appreciate your thoughtful review and valuable feedback on our manuscript. Your comments have been instrumental in helping us refine and improve our work. Thank you for your time and effort in carefully evaluating our research and for providing insights that have greatly contributed to its development.
> > >
> > > **Regarding your question, "Could the authors provide an example of how this model might be applied to another prediction task?", we now have experimental evidence to support this point**. Specifically:
> > >
> > > We validate the effectiveness of the proposed TraceGrad method on the energy/force field prediction task. Due to the limited time during the discussion period, we conducted experiments on only one dataset, MD17-aspirin [1, 2, 3], as a representative example. We use the same setup of this dataset as Liao and Smidt [4], and a brief introduction is included in the following Table:
> > >
> > > **Table 1: Statistics of MD17-aspirin dataset.**
> > > | Training Samples | Validation Samples | Testing Samples | Formula       | Number of Atoms |
> > > |------------------|--------------------|-----------------|---------------|-----------------|
> > > | 950              | 50                 | 210,762         | C₉H₈O₄        | 21              |
> > >
> > > Note that in this task, the regression target $ E $ (energy) is an SO(3)-invariant quantity ($ l=0 $), while $\mathbf{F}$ (force) is an SO(3)-equivariant quantity ($ l=1 $). The model typically learns SO(3)-equivariant features $\mathbf{f}$, which are then transformed into SO(3)-invariant features, from which $E$ is regressed.
> > >  Subsequently, the force field at a given position is obtained by differentiating $ E $ with respect to the atomic coordinates: $\mathbf{F}_i = -\frac{\partial E}{\partial \mathbf{r}_i}$, where $\mathbf{r}_i$ is the position vector of the $ i $-th atom. This approach ensures energy conservation. Given the specificity of this task, we integrate the baseline model, namely Equiformer [4] with our proposed TraceGrad method as follows:
> > >
> > >
> > > First, we use the SO(3)-equivariant features $\mathbf{f}$ encoded by the baseline model Equiformer as input, and construct SO(3)-invariant non-linear features $z$ according to our method (Sections 4 and 5 of our paper). We then use the trace quantity $\mathbf{T}$ to supervise the learning of $z$. Given $\mathbf{F}$ as a column vector with $l=1$, here $\mathbf{T}$ simplifies to $\mathbf{T} = \mathbf{F}^T \cdot \mathbf{F}$. From $z$, we induce the SO(3)-equivariant features $\mathbf{v}$ with more non-linearity, which are then fed back into the baseline model for the subsequent encoding and decoding phases, where $E$ is regressed and finally $\mathbf{F}$ is constructed from the gradients of $E$.
> > >
> > > We train Equiformer+TraceGrad under the same training conditions as the original paper of Equiformer. The optimizer used is Adam, and a cosine learning rate scheduler with linear warmup is employed, where the warmup epochs set as 10. The maximum learning rate is set to $5 \times 10^{-4}$, and the batch size is 8. The total number of training epochs is 1,500. The weight decay parameter is set to $1 \times 10^{-6}$. The weight for the energy loss is set to 1, while the weight for the force loss is set to 80. The experimental results are listed in the table below, where the results for Equiformer are taken from the original paper.
> > >
> > > **Table 1: MAE results of Equiformer and Equiformer+TraceGrad methods on energy and force prediction, evaluated on the MD17-aspirin dataset. $l_{max}$ corresponds to the maximum degree of features used. MAE of energy and force are in units of meV and meV/Å, respectively.**
> > > | **Method**                       | **Energy (meV)** | **Force (meV/Å)** |
> > > |----------------------------------|------------------|-------------------|
> > > | Equiformer ($l_{max}=2$)         | 5.3              | 7.2               |
> > > | Equiformer+TraceGrad ($l_{max}=2$) | **5.06**         | **5.65**          |
> > >
> > > (continued in part2)

---

> ### Author Response · Authors · 2024-11-29
> **Experimental evidence on the applicability of our method to energy/force field prediction tasks (part 2)**
>
> (continued from part 1)
>
> The experimental results show that our TraceGrad method improves the prediction accuracy for both energy and force. Particularly, the accuracy improvement for force, i.e., the SO(3)-equivariant regression target, is especially significant, as the new supervision signal ($\mathbf{T} = \mathbf{F}^T \cdot \mathbf{F}$)  we introduced is designed targeting force. Nevertheless, it is very interesting that even for the energy, i.e., the SO(3)-invariant quantity, TraceGrad also leads to an accuracy improvement. This is because both energy and force predictions are based on the SO(3)-equivariant features of the network, and our approach enhances the non-linear expressiveness of these SO(3)-equivariant features, thus improving the accuracy of the induced physical quantities. These results show that our method can be effectively transferred to energy/force prediction, and its effectiveness is not limited to the prediction of electronic-structure Hamiltonians and their downstream physical quantities.
>
>
> [1] Stefan Chmiela, Alexandre Tkatchenko, Huziel E. Sauceda, Igor Poltavsky, Kristof T. Schütt, and Klaus-Robert Müller. Machine learning of accurate energy-conserving molecular force fields. Science Advances, 3(5):e1603015, 2017. doi: 10.1126/sciadv.1603015.
>
> [2] Kristof T. Schütt, Farhad Arbabzadah, Stefan Chmiela, Klaus R. Müller, and Alexandre Tkatchenko.Quantum-chemical insights from deep tensor neural networks. Nature Communications, 8(1), jan 2017. doi: 10.1038/ncomms13890.
>
> [3] Stefan Chmiela, Huziel E. Sauceda, Klaus-Robert Müller, and Alexandre Tkatchenko. Towards exact molecular dynamics simulations with machine-learned force fields. Nature Communications, 9(1), sep 2018. doi: 10.1038/s41467-018-06169-2
>
> [4] Yi-Lun Liao and Tess E. Smidt. Equiformer: Equivariant graph attention transformer for 3d atomistic graphs. In ICLR, 2023.
>
> **These new results and analysis will be added to our paper after the paper can be revised in the OpenReview system.**

---

### Official Review · Reviewer_G9tS · 2024-10-26

**Soundness:** 2
**Presentation:** 1
**Contribution:** 2
**Rating:** 6
**Confidence:** 3

**Summary:**

The authors present a novel architecture and loss function designed for learning targets in SO(3) symmetry. Their focus is on the challenge of predicting the Hamiltonian within Kohn-Sham Density Functional Theory (DFT). Their contribution is twofold: First, they suggest fitting the Forbenius norm of the target alongside the direct fitting of the equivariant targets. Second, they propose a method for differentiating the predicted Forbenius norm of the Hamiltonian, utilizing invariant representations to produce equivariant embeddings that capture non-linear dependencies within SO(3)-equivariant embeddings.

**Strengths:**

* The proposed TraceGrad method shows significant improvements over previous approaches.
* The use of differentiation to achieve higher-order SO(3)-equivariant embeddings is a novel concept, as it has not been extensively explored before, apart from its application to atomic forces.

**Weaknesses:**

* The paper suffers from unclear writing, particularly in the abstract and introduction, which fail to clearly convey the contributions and include overly verbose sections (e.g., lines 66-72).
* Although the paper claims to present a general framework for SO(3) equivariance, it is heavily focused on Hamiltonian prediction, which limits its broader applicability.
* The method is inherently designed to learn conservative vector fields on the embeddings, a point that is not addressed but could significantly influence its empirical performance.
* l.57-61, SO(3)-equivariant models include non-linearities through gating or activating the norm of a vector.
* l.129,130, "Nevertheless, multiplying SO(3)-equivariant features with linear coefficients may not fundamentally improve their non-linear expressiveness." is a claim without evidence.
* l.43, In DFT, the Hamiltonian is not predicted but computed or solved for.
* l.46, Hamiltonian prediction cannot work in O(N) given that the matrix itself is O(N²).

**Questions:**

* Are the models in the experiments adjusted for size and compute time? The extra computations through TraceGrad could equally be spent on larger networks, how does this compare?

---

> ### Author Response · Authors · 2024-11-25
> **Response to Reviewer 3**
>
> Thank you very much for your valuable and insightful feedback. We are grateful for the time and effort you have dedicated to reviewing our paper. Your suggestions have been crucial in enhancing the depth and clarity of our work, and we have carefully addressed each of your comments in our revisions. We sincerely hope that the changes and explanations we provided fulfill your expectations. Below, we offer a detailed, point-by-point response to your feedback and summarize the corresponding modifications made in the manuscript.

---

> ### Author Response · Authors · 2024-11-25
> **Response to Weakness 1**
>
> **Comment from the reviewer**: The paper suffers from unclear writing, particularly in the abstract and introduction, which fail to clearly convey the contributions and include overly verbose sections (e.g., lines 66-72).
>
> **Response**:
>
> We appreciate the reviewer's suggestion. We have streamlined the original text in lines 66-72 (now condensed to lines 73-76 in the revised paper) to better highlight our contributions. We also aim to further revise the abstract and introduction to enhance readability. If you have any specific suggestions, we would be grateful and will certainly follow them.

---

> ### Author Response · Authors · 2024-11-25
> **Response to Weakness 2**
>
> **Comment from the reviewer**: Although the paper claims to present a general framework for SO(3) equivariance, it is heavily focused on Hamiltonian prediction, which limits its broader applicability.
>
> **Response**:
>
> Thank you for this valuable comment. We acknowledge that the current focus on Hamiltonian prediction might give the impression of limited applicability. However, we would like to emphasize the following:
>
> First, the electronic Hamiltonian is a cornerstone of quantum mechanics and condensed matter physics. Its accurate prediction is critical for a wide range of applications, including material design, chemical reaction simulations, and quantum technologies, etc. Therefore, Hamiltonian prediction is itself a task of fundamental importance, deserving standalone consideration due to its central role in describing and understanding quantum systems.
>
> Second, the current methodology extends beyond predicting electronic-structure Hamiltonians and is capable of predicting a wide range of physical quantities and properties that exhibit equivariance, such as force constant matrices, Born effective charges, and more. Furthermore, as our approach provides an effective framework for equivariant learning in the deep learning paradigm, the methodology is not limited to physical research and may also find applications in fields such as robotics, autonomous vehicles, and motion tracking systems.
> In these areas, data is typically represented as 3D point clouds, which is conceptually similar to describing the 3D structure of atomic systems using atomic point clouds. This similarity facilitates the transferability of our method to these domains.
> For example, in vision tasks such as autonomous vehicles [1], the relative positions of cameras and objects are not fixed, causing the sampled 3D point clouds to undergo coordinate transformations, with rotation being a common example. Given the safety-critical nature of these tasks, ensuring the reliability and robustness of pattern recognition systems are of utmost importance.  Consequently, there is a significant demand for systems that are robust to coordinate transformations of 3D point clouds.
> The mainstream approach has been to approximate SO(3)-equivariance through data augmentation. However, this method does not ensure absolute reliability or safety. Our work demonstrates significant potential for constructing deep models with high generalization performance, grounded in strict SO(3)-equivariance, and could revolutionize techniques in these fields.
> Therefore, our next step may involve applying our method to the autonomous vehicles domain for 3D point cloud object segmentation and recognition, with the goal of achieving more robust recognition results.
> Specifically, in the task of 3D point cloud object recognition, the position vectors of the corner points of 3D bounding boxes relative to their center point are SO(3)-equivariant quantities, corresponding to $\mathbf{Q}$ in this work, while their magnitudes are SO(3)-invariant quantities, corresponding to $\mathbf{T}$ in this work. These can serve as regression targets and supervision signals for the SO(3)-equivariant and SO(3)-invariant branches of our method, respectively, enabling the learning of informative SO(3)-equivariant non-linear features for regressing 3D bounding boxes. In conclusion, the foundational and pivotal role of this work in advancing future endeavors is evident.
>
> Due to space and focus considerations, we chose to demonstrate  the power of the framework through Hamiltonian prediction as a representative, impactful application.  We have added the discuss on potential extensions of the framework  in Appendix M of the revised paper. We hope this addresses the reviewer’s concerns.
>
> [1] Yingjie Wang, Qiuyu Mao, Hanqi Zhu, Jiajun Deng, Yu Zhang, Jianmin Ji, Houqiang Li, and Yanyong Zhang. Multi-modal 3d object detection in autonomous driving: a survey. International Journal of Computer Vision, 131(8):2122–2152, 2023.
>
> ---------------------------------------------------------------------------------------------------------------------------------------------------------------------------
>
> **Notes for the Area Chair**: Regarding this issue, the reviewer has provided us with further suggestions during subsequent discussions. Following the reviewer's suggestions, we have revised our paper, making adjustments to the claims in the paper and including the new experimental evidence demonstrating the applicability of our method to the additional task, i.e., the energy/force field prediction task, which addresses the reviewer's concerns. For further details, please refer to the discussion below.

---

> ### Author Response · Authors · 2024-11-25
> **Response to Weakness 3**
>
> **Comment from the reviewer**: The method is inherently designed to learn conservative vector fields on the embeddings, a point that is not addressed but could significantly influence its empirical performance.
>
>
> **Response**:
> We would like to thank the reviewer for pointing out such an interesting perspective,
> which is also one of the original motivations for this work. The reviewer is correct that conservative vector fields, such as forces, can be computed by taking derivatives of the energy, which indeed has certain similarities to some aspects of the current work. However, learning the Hamiltonian is a more complex task because there is no analogous quantity, like energy, that can be directly differentiated to calculate the Hamiltonian. In more general tasks, like point cloud detection and segmentation tasks in 3D computer vision,
> finding an energy-force type relationship to compute the target quantities we aim to learn is not always straightforward. The approach developed in this work, however, are applicable in such cases. We construct an SO(3)-invariant physical quantity $\mathbf{T} = \text{tr}(\mathbf{Q} \cdot \mathbf{Q}^\dagger)$ corresponding to the regression target, a general SO(3)-equivariant physical quantity $\mathbf{Q}$, and use the label of $\mathbf{T}$ to supervise the learning of non-linear SO(3)-invariant features $z$ with high expressiveness. We then use $z$ to construct non-linear SO(3)-equivariant features $\mathbf{v}$, providing sufficient expressiveness for regressing $\mathbf{Q}$. This approach is general and does not rely on specific physical quantity relationships, such as energy-force, making it  highly applicable to tasks like Hamiltonian prediction.
> Further explanation of the mechanism can be found below.

---

> ### Author Response · Authors · 2024-11-25
> **Response to Weakness 4 (part 1)**
>
> **Comment from the reviewer**: l.57-61, SO(3)-equivariant models include non-linearities through gating or activating the norm of a vector.
>
> **Response**:
>
> The reviewer is correct that some non-linearities have been included through gating or activating the norm of a vector. However, this process is not as effective as our TraceGrad framework.
>
> The motivation for our framework stems from the observation that, in deep learning, SO(3)-invariant quantities are easier to learn because non-linear mappings can be applied without restrictions, providing sufficient expressiveness to approximate complex mappings.
> In contrast, SO(3)-equivariant features cannot directly utilize non-linear activation functions, which restricts their expressiveness.
> Therefore, we construct SO(3)-invariant features as a bridge, leveraging their strict mathematical relationship with equivariant features to enable effective regression of complex SO(3)-equivariant quantities.
>
> To this end, we construct an SO(3)-invariant quantity $\mathbf{T} = \text{tr}(\mathbf{Q} \cdot \mathbf{Q}^\dagger)$, whose regression target is the SO(3)-equivariant quantity $\mathbf{Q}$. The two are related by $\mathbf{Q} = \frac{\partial \mathbf{T}}{\partial \text{Conj}(\mathbf{Q})}$, where $\text{Conj}(\cdot)$ denotes the complex conjugate. This shows that $\mathbf{Q}$ can be derived via a differentiable mapping from the invariant space to the equivariant space, which inherently imposes a strong constraint on the relationships between the components of $\mathbf{Q}$.
> This mechanism is universal and extends beyond labels to representation learning in neural networks.
>
> We propose a construction of SO(3)-equivariant features: $\mathbf{v} = \frac{\partial z}{\partial \mathbf{f}}$, where $\mathbf{v}$ (regressing $\mathbf{Q}$) and $z$ (regressing $\mathbf{T}$) reflect the partial derivative relationship between $\mathbf{Q}$ and $\mathbf{T}$. Compared to the current gated activation mechanism which can be expressed as $\mathbf{v} = z \cdot \mathbf{f}$, this approach imposes stronger physical constraints on the relationships between the components of the equivariant features and enables effective joint learning of $z$ and $\mathbf{v}$, supervised by $\mathbf{T}$ and $\mathbf{Q}$.
>
> Notably, while $\mathbf{Q} = \frac{\partial \mathbf{T}}{\partial \text{Conj}(\mathbf{Q})}$ involves a derivative with respect to the complex conjugate of $\mathbf{Q}$, and $\mathbf{v} = \frac{\partial z}{\partial \mathbf{f}}$ involves a derivative with respect to $\mathbf{f}$ (since $\mathbf{v}$ is unknown), both are consistent at a higher abstraction level. They show that an SO(3)-equivariant quantity (or feature) can be derived from an associated SO(3)-invariant quantity (or feature) through a derivative relationship, whether for $\mathbf{T} = \text{tr}(\mathbf{Q} \cdot \mathbf{Q}^\dagger)$ or $z = s_{\text{nonlin}}(\text{CGDecomp}(\mathbf{f} \otimes \mathbf{f}, 0))$. In summary, our mechanism leverages stronger non-linearity via invariant functions and enforces strict equivariance with gradient-based physical constraints that regulate the relationships between components, making it more effective than the gated mechanism.
>
> **The above theoretical analysis has been added in Remark 1 of the revised paper.**
>
> (Continued in Part 2)

---

> > ### Author Response · Authors · 2024-11-25
> > **Response to Weakness 4 (part 2)**
> >
> > (Continued from Part 1)
> >
> > To empirically verify  whether our proposed gradient-based mechanism outperforms the gated activation mechanism, we conducted a study comparing the two approaches, **as detailed in Appendix H of the revised paper**. Due to time constraints during the rebuttal phase,  our experiments focused on the Bilayer Graphene ($BG$) and Bilayer Bismuthene ($BB$) databases. The study evaluates the accuracy performance of the following four experimental setups:
> >
> >
> > - DeepH-E3: the baseline model.
> >
> > - DeepH-E3+Grad: this setup implements  half part of our method. Specifically, it incorporates our SO(3)-invariant encoder branch as well as the **grad**ient-induced operator to deliver non-linear expressiveness from SO(3)-invariant features to encode SO(3)-equivariant features. As for ablation study, this setup continues to use the single-task training pipeline of DeepH-E3, supervised  only with the Hamiltonian label without joint supervised training through the **trace** of Hamiltonian.
> >
> > - DeepH-E3+Gate: This variant modifies the experimental setup from DeepH-E3+Grad by replacing the gradient mechanism, which constructs equivariant features as $\mathbf{v} = \frac{\partial z}{\partial \mathbf{f}}$, with a gated activation mechanism, constructing equivariant features as $\mathbf{v} = z \cdot \mathbf{f}$. All other aspects remain the same.
> >
> > - DeepH-E3+TraceGrad: this is a complete implementation of our framework extending beyond of the architecture and training pipeline of DeepH-E3, at the label level, we introduce the **trace** quantity  to guide the learning of SO(3)-invariant features; Meanwhile, at the representation level, we leverage the **grad**ient operator to yield SO(3)-equivariant non-linear  features for Hamiltonian prediction.
> >
> > - DeepH-E3+TraceGate: This variant modifies the experimental setup from DeepH-E3+TraceGrad by replacing the gradient mechanism, which constructs equivariant features as $\mathbf{v} = \frac{\partial z}{\partial \mathbf{f}}$, with a gated activation mechanism, constructing equivariant features as $\mathbf{v} = z \cdot \mathbf{f}$. All other aspects remain the same.
> >
> > (Continued in Part 3)

---

> > > ### Author Response · Authors · 2024-11-25
> > > **Response to Weakness 4 (part 3)**
> > >
> > > (Continued from Part 2)
> > >
> > > Experimental results are recorded in the following Table:
> > >
> > > **Table**: MAE results (meV) on the Bilayer Graphene ($BG$) and Bilayer Bismuthene ($BB$) databases. The superscripts $nt$ and $t$ respectively denote the non-twisted and twisted subsets. $\downarrow$ indicates that lower values of the metrics correspond to better accuracy.
> > > | **Methods**              | **$BG^{nt}$**                    |                              |                              | **$BG^{t}$**                    |                              |                              |
> > > |--------------------------|----------------------------------|------------------------------|------------------------------|---------------------------------|------------------------------|------------------------------|
> > > |                          | **$MAE^H_{all}$**                | **$MAE^{H}_{cha\_s}$**        | **$MAE^{H}_{cha\_b}$**        | **$MAE^H_{all}$**               | **$MAE^{H}_{cha\_s}$**        | **$MAE^{H}_{cha\_b}$**        |
> > > | DeepH-E3 (Baseline)      | 0.389                            | 0.453                        | 0.644                        | 0.264                           | 0.429                        | 0.609                        |
> > > | DeepH-E3+Grad            | 0.320                            | 0.356                        | 0.511                        | 0.222                           | 0.389                        | 0.446                        |
> > > | DeepH-E3+Gate            | 0.368                            | 0.441                        | 0.626                        | 0.241                           | 0.405                        | 0.602                        |
> > > | DeepH-E3+TraceGrad       | **0.291**                        | **0.323**                    | **0.430**                    | **0.198**                       | **0.372**                    | **0.406**                    |
> > > | DeepH-E3+TraceGate       | 0.354                            | 0.403                        | 0.580                        | 0.239                           | 0.388                        | 0.464                        |
> > > | **Methods**              | **$BB^{nt}$**                    |                              |                              | **$BB^{t}$**                    |                              |                              |
> > > |                          | **$MAE^H_{all}$**                | **$MAE^{H}_{cha\_s}$**        | **$MAE^{H}_{cha\_b}$**        | **$MAE^H_{all}$**               | **$MAE^{H}_{cha\_s}$**        | **$MAE^{H}_{cha\_b}$**        |
> > > | DeepH-E3 (Baseline)      | 0.274                            | 0.304                        | 1.042                        | 0.468                           | 0.602                        | 2.399                        |
> > > | DeepH-E3+Grad            | 0.243                            | 0.272                        | 0.824                        | 0.406                           | 0.542                        | 1.431                        |
> > > | DeepH-E3+Gate            | 0.268                            | 0.301                        | 0.991                        | 0.450                           | 0.593                        | 2.276                        |
> > > | DeepH-E3+TraceGrad       | **0.226**                        | **0.256**                    | **0.740**                    | **0.384**                       | **0.503**                    | **1.284**                    |
> > > | DeepH-E3+TraceGate       | 0.252                            | 0.279                        | 0.908                        | 0.417                           | 0.561                        | 1.740                        |
> > >
> > >
> > > From these results, it is evident that both DeepH-E3+Gate and DeepH-E3+TraceGate show improvement over the baseline DeepH-E3, as these configurations introduce an additional neural network module $s_{nonlin}(\cdot)$ in learning the feature $z$, which increases model capacity. Moreover, DeepH-E3+TraceGate also incorporates our proposed trace supervision signal, which guides the learning of SO(3)-invariant features. However, their performance falls short of DeepH-E3+Grad and DeepH-E3+TraceGrad, respectively, indicating that the gated activation mechanism may not fully capture the system's non-linearity patterns, whereas the proposed gradient-based mechanism demonstrates stronger generalization performance and may be a better choice in terms of expressive capability.

---

> ### Author Response · Authors · 2024-11-25
> **Response to Weakness 5**
>
> **Comment from the reviewer**: l.129,130, "Nevertheless, multiplying SO(3)-equivariant features with linear coefficients may not fundamentally improve their non-linear expressiveness." is a claim without evidence.
>
> **Response**:
>
> Although the experiments above have demonstrated the superiority of our approach over the gated activation mechanism, we have taken the reviewer’s feedback into full consideration. We have withdrawn our original claim on the gated activation mechanism that "multiplying SO(3)-equivariant features with linear coefficients may not fundamentally improve their non-linear expressiveness" and replaced it with a more precise statement: "Nevertheless, as demonstrated in our numerical study in Appendix H, multiplying SO(3)-equivariant features with linear coefficients may not be fully effective in enhancing non-linear expressiveness."

---

> ### Author Response · Authors · 2024-11-25
> **Response to Weakness 6**
>
> **Comment from the reviewer**: l.43, In DFT, the Hamiltonian is not predicted but computed or solved for.
>
> **Response**:
> Thank you for pointing this out. We have revised the word as ``solve" according to the reviewer’s suggestion.

---

> ### Author Response · Authors · 2024-11-25
> **Response to Weakness 7**
>
> **Comment from the reviewer**: l.46, Hamiltonian prediction cannot work in O(N) given that the matrix itself is O(N²).
>
> **Response**:
> Thank you for your question regarding the computational complexity of Hamiltonian prediction. A detailed explanation is provided in Appendix I of the revised paper, titled ``Theoretical Analysis on Computational Complexity", which is outlined below:
>
> The total number of non-zero Hamiltonian matrix elements to be calculated is proportional to the number of atomic pairs in the system, with a complexity of $\mathcal{O}(N\overline{E})$, where: $N$ is the total number of atoms in the system and $\overline{E}$ is the average number of neighboring atoms within the cutoff radius for each atom. The atomic orbital basis set we use has a finite range, meaning that Hamiltonian matrix elements are zero beyond a certain distance. In small systems, where all atoms fall within the cutoff radius of each other, $\overline{E}$ is propotional to $N$, and the total number of nonzero elements is proportional to $N^2$. However, for sufficiently large systems, the finite range of the Hamiltonian ensures that $\overline{E}$ remains constant and does not grow with $N$. Consequently, for large atomic systems, the total number of non-zero matrix elements simplifies to $\mathcal{O}(N)$.
>
> The baseline models we select, whether DeepH-E3 or QHNet, are SO(3)-equivariant graph neural network models with efficient information aggregation and message-passing mechanisms. These models cleverly balance the locality of Hamiltonian definitions with the long-range interactions present in the system. As a result, the computational complexity asymptotically scales as $\mathcal{O}(N)$ as $N$ increases, which is consistent with the growth of the scales of Hamiltonian matrices. The proposed TraceGrad method directly updates each SO(3)-equivariant feature of the baseline models, and the computational amount is proportinal to the number of features of the baseline models. Therefore, combining TraceGrad, the computational complexity also scales as $\mathcal{O}(N)$.
>
> Traditional DFT methods require $T$ rounds of iterative diagonalization of $N \times N$ matrices to compute the charge density, which requires calculating all the occupied orbitals. The time complexity of this process is $\mathcal{O}(T N^3)$.
> As $N$ increases, the cubic complexity results in significant computational overhead, making it very challenging to simulate large atomic systems within an acceptable time frame.
>
>  In contrast, our deep learning framework offers an efficient and accurate solution for simulating large atomic systems without the need for multiple rounds of iterations and the computation of all occupied orbitals.
> Since the Hamiltonian matrix is sparse and only a limited number of bands near the Fermi surface are required  for many physical properties, such as electrical transport, optical response, and topological band properties, these bands can be efficiently computed from the Hamiltonian matrix using methods like the shift-invert approach in the ARPACK package, with a computational complexity of $O(N)$.
>
>
>
> [1] Xiaoxun Gong, He Li, Nianlong Zou, Runzhang Xu, Wenhui Duan, and Yong Xu. General framework
> for e(3)-equivariant neural network representation of density functional theory hamiltonian.Nature Communications, 14(1):2848, 2023.
>
> [2] Haiyang Yu, Zhao Xu, Xiaofeng Qian, Xiaoning Qian, and Shuiwang Ji. Efffcient and equivariant graph networks for predicting quantum hamiltonian. In ICML, pp. 40412–40424, 2023b.
>
> [3] Walter Kohn and Lu Jeu Sham. Self-consistent equations including exchange and correlation effects. Physical Review, 140(4A):A1133, 1965

---

> > ### Comment · Reviewer_G9tS · 2024-11-25
> >
> > Thank you for addressing some of my concerns, I increased my score. I will not increase my score higher due to the claims without empirical evidence about the generality of the framework in other domains.

---

> ### Author Response · Authors · 2024-11-26
> **Response to Question 1 (part 1)**
>
> **Comment from the reviewer**: Are the models in the experiments adjusted for size and compute time? The extra computations through TraceGrad could equally be spent on larger networks, how does this compare?
>
> **Response**:
> The network size of the baseline models, whether DeepH-E3 or QHNet, which are embedded in our method, are exactly the same as their official settings. The implementation details of our experiments are provided in Appendix E.
>
> In response to the reviewer’s concern, in the revised paper’s Appendix J, titled "A Joint Discussion on GPU Time Costs and Performance Gains", we provide a joint comparison of the inference time cost and corresponding accuracy of different models across four databases as representative: Monolayer Graphene ($MG$), Monolayer MoS2 ($MM$), QH9-stable ($QS$), and QH9-dynamic ($QD$). This comparison includes the average inference time  per sample for each model, with the test batch size set as 1 and hardware environment set as Nvidia RTX A6000 GPU in single-task mode without computational sharing with other processes.
>
> For $MG$ and $MM$, we compare among DeepH-E3, DeepH-E3+TraceGrad, and DeepH-E3$^{×2}$, where DeepH-E3$^{×2}$ refers to a model obtained by doubling the number of encoding blocks in DeepH-E3 and training it from scratch until convergence. For $QS$ and $QD$, we compare among QHNet, QHNet+TraceGrad, and QHNet$^{×2}$, where QHNet$^{×2}$ refers to a model obtained by doubling the number of encoding blocks in QHNet and training it from scratch until convergence. The experimental results are documented in the table below:
>
> **Table**: Average inference time per testing sample (abbreviated as Time) in seconds and average MAE performance in meV ($MAE^H_{all}$) on the Monolayer Graphene (MG), Monolayer MoS2 (MM), QH9-stable (QS), and QH9-dynamic (QD) databases. *↓* indicates that lower values of the metrics correspond to better accuracy.
> | **Methods**              | **$MG$**                           |                              |                              | **$MM$**                           |                              |                              |
> |--------------------------|------------------------------------|------------------------------|------------------------------|------------------------------------|------------------------------|------------------------------|
> |                          | **$Time$ (↓)**                     | **$MAE^H_{all}$ (↓)**         |                              | **$Time$ (↓)**                     | **$MAE^H_{all}$ (↓)**         |                              |
> | DeepH-E3 (Baseline)      | 0.247                              | 0.251                         |                              | 0.256                              | 0.406                         |                              |
> | DeepH-E3$^{×2}$     | 0.483                              | 0.244                         |                              | 0.510                              | 0.387                         |                              |
> | DeepH-E3+TraceGrad       | 0.264                              | 0.175                         |                              | 0.274                              | 0.285                         |                              |
> | **Methods**              | **$QS$**                           |                              |                              | **$QD$**                           |                              |                              |
> |                          | **$Time$ (↓)**                     | **$MAE^H_{all}$ (↓)**         |                              | **$Time$ (↓)**                     | **$MAE^H_{all}$ (↓)**         |                              |
> | QHNet (Baseline)         | 0.233                              | 1.962                         |                              | 0.174                              | 4.733                         |                              |
> | QHNet$^{×2}$        | 0.497                              | 1.845                         |                              | 0.385                              | 4.532                         |                              |
> | QHNet+TraceGrad          | 0.248                              | 1.191                         |                              | 0.187                              | 2.819                         |                              |
>
>
>
>
>
>
> (Continued in part 2)

---

> ### Author Response · Authors · 2024-11-26
> **Response to Question 1 (part 2)**
>
> From this Table, we  find that incorporating the TraceGrad method results in only a slight increase in inference time compared to the baseline models. Given the substantial accuracy improvements introduced by the TraceGrad method, this minor increase in computational time is considered acceptable for practical applications. In contrast, simply increasing the depth of DeepH-E3 or QHNet leads to a significant rise in inference time while providing only limited accuracy improvements.
> In contrast, DeepH-E3+TraceGrad and QHNet+TraceGrad exhibit significantly better accuracy performance compared to DeepH-E3$^{×2}$ and QHNet$^{×2}$, respectively. At the same time, the inference times of DeepH-E3+TraceGrad and QHNet+TraceGrad are considerably lower than those of DeepH-E3$^{×2}$ and QHNet$^{×2}$, respectively. These findings highlight the superiority of the TraceGrad method in enhancing model expressiveness and improving accuracy performance while maintaining computational efficiency.
>
> In addition to reporting the GPU inference time of the deep models, we also present the inference times of QHNet and QHNet+TraceGrad on a single thread of an Intel(R) Xeon(R) Gold 6330 @ 2.00GHz CPU in single-task mode in the revised paper’s Appendix K, titled "Acceleration Performance for the Convergence of Traditional DFT Algorithms", which quantifies the acceleration performance of the deep learning methods for DFT computations. In these experiments, the predictions from deep models, e.g., QHNet or QHNet+TraceGrad, are used as initial matrices to accelerate the convergence of traditional DFT algorithms. Experimental results show that while combining TraceGrad introduces only a slight increase from the inference time of QHNet on the CPU, it delivers significant improvements in accelerating the convergence of DFT methods. Notably, the time saved by TraceGrad for DFT calculations  far exceeds the minimal additional time introduced by TraceGrad for the deep model's inference.

---

> ### Author Response · Authors · 2024-11-26
> **Response to the new comments by reviewer G9tS**
>
> **Comment from the reviewer**: I will not increase my score higher due to the claims without empirical evidence about the generality of the framework in other domains.
>
> **Response**:
>
> We sincerely thank the reviewer for the suggestion to validate our method through experiments in a broader range of domains to demonstrate its effectiveness. However, we would like to emphasize that, in recent years, the task of electronic structure Hamiltonian prediction has emerged as a rapidly growing field and is already a sufficiently core, important, and representative research challenge. It has become as significant as other AI for Science fields, such as force field development and molecular structure prediction, etc.. The development of this direction has opened up new possibilities for analyzing extremely large atomic systems and has enabled efficient materials simulation and design, which were previously unimaginable. As a result, this field has become a highly promising and emerging area of research. In recent years, numerous deep learning studies based on this task have been published in Nature sub-journals or top-tier AI conferences due to their significant contributions. For example:
>
> [1] He Zhang, Chang Liu, Zun Wang, Xinran Wei, Siyuan Liu, Nanning Zheng, Bin Shao, and Tie-Yan Liu. Self-Consistency Training for Density-Functional-Theory Hamiltonian Prediction, In ICML, 2024.
>
> [2] Xiaoxun Gong, He Li, Nianlong Zou, Runzhang Xu, Wenhui Duan, and Yong Xu. General framework for e(3)-equivariant neural network representation of density functional theory hamiltonian. Nature Communications, 14(1):2848, 2023.
>
> [3] Haiyang Yu, Meng Liu, Youzhi Luo, Alex Strasser, Xiaofeng Qian, Xiaoning Qian, and Shuiwang Ji. QH9: A quantum hamiltonian prediction benchmark for QM9 molecules. In NeurIPS, 2023a.
>
> [4] Haiyang Yu, Zhao Xu, Xiaofeng Qian, Xiaoning Qian, and Shuiwang Ji. Efffcient and equivariant graph networks for predicting quantum hamiltonian. In ICML, pp. 40412–40424, 2023b.
>
> [5] He Li, Zun Wang, Nianlong Zou, Meng Ye, Runzhang Xu, Xiaoxun Gong, Wenhui Duan, and Yong Xu. Deep-learning density functional theory hamiltonian for efficient ab initio electronic-structure calculation. Nature Computational Science, 2(6):367–377, 2022.
>
>
>
>
>
> **However, we are also willing to humbly accept the reviewers' suggestions. Fortunately, the rebuttal deadline has been extended, which may allow us to conduct additional experiments. We might still have time to validate the effectiveness of our method on the energy/force field prediction task [6,7] for atomic systems, and we will do our best to explore this. May we ask whether you believe that this task could help demonstrate the generality of our method to broader domains, or if you have other suggestions? We sincerely look forward to your feedback.**
>
> [6] Stefan Chmiela, Alexandre Tkatchenko, Huziel E. Sauceda, Igor Poltavsky, Kristof T. Schütt, and Klaus-Robert Müller. Machine learning of accurate energy-conserving molecular force fields. Science Advances, 3(5):e1603015, 2017. doi: 10.1126/sciadv.1603015.
>
> [7] Yi-Lun Liao and Tess E. Smidt. Equiformer: Equivariant graph attention transformer for 3d atomistic graphs. In ICLR, 2023.

---

> > ### Comment · Reviewer_G9tS · 2024-11-26
> >
> > I appreciate the thorough response and do not want to downplay the importance of Hamiltonian prediction. My issues were resolved around the unsupported claim of the paper that it was a general solution while being very specific to Hamiltonian prediction. Either the writing must reflect the single application and remove the general claim, or empirical evidence must be presented. To do the latter, I would like to see the author's suggestion for testing force fields to support their claims.

---

> ### Author Response · Authors · 2024-11-27
> **Response to the comments by reviewer G9tS**
>
> Thank you for your thoughtful feedback. We sincerely appreciate your perspective and the opportunity to clarify our work.
>
> We acknowledge that some of our claims regarding the general applicability of our method in other domains were not sufficiently supported by experimental validation. Since the end of the discussion period extends to December 2nd, we are attempting to validate the effectiveness of our approach on a representative subset of energy/force field prediction benchmarks to provide further experimental evidence of our method's applicability in domains beyond Hamiltonian prediction. Yet, considering no revisions to the paper are allowed after November 26th in this phase, we are unable  to incorporate experiments related to  energy/force predictions into the revised paper, and we will present the results in the following response area after our added experiments are done.
>
> Nevertheless, we follow the reviewers' suggestions and have revised our manuscript to focus on the demonstrated strength of our method in the specific task of electronic-structure Hamiltonian prediction (highlighted in line 88 of the revised paper), where it achieves state-of-the-art performance. While our theoretical framework detailed in Theorem 1 and Theorem 2, incorporating concepts such as SO(3)-equivariant quantities $\mathbf{Q}$, SO(3)-invariant quantities $\mathbf{T}$, SO(3)-equivariant features $\mathbf{f}$ and $\mathbf{v}$, and SO(3)-invariant features $z$, along with their mathematical relationships (proven in Appendix C), is general in nature and not limited to specific physical quantities, we agree that the empirical effectiveness of our method beyond predicting Hamiltonian and its downstream quantities remains to be validated. We have explicitly stated this in the Future Work Section (lines 1593 to 1601 in the revised paper).
>
> In the revised manuscript, we have withdrawn claims about the general applicability of our method, removing the terms "general" and "unified" from lines 99, 255, and 367.
> We revised Appendix M, titled "FUTURE WORK'', to state that our methods can potentially be extended to other domains. Additionally, in lines 1622 to 1624 of the revised paper, we explicitly clarified: "To summarize, while our method is theoretically applicable to other tasks, its effectiveness in those areas has yet to be demonstrated. Much work remains to be done.''

---

> ### Author Response · Authors · 2024-11-29
> **Experimental evidence on the applicability of our method to energy/force field prediction tasks （part 1）**
>
> We validate the effectiveness of the proposed TraceGrad method on the energy/force field prediction task. Due to the limited time during the discussion period, we conducted experiments on only one dataset, MD17-aspirin [1, 2, 3], as a representative example. We use the same setup of this dataset as Liao and Smidt [4], and a brief introduction is included in the following Table:
>
> **Table 1: Statistics of MD17-aspirin dataset.**
> | Training Samples | Validation Samples | Testing Samples | Formula       | Number of Atoms |
> |------------------|--------------------|-----------------|---------------|-----------------|
> | 950              | 50                 | 210,762         | C₉H₈O₄        | 21              |
>
> Note that in this task, the regression target $ E $ (energy) is an SO(3)-invariant quantity ($ l=0 $), while $\mathbf{F}$ (force) is an SO(3)-equivariant quantity ($ l=1 $). The model typically learns SO(3)-equivariant features $\mathbf{f}$, which are then transformed into SO(3)-invariant features, from which $E$ is regressed.
>  Subsequently, the force field at a given position is obtained by differentiating $ E $ with respect to the atomic coordinates: $\mathbf{F}_i = -\frac{\partial E}{\partial \mathbf{r}_i}$, where $\mathbf{r}_i$ is the position vector of the $ i $-th atom. This approach ensures energy conservation. Given the specificity of this task, we integrate the baseline model, namely Equiformer [4] with our proposed TraceGrad method as follows:
>
>
> First, we use the SO(3)-equivariant features $\mathbf{f}$ encoded by the baseline model Equiformer as input, and construct SO(3)-invariant non-linear features $z$ according to our method (Sections 4 and 5 of our paper). We then use the trace quantity $\mathbf{T}$ to supervise the learning of $z$. Given $\mathbf{F}$ as a column vector with $l=1$, here $\mathbf{T}$ simplifies to $\mathbf{T} = \mathbf{F}^T \cdot \mathbf{F}$. From $z$, we induce the SO(3)-equivariant features $\mathbf{v}$ with more non-linearity, which are then fed back into the baseline model for the subsequent encoding and decoding phases, where $E$ is regressed and finally $\mathbf{F}$ is constructed from the gradients of $E$.
>
> We train Equiformer+TraceGrad under the same training conditions as the original paper of Equiformer. The optimizer used is Adam, and a cosine learning rate scheduler with linear warmup is employed, where the warmup epochs set as 10. The maximum learning rate is set to $5 \times 10^{-4}$, and the batch size is 8. The total number of training epochs is 1,500. The weight decay parameter is set to $1 \times 10^{-6}$. The weight for the energy loss is set to 1, while the weight for the force loss is set to 80. The experimental results are listed in the table below, where the results for Equiformer are taken from the original paper.
>
> **Table 1: MAE results of Equiformer and Equiformer+TraceGrad methods on energy and force prediction, evaluated on the MD17-aspirin dataset. $l_{max}$ corresponds to the maximum degree of features used. MAE of energy and force are in units of meV and meV/Å, respectively.**
> | **Method**                       | **Energy (meV)** | **Force (meV/Å)** |
> |----------------------------------|------------------|-------------------|
> | Equiformer ($l_{max}=2$)         | 5.3              | 7.2               |
> | Equiformer+TraceGrad ($l_{max}=2$) | **5.06**         | **5.65**          |
>
> (continued in part 2)

---

> ### Author Response · Authors · 2024-11-29
> **Experimental evidence on the applicability of our method to energy/force field prediction tasks （part 2）**
>
> (continued from part 1)
>
> The experimental results show that our TraceGrad method improves the prediction accuracy for both energy and force. Particularly, the accuracy improvement for force, i.e., the SO(3)-equivariant regression target, is especially significant, as the new supervision signal ($\mathbf{T} = \mathbf{F}^T \cdot \mathbf{F}$)  we introduced is designed targeting force. Nevertheless, it is very interesting that even for the energy, i.e., the SO(3)-invariant quantity, TraceGrad also leads to an accuracy improvement. This is because both energy and force predictions are based on the SO(3)-equivariant features of the network, and our approach enhances the non-linear expressiveness of these SO(3)-equivariant features, thus improving the accuracy of the induced physical quantities. These results show that our method can be effectively transferred to energy/force prediction, and its effectiveness is not limited to the prediction of electronic-structure Hamiltonians and their downstream physical quantities.
>
> [1] Stefan Chmiela, Alexandre Tkatchenko, Huziel E. Sauceda, Igor Poltavsky, Kristof T. Schütt, and Klaus-Robert Müller. Machine learning of accurate energy-conserving molecular force fields. Science Advances, 3(5):e1603015, 2017. doi: 10.1126/sciadv.1603015.
>
> [2] Kristof T. Schütt, Farhad Arbabzadah, Stefan Chmiela, Klaus R. Müller, and Alexandre Tkatchenko.Quantum-chemical insights from deep tensor neural networks. Nature Communications, 8(1), jan 2017. doi: 10.1038/ncomms13890.
>
> [3] Stefan Chmiela, Huziel E. Sauceda, Klaus-Robert Müller, and Alexandre Tkatchenko. Towards exact molecular dynamics simulations with machine-learned force fields. Nature Communications, 9(1), sep 2018. doi: 10.1038/s41467-018-06169-2
>
> [4] Yi-Lun Liao and Tess E. Smidt. Equiformer: Equivariant graph attention transformer for 3d atomistic graphs. In ICLR, 2023.
>
> **These new results and analysis will be added to our paper after the paper can be revised in the OpenReview system.**

---

> ### Author Response · Authors · 2024-11-30
>
> We believe that our response, including the adjustments made to the claims in the paper, and the experimental evidence on the applicability of our method to energy/force field prediction task, addresses the reviewer's concerns. **We look forward to your feedback**. Should you have any further questions or concerns, please do not hesitate to let us know. We would also like to express our gratitude for the time and effort you have dedicated to reviewing our work.

---

> > ### Comment · Reviewer_G9tS · 2024-12-01
> >
> > Thanks a lot for the additional experiment, I will increase my score to 6.

---

> > > ### Author Response · Authors · 2024-12-02
> > >
> > > Thank you so much for taking the time to review our response and reassess our manuscript. Your insightful comments have played a crucial role in helping us refine and enhance the quality of our work.

---

### Official Review · Reviewer_7K3e · 2024-11-02

**Soundness:** 3
**Presentation:** 3
**Contribution:** 3
**Rating:** 6
**Confidence:** 5

**Summary:**

This paper addresses the challenge of incorporating non-linear expressiveness into SO(3)-equivariant deep learning frameworks for physical systems. The authors propose a theoretical framework that constructs SO(3)-invariant quantities tr(Q·Q†) from SO(3)-equivariant targets and utilizes gradient operations to induce SO(3)-equivariant features while maintaining non-linear expressiveness. The method is applied to electronic-structure Hamiltonian prediction tasks and evaluated on eight benchmark databases, showing improvements in prediction accuracy and downstream physical quantities. The method demonstrates generalization ability across various scenarios including thermal motions and bilayer twists, and aids in accelerating traditional DFT calculations through better initialization. The main contribution is providing a framework that attempts to combine strict SO(3)-equivariance with non-linear expressiveness in deep learning models for physical systems.

**Strengths:**

Originality:
- Proposes a novel perspective on bridging SO(3)-equivariance and non-linear expressiveness through invariant quantities and gradient operations
- Creates a theoretical connection between invariant and equivariant representations for l>1.
- Introduces a gradient-based mechanism for inducing equivariant features while preserving non-linear information

Quality:
- Provides rigorous mathematical proofs for the theoretical framework
- Conducts comprehensive experiments across several different benchmark databases

Clarity:
- Well-structured presentation of the theoretical framework
- Clear mathematical derivations with detailed proofs in appendices
- Thorough experimental analysis with multiple evaluation metrics

**Weaknesses:**

While proposing a new method, it fails to provide detailed comparisons of parameter counts, memory consumption, training time, and convergence behavior between the baseline and proposed models. The claimed superiority of gradient-based transfer over traditional gating activation mechanisms is not convincingly demonstrated through empirical evidence. Moreover, the practical significance of the accuracy improvements is not well justified - the paper does not discuss what constitutes a meaningful accuracy threshold for Hamiltonian prediction tasks, nor does it adequately analyze the trade-offs between these moderate accuracy improvements and their associated computational costs. These omissions make it difficult to fully evaluate the practical value and efficiency of the proposed method relative to existing approaches. A comprehensive analysis of these aspects would significantly strengthen the paper's contributions and help readers better understand the method's practical advantages and limitations.

**Questions:**

1. The paper proposes using gradient operation (∂z/∂f) to transfer nonlinearity from l=0 features to equivariant features, where z = S_nonlin(u) and u = CGDecomp(f⊗f, 0). However, similar to the gated activation mechanisms, this method still essentially applies nonlinearity only to l=0 components. The fundamental advantage of gradient-based transfer over the gated activation mechanism remains unclear - could the authors provide theoretical insights on why gradient operations might be more effective than gated activation for transferring nonlinear information from l=0 to higher-degree features?

2. The paper lacks crucial implementation details comparing the baseline and proposed method, such as parameter counts, training epochs, convergence curves, and computational overhead. More importantly, the reported improvements are moderate (e.g., MAE reduction from 0.274 to 0.226 on BB dataset) - such accuracy gains could potentially be achieved through increasing training epoch, learning rate adjustment, or other hyperparameter tuning of the baseline models. To demonstrate the superiority of the proposed method, one needs to compare the upper limits of model expressiveness after controlling for various training factors (number of epochs, hyperparameters, optimization strategies, random seeds, etc.). The current empirical evidence does not adequately address this comparison.

3. For electronic-structure Hamiltonian prediction tasks, what is the practically meaningful accuracy threshold? If, for instance, 1 meV accuracy is sufficient for most practical applications, how do we justify the value of further accuracy improvements (e.g., from 0.274 to 0.226 meV on BB dataset)? Could the authors discuss the practical significance of such fine-grained improvements in the context of real-world applications, such as materials design and DFT calculations?

4. In line 1116, The paper compares the O(N) complexity of Hamiltonian prediction with the O(N³) complexity of traditional DFT methods. However, this comparison seems unfair: the O(N³) complexity in traditional DFT includes solving the Kohn-Sham equations, while the O(N) complexity here only covers Hamiltonian prediction. To obtain physical properties from the predicted Hamiltonian, one still needs O(N³) operations for diagonalization. Could the authors clarify this comparison and provide a more comprehensive complexity analysis that includes all steps necessary for obtaining the same physical properties in both approaches?

---

> ### Author Response · Authors · 2024-11-25
> **Response to Reviewer 2**
>
> Thank you sincerely for your thoughtful and detailed feedback. We greatly appreciate the time and effort you dedicated to reviewing our work. Your insights have been invaluable in enhancing both the clarity and precision of our paper. We hope the revisions and responses we have provided meet your expectations. Below, we present a point-by-point response to each of your comments, along with an outline of the corresponding revisions.

---

> > ### Comment · Reviewer_7K3e · 2024-11-25
> >
> > Thanks for addressing my questions and I will increase the soundness score.

---

> > > ### Author Response · Authors · 2024-11-29
> > >
> > > Thank you very much for taking the time to review our response and for reassessing our manuscript. We greatly value your constructive feedback, which has played a crucial role in enhancing the quality of our work. We deeply appreciate your recognition of the improvements we made and your decision to adjust the evaluation on soundness score. Your thoughtful insights have been incredibly helpful, and we are truly grateful for your support.

---

> ### Author Response · Authors · 2024-11-25
> **Response to Question 1 (part 1)**
>
> **Comment from the reviewer**: The paper proposes using gradient operation (∂z/∂f) to transfer nonlinearity from l=0 features to equivariant features, where z = S_nonlin(u) and u = CGDecomp(f⊗f, 0). However, similar to the gated activation mechanisms, this method still essentially applies nonlinearity only to l=0 components. The fundamental advantage of gradient-based transfer over the gated activation mechanism remains unclear - could the authors provide theoretical insights on why gradient operations might be more effective than gated activation for transferring nonlinear information from l=0 to higher-degree features?
>
> **Response**:
> The reviewer is correct that our proposed gradient-based mechanism directly applies a non-linear mapping to the $l=0$ feature $u$. Yet, this feature $u$ is derived from the input feature $\textbf{f}$ with all $l$ through a direct-product operation and Clebsch-Gordan decomposition, thereby retaining the information of all $l$ from the original $\textbf{f}$. Consequently, the newly constructed non-linear SO(3)-invariant feature $z = s_{nonlin}(u)$ also contains information from $\textbf{f}$ and is encoded through a non-linear mapping. Furthermore, at the label layer, we construct the SO(3)-invariant quantity $\mathbf{T} = \text{tr}(\mathbf{Q} \cdot \mathbf{Q}^\dagger)$ from the SO(3)-equivariant target quantity $\mathbf{Q}$, so that $\mathbf{T}$ also carries information from $\mathbf{Q}$. By using $\mathbf{T}$ as a label to supervise the learning of $z$, we enable it to intrinsically capture the information from the SO(3)-equivariant target.
>
> The motivation for our framework stems from the observation that, in deep learning, SO(3)-invariant quantities are easier to learn because non-linear mappings can be applied without restrictions, providing sufficient expressiveness to approximate complex mappings.
> In contrast, SO(3)-equivariant features cannot directly utilize non-linear activation functions, which restricts their expressiveness.
> Therefore, we construct SO(3)-invariant features as a bridge, leveraging their strict mathematical relationship with equivariant features to enable effective regression of complex SO(3)-equivariant quantities.
>
> To this end, we construct an SO(3)-invariant quantity $\mathbf{T} = \text{tr}(\mathbf{Q} \cdot \mathbf{Q}^\dagger)$, whose regression target is the SO(3)-equivariant quantity $\mathbf{Q}$. The two are related by $\mathbf{Q} = \frac{\partial \mathbf{T}}{\partial \text{Conj}(\mathbf{Q})}$, where $\text{Conj}(\cdot)$ denotes the complex conjugate. This shows that $\mathbf{Q}$ can be derived via a differentiable mapping from the invariant space to the equivariant space, which inherently imposes a strong constraint on the relationships between the components of $\mathbf{Q}$.
> This mechanism is universal and extends beyond labels to representation learning in neural networks.
>
> We propose a construction of SO(3)-equivariant features: $\mathbf{v} = \frac{\partial z}{\partial \mathbf{f}}$, where $\mathbf{v}$ (regressing $\mathbf{Q}$) and $z$ (regressing $\mathbf{T}$) reflect the partial derivative relationship between $\mathbf{Q}$ and $\mathbf{T}$. Compared to the current gated activation mechanism which can be expressed as $\mathbf{v} = z \cdot \mathbf{f}$, this approach imposes stronger physical constraints on the relationships between the components of the equivariant features and enables effective joint learning of $z$ and $\mathbf{v}$, supervised by $\mathbf{T}$ and $\mathbf{Q}$.
>
> Notably, while $\mathbf{Q} = \frac{\partial \mathbf{T}}{\partial \text{Conj}(\mathbf{Q})}$ involves a derivative with respect to the complex conjugate of $\mathbf{Q}$, and $\mathbf{v} = \frac{\partial z}{\partial \mathbf{f}}$ involves a derivative with respect to $\mathbf{f}$ (since $\mathbf{v}$ is unknown), both are consistent at a higher abstraction level. They show that an SO(3)-equivariant quantity (or feature) can be derived from an associated SO(3)-invariant quantity (or feature) through a derivative relationship, whether for $\mathbf{T} = \text{tr}(\mathbf{Q} \cdot \mathbf{Q}^\dagger)$ or $z = s_{\text{nonlin}}(\text{CGDecomp}(\mathbf{f} \otimes \mathbf{f}, 0))$. In summary, our mechanism leverages stronger non-linearity via invariant functions and enforces strict equivariance with gradient-based physical constraints that regulate the relationships between components, making it more effective than the gated mechanism.
>
> **The above theoretical analysis has been added in Remark 1 of the revised paper.**
>
> (Continued in Part 2)

---

> > ### Author Response · Authors · 2024-11-25
> > **Response to Question 1 (part 2)**
> >
> > To empirically verify  whether our proposed gradient-based mechanism outperforms the gated activation mechanism, we conducted a study comparing the two approaches, **as detailed in Appendix H of the revised paper**. Due to time constraints during the rebuttal phase,  our experiments focused on the Bilayer Graphene ($BG$) and Bilayer Bismuthene ($BB$) databases. The study evaluates the accuracy performance of the following four experimental setups:
> >
> >
> > - DeepH-E3: the baseline model.
> >
> > - DeepH-E3+Grad: this setup implements  half part of our method. Specifically, it incorporates our SO(3)-invariant encoder branch as well as the **grad**ient-induced operator to deliver non-linear expressiveness from SO(3)-invariant features to encode SO(3)-equivariant features. As for ablation study, this setup continues to use the single-task training pipeline of DeepH-E3, supervised  only with the Hamiltonian label without joint supervised training through the **trace** of Hamiltonian.
> >
> > - DeepH-E3+Gate: This variant modifies the experimental setup from DeepH-E3+Grad by replacing the gradient mechanism, which constructs equivariant features as $\mathbf{v} = \frac{\partial z}{\partial \mathbf{f}}$, with a gated activation mechanism, constructing equivariant features as $\mathbf{v} = z \cdot \mathbf{f}$. All other aspects remain the same.
> >
> > - DeepH-E3+TraceGrad: this is a complete implementation of our framework extending beyond of the architecture and training pipeline of DeepH-E3, at the label level, we introduce the **trace** quantity  to guide the learning of SO(3)-invariant features; Meanwhile, at the representation level, we leverage the **grad**ient operator to yield SO(3)-equivariant non-linear  features for Hamiltonian prediction.
> >
> > - DeepH-E3+TraceGate: This variant modifies the experimental setup from DeepH-E3+TraceGrad by replacing the gradient mechanism, which constructs equivariant features as $\mathbf{v} = \frac{\partial z}{\partial \mathbf{f}}$, with a gated activation mechanism, constructing equivariant features as $\mathbf{v} = z \cdot \mathbf{f}$. All other aspects remain the same.
> >
> > (Continued in Part 3)

---

> > > ### Author Response · Authors · 2024-11-25
> > > **Response to Question 1 (part 3)**
> > >
> > > (Continued from Part 2)
> > >
> > > Experimental results are recorded in the following Table:
> > >
> > > **Table**: MAE results (meV) on the Bilayer Graphene ($BG$) and Bilayer Bismuthene ($BB$) databases. The superscripts $nt$ and $t$ respectively denote the non-twisted and twisted subsets. $\downarrow$ indicates that lower values of the metrics correspond to better accuracy.
> > > | **Methods**              | **$BG^{nt}$**                    |                              |                              | **$BG^{t}$**                    |                              |                              |
> > > |--------------------------|----------------------------------|------------------------------|------------------------------|---------------------------------|------------------------------|------------------------------|
> > > |                          | **$MAE^H_{all}$**                | **$MAE^{H}_{cha\_s}$**        | **$MAE^{H}_{cha\_b}$**        | **$MAE^H_{all}$**               | **$MAE^{H}_{cha\_s}$**        | **$MAE^{H}_{cha\_b}$**        |
> > > | DeepH-E3 (Baseline)      | 0.389                            | 0.453                        | 0.644                        | 0.264                           | 0.429                        | 0.609                        |
> > > | DeepH-E3+Grad            | 0.320                            | 0.356                        | 0.511                        | 0.222                           | 0.389                        | 0.446                        |
> > > | DeepH-E3+Gate            | 0.368                            | 0.441                        | 0.626                        | 0.241                           | 0.405                        | 0.602                        |
> > > | DeepH-E3+TraceGrad       | **0.291**                        | **0.323**                    | **0.430**                    | **0.198**                       | **0.372**                    | **0.406**                    |
> > > | DeepH-E3+TraceGate       | 0.354                            | 0.403                        | 0.580                        | 0.239                           | 0.388                        | 0.464                        |
> > > | **Methods**              | **$BB^{nt}$**                    |                              |                              | **$BB^{t}$**                    |                              |                              |
> > > |                          | **$MAE^H_{all}$**                | **$MAE^{H}_{cha\_s}$**        | **$MAE^{H}_{cha\_b}$**        | **$MAE^H_{all}$**               | **$MAE^{H}_{cha\_s}$**        | **$MAE^{H}_{cha\_b}$**        |
> > > | DeepH-E3 (Baseline)      | 0.274                            | 0.304                        | 1.042                        | 0.468                           | 0.602                        | 2.399                        |
> > > | DeepH-E3+Grad            | 0.243                            | 0.272                        | 0.824                        | 0.406                           | 0.542                        | 1.431                        |
> > > | DeepH-E3+Gate            | 0.268                            | 0.301                        | 0.991                        | 0.450                           | 0.593                        | 2.276                        |
> > > | DeepH-E3+TraceGrad       | **0.226**                        | **0.256**                    | **0.740**                    | **0.384**                       | **0.503**                    | **1.284**                    |
> > > | DeepH-E3+TraceGate       | 0.252                            | 0.279                        | 0.908                        | 0.417                           | 0.561                        | 1.740                        |
> > >
> > >
> > > From these results, it is evident that both DeepH-E3+Gate and DeepH-E3+TraceGate show improvement over the baseline DeepH-E3, as these configurations introduce an additional neural network module $s_{nonlin}(\cdot)$ in learning the feature $z$, which increases model capacity. Moreover, DeepH-E3+TraceGate also incorporates our proposed trace supervision signal, which guides the learning of SO(3)-invariant features. However, their performance falls short of DeepH-E3+Grad and DeepH-E3+TraceGrad, respectively, indicating that the gated activation mechanism may not fully capture the system's non-linearity patterns, whereas the proposed gradient-based mechanism demonstrates stronger generalization performance and may be a better choice in terms of expressive capability.

---

> ### Author Response · Authors · 2024-11-25
> **Response to Question 2 (part 1)**
>
> **Comment from the reviewer**: The paper lacks crucial implementation details comparing the baseline and proposed method, such as parameter counts, training epochs, convergence curves, and computational overhead. More importantly, the reported improvements are moderate (e.g., MAE reduction from 0.274 to 0.226 on BB dataset) - such accuracy gains could potentially be achieved through increasing training epoch, learning rate adjustment, or other hyperparameter tuning of the baseline models. To demonstrate the superiority of the proposed method, one needs to compare the upper limits of model expressiveness after controlling for various training factors (number of epochs, hyperparameters, optimization strategies, random seeds, etc.). The current empirical evidence does not adequately address this comparison.
>
> **Response**:
> Thanks for the valuable feedback on the implementation details and  computational overhead. Due to the introduction of new modules and an increased number of model parameters, our method do use a greater number of training epochs (uniformly set to 5,000 epochs) in the DeepH Benchmark Series compared to the official setting of baseline model, i.e., DeepH-E3 (https://github.com/Xiaoxun-Gong/DeepH-E3). We find that this is necessary for our method to fully converge in databases from the DeepH. For other training configurations, such as random seed, initial learning rate, optimizer, and scheduler, we ensured consistency with the DeepH-E3's official setup. For experiments on QH9, we match all training hyperparameters with the official configurations of QHNet (https://github.com/divelab/AIRS). The network hyperparameters of the baseline models, whether DeepH-E3 or QHNet, which are embedded in our method, also align exactly with their official settings. The implementation details of our experiments are provided in Appendix E.
>
> To fully resolve the reviewer’s concerns, we also conduct experiments in which we extend the training epochs of the baseline model, DeepH-E3, to match those of DeepH-E3+TraceGrad, setting a maximum training epoch of $5,000$ to examine whether this increase would improve accuracy of DeepH-E3. Due to time constraints in the rebuttal phase, we conduct these experiments on two databases as representative: Monolayer MoS2 ($MM$) and Bilayer Bi2Te3 ($BT$). For DeepH-E3, the original training epochs on $MM$ and $BT$ are $2,202$ and $2,520$, respectively, achieving average Hamiltonian matrix prediction errors ($MAE^H_{all}$) of $0.406$, $0.447$, and $0.831$, on $MM$, $BT^{nt}$, and $BT^{t}$, respectively; after extending the epochs to $5,000$, we observed only marginal decreases in error to $0.401$, $0.442$, and $0.827$, respectively, with no significant improvement. In contrast, DeepH-E3+TraceGrad achieved $MAE^H_{all}$ values of $0.285$, $0.295$, and $0.735$ on the respective datasets, demonstrating a substantial improvement over both sets of DeepH-E3 results. These findings indicate that the performance gains from our proposed TraceGrad method are robust and not primarily due to extended training epochs.
>
>
> Regarding computational overhead, in the revised paper’s Appendix J, titled "A Joint Discussion on GPU Time Costs and Performance Gains", we provide a joint comparison of the inference time cost and corresponding accuracy of different models across four databases as representative: Monolayer Graphene ($MG$), Monolayer MoS2 ($MM$), QH9-stable ($QS$), and QH9-dynamic ($QD$). This comparison includes the average inference time  per sample for each model, with the test batch size set as 1 and hardware environment set as Nvidia RTX A6000 GPU in single-task mode without computational sharing with other processes.
>
> For $MG$ and $MM$, we compare among DeepH-E3, DeepH-E3+TraceGrad, and DeepH-E3$^{×2}$, where DeepH-E3$^{×2}$ refers to a model obtained by doubling the number of encoding blocks in DeepH-E3 and training it from scratch until convergence. For $QS$ and $QD$, we compare among QHNet, QHNet+TraceGrad, and QHNet$^{×2}$, where QHNet$^{×2}$ refers to a model obtained by doubling the number of encoding blocks in QHNet and training it from scratch until convergence. The experimental results are documented in the table below:
>
> (Continued in Part 2)

---

> > ### Author Response · Authors · 2024-11-25
> > **Response to Question 2 (part 2)**
> >
> > (Continued from Part 1)
> >
> > **Table**: Average inference time per testing sample (abbreviated as Time) in seconds and average MAE performance in meV ($MAE^H_{all}$) on the Monolayer Graphene (MG), Monolayer MoS2 (MM), QH9-stable (QS), and QH9-dynamic (QD) databases. *↓* indicates that lower values of the metrics correspond to better accuracy.
> > | **Methods**              | **$MG$**                           |                              |                              | **$MM$**                           |                              |                              |
> > |--------------------------|------------------------------------|------------------------------|------------------------------|------------------------------------|------------------------------|------------------------------|
> > |                          | **$Time$ (↓)**                     | **$MAE^H_{all}$ (↓)**         |                              | **$Time$ (↓)**                     | **$MAE^H_{all}$ (↓)**         |                              |
> > | DeepH-E3 (Baseline)      | 0.247                              | 0.251                         |                              | 0.256                              | 0.406                         |                              |
> > | DeepH-E3$^{×2}$     | 0.483                              | 0.244                         |                              | 0.510                              | 0.387                         |                              |
> > | DeepH-E3+TraceGrad       | 0.264                              | 0.175                         |                              | 0.274                              | 0.285                         |                              |
> > | **Methods**              | **$QS$**                           |                              |                              | **$QD$**                           |                              |                              |
> > |                          | **$Time$ (↓)**                     | **$MAE^H_{all}$ (↓)**         |                              | **$Time$ (↓)**                     | **$MAE^H_{all}$ (↓)**         |                              |
> > | QHNet (Baseline)         | 0.233                              | 1.962                         |                              | 0.174                              | 4.733                         |                              |
> > | QHNet$^{×2}$        | 0.497                              | 1.845                         |                              | 0.385                              | 4.532                         |                              |
> > | QHNet+TraceGrad          | 0.248                              | 1.191                         |                              | 0.187                              | 2.819                         |                              |
> >
> >
> >
> > From this Table, we  find that incorporating the TraceGrad method results in only a slight increase in inference time compared to the baseline models. Given the substantial accuracy improvements introduced by the TraceGrad method, this minor increase in computational time is considered acceptable for practical applications. In contrast, simply increasing the depth of DeepH-E3 or QHNet leads to a significant rise in inference time while providing only limited accuracy improvements.
> > In contrast, DeepH-E3+TraceGrad and QHNet+TraceGrad exhibit significantly better accuracy performance compared to DeepH-E3$^{×2}$ and QHNet$^{×2}$, respectively. At the same time, the inference times of DeepH-E3+TraceGrad and QHNet+TraceGrad are considerably lower than those of DeepH-E3$^{×2}$ and QHNet$^{×2}$, respectively. These findings highlight the superiority of the TraceGrad method in enhancing model expressiveness and improving accuracy performance while maintaining computational efficiency.
> >
> >
> > In addition to reporting the GPU inference time of the deep models, we also present the inference times of QHNet and QHNet+TraceGrad on a single thread of an Intel(R) Xeon(R) Gold 6330 @ 2.00GHz CPU in single-task mode in the revised paper’s Appendix K, titled "Acceleration Performance for the Convergence of Traditional DFT Algorithms", which quantifies the acceleration performance of the deep learning methods for DFT computations. In these experiments, the predictions from deep models, e.g., QHNet or QHNet+TraceGrad, are used as initial matrices to accelerate the convergence of traditional DFT algorithms. Experimental results show that while combining TraceGrad introduces only a slight increase from the inference time of QHNet on the CPU, it delivers significant improvements in accelerating the convergence of DFT methods. Notably, the time saved by TraceGrad for DFT calculations  far exceeds the minimal additional time introduced by TraceGrad for the deep model's inference.

---

> ### Author Response · Authors · 2024-11-25
> **Response to Question 3**
>
> **Comment from the reviewer**: For electronic-structure Hamiltonian prediction tasks, what is the practically meaningful accuracy threshold? If, for instance, 1 meV accuracy is sufficient for most practical applications, how do we justify the value of further accuracy improvements (e.g., from 0.274 to 0.226 meV on BB dataset)? Could the authors discuss the practical significance of such fine-grained improvements in the context of real-world applications, such as materials design and DFT calculations?
>
> **Response**:
>
> We appreciate the reviewer's question on  "what is the practically meaningful accuracy threshold". Actually, the required precision for Hamiltonian matrices can vary significantly depending on the specific application context. The errors in the Hamiltonian effectively introduce noise into the electronic structure. As the reviewer correctly pointed out, in many general electronic structure calculations, a precision of 1 meV is often sufficient to meet practical needs. However, for applications that demand higher energy resolution, this level of precision may not be enough. For example, in magic-angle twisted bilayer graphene (MATBG), a system characterized by a very narrow flat band structure near the magic angle, the flat band drastically reduces the electron's kinetic energy, significantly enhancing electron-electron interaction effects. The width of this flat band is only a few meV, necessitating a highly precise Hamiltonian for accurate modeling. Even more stringent demands arise in quantum dot-based quantum computing, where the required energy resolution typically reaches the microelectronvolt ($\mu  eV$) level. In such systems, quantum states are exceptionally sensitive to the tiny energy differences, necessitating far more precision to the 1 meV level. Therefore, while a precision of 1 meV suffices for many applications, achieving sub-meV or even greater precision is crucial in high-accuracy scenarios, such as quantum computing. Such advancements not only enhance model robustness and versatility but also meet the stringent demands of specific applications, enabling the exploration of more complex and subtle quantum effects.
>
> Moreover, even assuming that the current application task faces a 1 meV precision requirement, there is still a need to continuously improve the precision of existing deep learning methods. It is important to note that the improvement in $MAE^H_{all}$ from 0.274 meV to 0.226 meV is achieved under the condition that  the original baseline method already performed well with the $MAE^H_{all}$ metric on the corresponding dataset. Thus, the improvement by our new algorithm is modest and may not fully reflect its potential. Yet, as shown in Table 2 of our paper, on the $QS$ and $QD$ databases, QHNet's $MAE^H_{all}$ are $1.962 \, \text{meV}$ and $4.733 \, \text{meV}$, far exceeding $1 \, \text{meV}$, while our method reduces these error metrics to $1.191 \, \text{meV}$ and $2.819 \, \text{meV}$, making significant progress towards the 1 meV precision level. Additionally, as shown in Fig. 6 of our paper, on the $MM$, $BB^{nt}$, $BB^{t}$, and $BT^{nt}$ datasets, DeepHE3's $MAE^H_{cha_b}$ (the MAE on the most challenging Hamiltonian block where the baseline model shows the poorest performance) are $1.103 \, \text{meV}$, $1.042 \, \text{meV}$, $2.399 \, \text{meV}$, and $1.387 \, \text{meV}$, respectively, while our method reduces these error metrics to $0.808 \, \text{meV}$, $0.740 \, \text{meV}$, $1.284 \, \text{meV}$, and $0.718 \, \text{meV}$, demonstrating substantial improvement in precision.
>
> Furthermore, on the $QS$ and $QD$ databases, we found that the accuracy of the occupied orbital energy $\epsilon$ and electronic wavefunction $\psi$ derived from our predicted Hamiltonians significantly surpasses the baseline results, with an improvement of up to 76\% and 17\%, respectively, as shown in Table 2 of our paper. These two quantities are critical for understanding various electronic properties, including optical characteristics, conductivity, and chemical reactivity of atomic systems, as they directly determine how electrons are distributed and interact within a system. The substantial improvements in both $\epsilon$ and $\psi$ demonstrate that our method not only excels in Hamiltonian prediction but also shows great promise in accurately capturing the fundamental physical properties of complex atomic systems.

---

> ### Author Response · Authors · 2024-11-25
> **Response to Question 4**
>
> **Comment from the reviewer**: In line 1116, The paper compares the O(N) complexity of Hamiltonian prediction with the O(N³) complexity of traditional DFT methods. However, this comparison seems unfair: the O(N³) complexity in traditional DFT includes solving the Kohn-Sham equations, while the O(N) complexity here only covers Hamiltonian prediction. To obtain physical properties from the predicted Hamiltonian, one still needs O(N³) operations for diagonalization. Could the authors clarify this comparison and provide a more comprehensive complexity analysis that includes all steps necessary for obtaining the same physical properties in both approaches?
>
> **Response**:
>
> Thanks for the question on the complexity of Hamiltonian prediction. In the traditional self-consistent solution of the Kohn-Sham equation, we must solve the eigenfunctions of all occupied states to obtain the charge density, leading to a computational complexity of $O(N^3)$. However, when using machine learning methods to predict the electronic Hamiltonian, the prediction process (for sufficiently large systems) achieves a computational complexity of $O(N)$. Moreover, since most physical properties depend only on the energy bands near the Fermi level, such as the transport, optical and topological properties of the physical system, we do not need to solve for the eigenfunctions of all occupied states once the Hamiltonian is known. Since the Hamiltonian matrix is sparse and only a few bands near the Fermi surface are needed, this can be efficiently achieved using methods like the shift-invert approach in the ARPACK package, with a complexity of $O(N)$.  We clarify this issue in Appendix I of the revised manuscript. This analysis has been added in Appendix I from line 1348 to line 1378 of the revised paper.

---

### Official Review · Reviewer_UXWk · 2024-11-03

**Soundness:** 2
**Presentation:** 2
**Contribution:** 3
**Rating:** 6
**Confidence:** 2

**Summary:**

This paper proposes a new framework that strictly satisfies the SO(3) equivalence and has the potential to enhance expressive power with nonlinearity. The effectiveness of the proposed ansatz is demonstrated with experiments on Hamiltonian prediction tasks and DFT.

**Strengths:**

-	Balancing between physics constraints (i.e. SO3-equivalence in this task) and the expressive power of the network is highly non-trivial. Therefore, an ansatz that both strictly satisfies the equivariance constraint and has the potential to leverage nonlinearity for expressiveness improvement is a noticeable contribution.
-	The authors test their new framework on various tasks

**Weaknesses:**

The main weakness of this paper is lack of readability. Many important background and detailed introduction to the problem, the task, and technical details are missing, making it hard to follow.

-	L156-160, it is very abrupt when concepts like Q and direct-product state are mentioned without any context. The *problem formulation* section should provide details (with clearly explained equations) on how the related concepts are defined, along with their physical meaning related to the task, which is also not clearly delivered.

-	In appendix A, it is hard to understand why the authors only include well-known concepts like definitions of group and equivariance but omit those more problem-specific and highly relevant ones, e.g. irreducible representation, properties of Wigner-D-matrix, Clebsch-Gordan decomposition, etc. These are crucial for readers to understand their methods. Hiding all these related contexts is not appreciated.

-	The learning task considered in the experiment section is not explicitly described. If the authors find themselves running out of pages, a detailed description on electronic-structure Hamiltonian prediction task should then be added in the appendix, including the physics background, the mapping to be learned, and methods adopted as well as model architecture. Same comments on the DFT experiment.

Also, I didn’t find a comparison on computation cost, e.g. GPU hours / FLOPS for the Hamiltonian prediction task.

**Questions:**

See Weakness.

---

> ### Author Response · Authors · 2024-11-25
> **Response to Reviewer 1**
>
> We thank you very much for your thorough and insightful feedback. We greatly appreciate the time and effort you have invested in reviewing our paper. We have carefully considered all of your comments and have addressed each one in detail in the revised manuscript. Your suggestions have been invaluable in enhancing both the clarity and comprehensiveness of our work, and we hope that our revisions align with your expectations. Please find below our point-by-point responses to your comments, along with details of the revisions made.

---

> ### Author Response · Authors · 2024-11-25
> **Response to Weakness 1**
>
> **Comment from the reviewer**: L156-160, it is very abrupt when concepts like Q and direct-product state are mentioned without any context. The problem formulation section should provide details (with clearly explained equations) on how the related concepts are defined, along with their physical meaning related to the task, which is also not clearly delivered.
>
> **Response**:
> Following the reviewer’s suggestion, we have incorporated the required definitions into Appendix A of the revised paper: Definition 9 for the direct-product state, Definition 10 for the direct-product physical quantity formed by two degrees, and Definition 11 for the direct-sum state.

---

> ### Author Response · Authors · 2024-11-25
> **Response to Weakness 2**
>
> **Comment from the reviewer**: In appendix A, it is hard to understand why the authors only include well-known concepts like definitions of group and equivariance but omit those more problem-specific and highly relevant ones, e.g. irreducible representation, properties of Wigner-D-matrix, Clebsch-Gordan decomposition, etc. These are crucial for readers to understand their methods. Hiding all these related contexts is not appreciated.
>
> **Response**: Following the reviewer’s suggestion, in the revised paper, we have added definitions in Appendix A  to introduce the concepts of irreducible representation (Definition 4), the Wigner-D matrix (Definition 6), and the Clebsch-Gordan decomposition for the SO(3) group (Definition 12). Additionally, we have provided the explicit formula for decomposing direct-product physical quantities $\mathbf{Q}^{l_p \otimes l_q}$ into direct-sum states via the Clebsch-Gordan decomposition.

---

> ### Author Response · Authors · 2024-11-25
> **Response to Weakness 3**
>
> **Comment from the reviewer**: The learning task considered in the experiment section is not explicitly described. If the authors find themselves running out of pages, a detailed description on electronic-structure Hamiltonian prediction task should then be added in the appendix, including the physics background, the mapping to be learned, and methods adopted as well as model architecture. Same comments on the DFT experiment.
>
> **Response**:
>
> To facilitate readers' understanding of the primary physical quantity studied in our work—the electronic-structure Hamiltonian—we have added a new section titled "Application Task Description: Electronic-structure Hamiltonian Calculation" in Appendix B of the revised paper. This section provides a comprehensive introduction to the task. We first highlight the central role of electronic-structure calculations in quantum mechanics, emphasizing that the Hamiltonian is a fundamental physical quantity in electronic structure, enabling the derivation of critical quantities such as the electronic wavefunction, orbital energy, and band structure, determining the electronic, optical, magnetic, and transport characteristics of the electron system.
>
> We then present the mathematical form of the Hamiltonian and describe the traditional DFT algorithm used to obtain and solve the Hamiltonian matrix, highlighting its high computational cost. This naturally leads to a discussion on the necessity and effectiveness of using deep learning methods for Hamiltonian matrix prediction. Furthermore, we clarify the challenge our method aims to address on deep learning Hamiltonian prediction.
>
> In addition, regarding the reviewer’s request for further details on the "methods adopted as well as model architecture", these topics are covered extensively in Section 5 and Appendix E of the revised paper. Regarding the details of the DFT calculations, these were performed using the PySCF package [1]. The experimental setup is identical to that described in Section 4 of the QH9 paper [2], except for the CPU environment, where we utilized a single thread of an Intel(R) Xeon(R) Gold 6330 @ 2.00GHz CPU in single-task mode. To ensure fairness in the comparison, when comparing the wall time of deep models with DFT calculations (Table 10 in the revised paper), all deep learning models are evaluated on the same CPU thread in single-task mode. This approach ensures consistency with the DFT software. Additionally, the experiments for the baseline method, QHNet, are reproduced under the same conditions to maintain a fair comparison.
>
> [1] Qiming Sun, Timothy C Berkelbach, Nick S Blunt, George H Booth, Sheng Guo, Zhendong Li, Junzi Liu, James D McClain, Elvira R Sayfutyarova, Sandeep Sharma, et al. PySCF: the Python-based simulations of chemistry framework. Wiley Interdisciplinary Reviews: Computational Molecular Science, 8(1):e1340, 2018.
>
> [2] Haiyang Yu, Meng Liu, Youzhi Luo, Alex Strasser, Xiaofeng Qian, Xiaoning Qian, and Shuiwang Ji. QH9: A quantum hamiltonian prediction benchmark for QM9 molecules. In NeurIPS, 2023a.

---

> ### Author Response · Authors · 2024-11-25
> **Response to Weakness 4 (part 1)**
>
> **Comment from the reviewer**:
> Also, I didn’t find a comparison on computation cost, e.g. GPU hours / FLOPS for the Hamiltonian prediction task.
>
>
> **Response**:
>
> In the revised paper’s Appendix J, titled "A Joint Discussion on GPU Time Costs and Performance Gains," we provide a comprehensive comparison of GPU computational costs and corresponding accuracy for different models across four representative databases: Monolayer Graphene (MG), Monolayer MoS2 (MM), QH9-stable (QS), and QH9-dynamic (QD). This comparison includes the average inference time per sample for each model, with the testing batch size set to 1. The hardware environment for all tests is an Nvidia RTX A6000 GPU operating in single-task mode without computational sharing with other processes.
>
> For $MG$ and $MM$, we compare DeepH-E3, DeepH-E3+TraceGrad, and DeepH-E3$^{×2}$, where DeepH-E3$^{×2}$ refers to a model obtained by doubling the number of encoding blocks in DeepH-E3 and training it from scratch until convergence. Similarly, for $QS$ and $QD$, we compare QHNet, QHNet+TraceGrad, and QHNet$^{×2}$, where QHNet$^{×2}$ represents a model obtained by doubling the number of encoding blocks in QHNet and training it from scratch until convergence. The experimental results are presented in the table below:
>
> **Table**: Average inference time per testing sample (abbreviated as Time) in seconds and average MAE performance in meV ($MAE^H_{all}$) on the Monolayer Graphene (MG), Monolayer MoS2 (MM), QH9-stable (QS), and QH9-dynamic (QD) databases. *↓* indicates that lower values of the metrics correspond to better accuracy.
> | **Methods**              | **$MG$**                           |                              |                              | **$MM$**                           |                              |                              |
> |--------------------------|------------------------------------|------------------------------|------------------------------|------------------------------------|------------------------------|------------------------------|
> |                          | **$Time$ (↓)**                     | **$MAE^H_{all}$ (↓)**         |                              | **$Time$ (↓)**                     | **$MAE^H_{all}$ (↓)**         |                              |
> | DeepH-E3 (Baseline)      | 0.247                              | 0.251                         |                              | 0.256                              | 0.406                         |                              |
> | DeepH-E3$^{×2}$     | 0.483                              | 0.244                         |                              | 0.510                              | 0.387                         |                              |
> | DeepH-E3+TraceGrad       | 0.264                              | 0.175                         |                              | 0.274                              | 0.285                         |                              |
> | **Methods**              | **$QS$**                           |                              |                              | **$QD$**                           |                              |                              |
> |                          | **$Time$ (↓)**                     | **$MAE^H_{all}$ (↓)**         |                              | **$Time$ (↓)**                     | **$MAE^H_{all}$ (↓)**         |                              |
> | QHNet (Baseline)         | 0.233                              | 1.962                         |                              | 0.174                              | 4.733                         |                              |
> | QHNet$^{×2}$        | 0.497                              | 1.845                         |                              | 0.385                              | 4.532                         |                              |
> | QHNet+TraceGrad          | 0.248                              | 1.191                         |                              | 0.187                              | 2.819                         |                              |
>
>
>
>
>
> (Continued in Part 2)

---

> ### Author Response · Authors · 2024-11-25
> **Response to Weakness 4 (part2)**
>
> (Continued from Part 1)
>
> From this Table, we  find that incorporating the TraceGrad method results in only a slight increase in inference time compared to the baseline models. Given the substantial accuracy improvements introduced by the TraceGrad method, this minor increase in computational time is considered acceptable for practical applications. In contrast, simply increasing the depth of DeepH-E3 or QHNet leads to a significant rise in inference time while providing only limited accuracy improvements.
> In contrast, DeepH-E3+TraceGrad and QHNet+TraceGrad exhibit significantly better accuracy performance compared to DeepH-E3$^{×2}$ and QHNet$^{×2}$, respectively. At the same time, the inference times of DeepH-E3+TraceGrad and QHNet+TraceGrad are considerably lower than those of DeepH-E3$^{×2}$ and QHNet$^{×2}$, respectively. These findings highlight the superiority of the TraceGrad method in enhancing model expressiveness and improving accuracy performance while maintaining computational efficiency.
>
>
> In addition to reporting the GPU inference time of the deep models, we also present the inference times of QHNet and QHNet+TraceGrad on a single thread of an Intel(R) Xeon(R) Gold 6330 @ 2.00GHz CPU in single-task mode in the revised paper’s Appendix K, titled "Acceleration Performance for the Convergence of Traditional DFT Algorithms", which quantifies the acceleration performance of the deep learning methods for DFT computations. In these experiments, the predictions from deep models, e.g., QHNet or QHNet+TraceGrad, are used as initial matrices to accelerate the convergence of traditional DFT algorithms. Experimental results show that while combining TraceGrad introduces only a slight increase from the inference time of QHNet on the CPU, it delivers significant improvements in accelerating the convergence of DFT methods. Notably, the time saved by TraceGrad for DFT calculations  far exceeds the minimal additional time introduced by TraceGrad for the deep model's inference.
>
> Unfortunately, due to incompatibility between the underlying equivariant operator library (E3NN) and the current PyTorch FLOPs profiling tools, we are currently unable to provide accurate FLOPs information. In future work, we plan to develop a custom library to enable precise FLOPs measurement on equivariant networks.

---

> ### Comment · Reviewer_UXWk · 2024-11-26
>
> Thanks for the response. My concern is resolved. I will upgrade my score.

---

> > ### Author Response · Authors · 2024-11-29
> >
> > We sincerely appreciate the time and effort you invested in reviewing our manuscript and carefully considering our response. Your thoughtful feedback and constructive suggestions have significantly contributed to improving our work. We are truly grateful that you found our revisions satisfactory and raised your evaluation. Thank you for your valuable insights and for supporting the development of our research.

---

### Author Response · Authors · 2024-12-02
**Letter to Area Chairs**

Dear Area Chairs:

We have completed the rebuttal process and sincerely thank the reviewers for their time and valuable comments, which have significantly helped us improve the quality of our paper.

In the revised paper, we have addressed all of the reviewers' comments and questions and have improved our paper in four aspects.

1.	We have improved the representation of the paper, clarifying the main learning tasks and some key concepts, thereby enhancing its readability.

2.	We have provided a clearer theoretical explanation of the effectiveness of the proposed method, included a more detailed comparison with related studies, and incorporated robust experimental evidence.

3.	We have supplemented more experiments to demonstrate the efficiency and effectiveness of our method.

4.	To demonstrate the generality of our method, we have supplemented our experiments to demonstrate its effectiveness on the additional task, namely energy and force field prediction.

Overall, we have thoroughly addressed the reviewers' concerns and refined our work according to their suggestions. The reviewers have responded positively, and a consensus has been reached regarding the evaluation scores.

**We would like to emphasize that our method demonstrates the significant potential of deep learning in addressing core tasks in quantum physics, representing a cross-disciplinary effort bridging AI and physics research. We believe this work holds substantial value for both the AI and quantum physics communities.**

We sincerely request your consideration of our work and would like to express our deep appreciation for your time and thoughtful oversight throughout the review process. Your support and guidance are invaluable to us.

Best regards,

The authors

---

> ### Author Response · Authors · 2024-12-04
> **Major Points of Revisions**
>
> To facilitate the AC's review, we have outlined the major points of revisions according to the structure of the paper. Please note that these are the main points. Additional discussions and clarifications regarding our approach and its implications, as well as responses to questions to help the reviewers better understand the work, along with minor wording corrections, can be found in the discussion boxes addressed to the reviewers.
>
>
> 1. In Section 2 of our paper, we revised our description of the gated activation mechanism to a more precise statement: "...Nevertheless, as demonstrated in our numerical study in Appendix H, multiplying SO(3)-equivariant features with linear coefficients may not be fully effective in enhancing non-linear expressiveness".
>
> 2. In Remark 1 of Section 4, we analyze the intrinsic relationship between Theorem 1 and Theorem 2 and explain the underlying principle behind using the gradient mechanism to induce SO(3)-equivariant features from SO(3)-invariant features.
>
> 3. We have added definitions in Appendix A to introduce the concepts of irreducible representations, the Wigner-D matrix, the direct-product state, the direct-product physical quantity formed by two degrees, the direct-sum state, and the Clebsch-Gordan decomposition for the SO(3) group.
>
> 4. To facilitate readers' understanding of the primary tasks and the physical quantity studied in our work—the electronic-structure Hamiltonian—we have added a new section titled ``Application Task Description: Electronic-structure Hamiltonian Calculation'' in Appendix B of the revised paper.  We first highlight the central role of electronic-structure calculations in quantum mechanics, present the mathematical form of the Hamiltonian, and describe the traditional DFT algorithm used to obtain and solve the Hamiltonian matrix, emphasizing its high computational cost. This naturally leads to a discussion on the necessity and effectiveness of using deep learning methods for Hamiltonian matrix prediction. Additionally, we clarify the challenge that our method aims to address in deep learning-based Hamiltonian prediction.
>
>
> 5. We have expanded Appendix E to provide more comprehensive implementation details.
>
> 6. We've added Appendix H, which reports the quantitative comparison between  the proposed gradient-based mechanism (Grad) with the existing gated activation mechanism (Gate), demonstrating better accuracy performance of the proposed gradient-based mechanism compared to the gated mechanism, indicating  that our method may be a better choice in terms of expressing complex non-linear mappings.
>
> 7. In Appendix I, we provide a more detailed theoretical analysis of the computational complexity advantage of our method over traditional DFT calculations.
>
>
> 8. We've added Appendix J,  a joint discussion on GPU time costs and performance gains brought by our TraceGrad method. Experimental results underscore the superiority of the TraceGrad method in enhancing model expressiveness and improving accuracy performance while maintaining computational efficiency.
>
>
>
> 9. In  the revised paper's Appendix K, titled ``Acceleration Performance for the Convergence of Traditional DFT Algorithms", we add a group of metrics measuring the wall time savings that deep learning methods contribute to DFT calculations, specifically quantifying the incremental time savings brought by our proposed TraceGrad method in accelerating DFT calculations. Experimental results show that while combining TraceGrad introduces only
> a slight increase from the inference time of the baseline model, it delivers significant improvements in
> accelerating the convergence of DFT methods, and the time saved by TraceGrad in DFT calculations far exceeds the minimal additional time required for its inference.
>
>
>
> 10. We have added experiments to validate the effectiveness of our method to energy/force field prediction. Experimental results on the MD17-aspirin dataset show that our TraceGrad method improves the prediction accuracy for both energy and force, indicating that our approach is not limited to Hamiltonian prediction tasks and demonstrates good transferability to other tasks. These new results and analysis will be added to our paper once it can be revised in the OpenReview system.
>
> 11. We have expanded Appendix M to provide a more detailed discussion on how this work might be applied to other tasks. Moreover, following the feedback from reviewer G9tS, we highlight that while our method is theoretically applicable to many tasks, its empirical effectiveness in those areas has yet to be demonstrated.

---

### Note · Authors · 2025-05-19

I have read and agree with the venue's withdrawal policy on behalf of myself and my co-authors.

---

### Meta-Review · Area_Chair_ou3F · 2024-12-23

**Metareview:**

The paper proposes a neural network architecture that preserves rotational equivariance for high-order tensors. The novelty is the introduced equivariant component constructed as the gradient of a nonlinear mapping on a scalar feature of the tensor. The architecture is applied to the DFT Hamiltonian prediction task which involves tensors in various orders, and preferred results are demonstrated.

Reviewers raised some concerns and insufficiencies, and after the rebuttal, they appreciated the noticeable efforts in providing more details in the background, more results on hardware and timing, accuracy-efficiency trade-off evaluation, more empirical comparisons with other architectures, demonstration on the effect to accelerate DFT in wall time, and extended results for predicting energy and force. Three out of four of them have raised their total scores as a result, and all scores are leaning towards positive.

Nevertheless, I still feel certain remaining concerns:
* Expressiveness comparison with other methods. If expanding the expression, the equivariant update in each component would be a nonlinear transformation on the scalar $\\mathrm{CGDecomp}(\\mathbf{f} \\otimes \\mathbf{f}, 0)$, times the gradient of $\\mathrm{CGDecomp}(\\mathbf{f} \\otimes \\mathbf{f}, 0)$, which is in turn a specific scalar times the tensor f. In comparison, many other equivariant architectures also follow this form of a nonlinear transformation on the scalar $\\mathrm{CGDecomp}(\\mathbf{f} \\otimes \\mathbf{f}, 0)$ times the tensor feature $\\mathbf{f}$. The authors are expected to provide more discussion/rationale on why the proposed formulation could be more expressive.

* Conservativeness. The gradient of a scalar function makes a conservative vector/tensor field (function), but not all vector/tensor fields are conservative. This is not a problem (even a desired thing) in machine learning force field since force fields are conservative. But the authors are expected to explain why it is still the case for predicting the Hamiltonian. Although the authors did not bing it up in the paper, the DFT Hamiltonian is the gradient of electronic energy w.r.t the ground-state density matrix (in the DFT sense). But the gradient taken in the model is not w.r.t a tensor that holds the semantics of a density matrix, and even if there is, the ground-state density matrix still needs to be inferred from the input conformation. I noticed the authors made a discussion on this point in response to Reviewer G9tS, but it does not seem to reflect my question.

  Moreover, in the domain of machine learning force field, it is an empirical observation that the gradient formulation does not seem to better fit the force than directly predicting the vector, possibly due to that neural networks are not typically designed to produce expressive gradient functions, and may have numerical difficulties for the gradient. The authors may also need to discuss this point when motivating the gradient-based approach.

* On learning the SO(3)-invariant quantity $\\mathbf{T} = \\mathrm{tr}(\\mathbf{Q} \\mathbf{Q}^\\dagger)$ simultaneously. I still have the concern why this could be consistent with letting the gradient learning an arbitrary equivariant target. The argument using $\\mathbf{Q} = \\frac{\\partial \\mathbf{T}}{\\partial \\mathrm{Conj}(\\mathbf{Q})}$ seems vague: the gradient in the model is not taken w.r.t $\\mathrm{Conj}(\\mathbf{Q})$ but w.r.t the $\\mathbf{f}$ equivariant feature, and if the authors were suggesting $\\mathbf{f}$ could learn $\\mathrm{Conj}(\\mathbf{Q})$, then what the gradient produces is just $\\mathbf{Q}$, which may restrict the capability in fitting the equivariant Hamitonian target.

I communicated these points with reviewers, and Reviewer G9tS expressed the same remaining concerns regarding the conservativeness, and the potential conflict between fitting $\\mathbf{T}$ and $\\mathbf{Q}$ simultaneously, as the explanation and derivation presented in the paper seem insufficient, despite the empirical results. (He/She also resolved my question on the linear scaling.)

As the reviewers all rated only a borderline accept, and there still remains concerns, I tend to recommend a reject for the current time, and encourage the authors to present a deeper understanding of their proposed architecture in future versions.

**Additional Comments On Reviewer Discussion:**

All the reviewers appreciated the novelty of using the differentiation form of a nonlinear scalar transformation to construct equivariant high-order tensor transformations, and the effectiveness over a range of applications. Reviewers also raised some concerns and insufficiencies. After the rebuttal, they appreciated the noticeable efforts in providing more details in the background, more results on hardware and timing, accuracy-efficiency trade-off evaluation, more empirical comparisons with other architectures, demonstration on the effect to accelerate DFT in wall time, and extended results for predicting energy and force. Three out of four of them have raised their total scores as a result, and all scores are leaning towards positive. I agree with these points, but I still have concerns as listed in the Metareview, which make me feel hesitant to accept the paper in the current form.

---

### Decision · Program_Chairs · 2025-01-22

Reject